# Seasonal, regional and vertical characteristics of high carbon monoxide plumes along with their associated ozone anomalies as seen by IAGOS between 2002 and 2019

Thibaut Lebourgeois[1,2], Bastien Sauvage[1], Pawel Wolff[3], Béatrice Josse[2], Virginie Marécal[2], Yasmine Bennouna[1], Romain Blot[1], Damien Boulanger[3], Hannah Clark[4], Jean-Marc Cousin[1], Philippe Nedelec[1], and Valérie Thouret[1]

[1]Laboratoire d'Aérologie, Université de Toulouse, CNRS, UPS, Toulouse, France
[2]CNRM, Université de Toulouse, Météo-France, CNRS, Toulouse, France
[3]Observatoire Midi-Pyrénées, Université de Toulouse, CNRS, UPS, 31400 Toulouse, France
[4]IAGOS-AISBL, 98 Rue du Trône, Brussels, Belgium

**Correspondence:** Thibaut Lebourgeois (thibaut.lebourgeois@aero.obs-mip.fr)

In-situ measurements from IAGOS are used to characterise extreme values of carbon monoxide (CO) in large regions of the globe in the troposphere between 2002 and 2019. The SOFT-IO model, combining the FLEXPART lagrangian dispersion model with emission inventories over the footprint region is used to identify the origins of the CO in the sampled plumes. The impact of biomass burning and anthropogenic emissions on such CO plumes are characterised through CO mixing ratios and simultaneously recorded ozone ($O_3$) ones.

In the Northern Hemisphere, CO maximum are reached in DJF in the lower troposphere because of the elevated anthropogenic emissions and reduced convective activity of the season. Due to the low photochemistry and the fresh age of the air masses the $O_3$ values of these plumes are low. CO plumes in the upper troposphere result from intense emissions and efficient vertical transport, peaking during JJA. The largest values of CO in the northern hemisphere are found in Eastern Asia in the lower and middle troposphere and in Siberia in the the upper troposphere.Among the anomalies detected in the upper troposphere in JJA, the ones with the higher associated $O_3$ values are the ones associated with biomass burning emissions. The middle troposphere combines the two previous vertical levels with contributions from both local emissions and long-range transport. Among the studied regions, the troposphere above Middle-East and the UT of Siberia presented extremely high O3 values.

Indian CO anomalies have drastically different characteristics depending on the season as the wet and dry phases of the monsoon have an important impact on the transport of the pollutant in these regions.

Similarly the shift of the inter-tropical convection zone drastically impacts the seasonality of the emissions and the transport patterns above Africa. In that region convection is no longer the limiting factor and the transport of the CO plumes is driven by the ITCZ shift, trade winds and the upper branch of the Hadley cell redistributing the pollution to higher latitudes.

# 1 Introduction

Extreme weather can sometimes be incorrectly reproduced and predicted by the global and regional models (e.g. Shastri et al. (2017); Lavaysse et al. (2019)). Extreme pollution events can also be difficult to predict, as they can be explained by multiple factors such as abnormal weather conditions and/or unusually intense emissions (either from anthropogenic or natural sources, or both). Hence, it is essential to better understand the distributions of pollutants or their precursors in the atmosphere under such circumstances, leading thus to a better representation by the models and an improvement of their ability to predict their peak values as well as their impact on climate. Among the short-lived climate forcers, tropospheric ozone ($O_3$) is a key component of our atmosphere, and carbon monoxide (CO) is one of its main precursors. First, $O_3$ is a pollutant dangerous for human life (Chen et al., 2007; Liu et al., 2018) and for crops (Fuhrer et al., 1997; Davison and Barnes, 1998; Ashmore, 2005). Secondly, it is a trace gas with major influence on the oxidative capacity of the atmosphere as it is the main source of hydroxyl radicals in the troposphere (Seinfeld and Pandis, 2008). Finally, it is a greenhouse gas (GHG) with a positive radiative effect in the troposphere. Moreover multiple studies have shown the upper troposphere-lower stratosphere (UTLS) to be the region with the largest changes in radiative effect from changes in $O_3$ mixing ratio (Riese et al., 2012; Xia et al., 2018).

$O_3$ can hence have an impact on air quality as much as on climate. This compound in the troposphere is photochemically produced from NOx and VOC (Volatile Organic Compounds) or CO (Seinfeld and Pandis, 2008). Hence, a good estimation of its chemical precursors as well as better understanding of the processes leading to their distributions at global scale is of prime importance.

For these reasons, this study is focused on analysing the most intense anomalies of CO throughout the troposphere over different regions of the world and how $O_3$ distributions behave in such plumes.

Apart from being a precursor of $O_3$, CO is also one of the biggest sinks of hydroxyl radical (Lelieveld et al., 2016) and thus has an impact on the oxidative capacity of the atmosphere which can lead to increased lifetimes of other greenhouse gases such as $CH_4$. Moreover, the oxidation of CO produces greenhouse gases like $O_3$ and $CO_2$. CO is hence believed to cause an indirect positive radiative forcing (IPCC, 2013). Finally, CO is a good tracer for pollution export pathways thanks to its long chemical lifetime in the troposphere of a few weeks in summer to a few months in winter (Lelieveld et al., 2016).

CO is mostly emitted in the planetary boundary layer (BL) and can be removed via different mechanisms. These mechanisms highly depend on the regions and seasons. Convective activity represents an important part of the pollution export pathways from the BL. Some regions are more prone than others to exporting pollutants. Tropical regions benefit from permanent convective activity due to the close proximity of the Inter-Tropical Convergence Zone (ITCZ) (Andreae et al., 2001; Lannuque et al., 2021). Regions like south and eastern Asia benefit from the different phases of the monsoon season (Ricaud et al., 2014; Kar et al., 2004; Park et al., 2007; Lawrence, 2004a) or cold front and warm conveyor belt activity (Liang et al., 2004; Ding et al., 2009; Dickerson et al., 2007). North American pollution is mostly exported through cold front and warm conveyor belt (Owen et al., 2006; Cooper and Parrish, 2004). CO from biomass burning in boreal regions can be emitted directly above the BL and as high as the upper troposphere (UT) through pyroconvection whereas tropical fires emit mainly in the lower troposphere (Rémy et al., 2017; Val Martin et al., 2010; Damoah et al., 2006). Once in the free troposphere, CO is

transported via westerlies or jet streams and can be rapidly transported across the hemisphere (Stohl, 2001; Stohl et al., 2002)

and influence the atmospheric composition of a downwind continent (Liang et al., 2004; Cooper et al., 2004). In special cases, heavily polluted air masses can reach the UT (e.g. Nedelec et al. (2005)). Those events happen when polluted air masses are transported upward by strong (pyro-)convective episodes and can have a relatively large impact on the chemistry in the UT.

CO is one of the only $O_3$ precursors with a chemical lifetime long enough to reach the UT, so in this part of the atmosphere CO is hence believed to have an impact on $O_3$ mixing ratio as long as reservoirs for NOx are available (Seinfeld and Pandis,

2008).

Moreover, large values of CO in the UT are an indication of surface influenced air masses potentially rich in many pollutants, which illustrates the importance of better understanding phenomena able to bring vast amounts of CO in the upper part of the troposphere.

Studies on the export of large quantities of CO in the free troposphere or above have been facilitated with the access to

satellite data. An important number of studies have been focused on the eastern/southern part of Asia and especially on the export of the CO emitted into different regions (e.g. Fadnavis et al. (2011); Barret et al. (2016); Smoydzin and Hoor (2022)). Barret et al. (2016) used data from IASI onboard MetOp-A satellite in order to analyse the provenance of the pollution in the upper tropospheric South Asian Monsoon Anticyclone (SAMA) and showed that emissions from the Indo-Gangetic plain were uplifted during the Asian summer monsoon and trapped in its upper level anticyclone. Smoydzin and Hoor (2022) recently used

MOPITT to investigate large CO anomalies in the North Pacific and attributed those extremes to emissions from East Asia. Some studies have used the IAGOS database to analyse the characteristics of CO and $O_3$ values in the troposphere and lower stratosphere. This is the case for Cohen et al. (2018), which used this dataset to study the climatology and trends in O3 and CO in the UTLS. Petetin et al. (2018b); Lannuque et al. (2021); Tsivlidou et al. (2022) used IAGOS to study the characteristics of CO in different regions or altitude layers of the world. Tsivlidou et al. (2022) studied CO and O3 characteristics in the tropical

regions. She highlighted the origins of the CO in the different regions of the tropics. She especially showed the importance of the Anthropogenic emissions to explain the values of CO in the tropical troposphere. Lannuque et al. (2021) studied the meridional distribution of O3 and CO over Africa using IAGOS and the satellite IASI (Infrared Atmospheric Sounding Interferometer). They showed the importance of the ITCZ and the upper branch of the Hadley cell for the redistribution of the pollutants over Africa. The Pollutant emitted at the surface is transported by trade winds toward the ITCZ where it is transported to the UT

and redistributed to higher latitude by the Hadley cell. Petetin et al. (2018b) studied the CO vertical profile over different airport clusters. They characterised their seasonal profile as well as the seasonality of the highest CO anomalies (95 and 99 percentiles). They showed a strong seasonal variability of the most extreme anomalies in northern America which were due to BB emissions. He also looked at the origins of the CO responsible for the CO anomalies at the different airport clusters.

This emphasises the importance of transport when studying CO extremes in remote parts of the atmosphere. Most of the

studies cited above focused on the export of plumes of high CO mixing ratios in one region at a certain altitude and only a few of them were focused on the most extreme CO anomalies. IAGOS (In-service Aircraft for a Global Observing System; http://www.iagos.org) is a European research infrastructure using commercial aircraft in order to measure the atmosphere

composition. Thanks to the IAGOS database, we benefit from a large, and long-term in-situ sampling of the atmosphere, complementing the dedicated field campaigns and more global satellite datasets.

The goal of this paper is to characterise the seasonal, regional and vertical CO mixing ratios anomalies for different regions over the globe for almost 20 years as seen by IAGOS along with the simultaneously recorded $O_3$ between 2002 and 2019. The analysis will explore CO anomalies and their source type (anthropogenic vs biomass burning) and region of emission (14 defined regions of the Global Fire Emissions Database (GFED) (Giglio et al., 2013)). It aims at characterising the distributions and origins of extreme events of polluted plumes in terms of (i) mixing ratios of CO and O3, (ii) frequency and seasonality

at different altitudes. $O_3$ values are presented as additional information, characterising thus the average $O_3$ content in those plumes of extreme CO. Note that detailed analysis of the $O_3$ values is outside the scope of the current paper. Such characteristics will form a set of diagnostics that are of particular importance to further test the ability of the models to reproduce extreme events and their impact on the distributions of CO and $O_3$ throughout the troposphere.

## 2    Methods and materials

## 2.1    IAGOS dataset

The data used in this study is from the European research infrastructure IAGOS (Petzold et al., 2015; Thouret et al., 2022)), which has measured different trace gases, particles and meteorological components from passenger airplanes over several decades. IAGOS builds on the experience of the MOZAIC programme (Marenco et al., 1998), which was originally set up in 1994. $O_3$ and water vapour were the initial compounds measured, with CO measurements added in December 2001. $O_3$ and

CO are measured with an UV and infrared absorption photometer respectively (Thouret et al., 1998; Nédélec et al., 2015), with a total uncertainty of $\pm$ 2 ppb $\pm$ 2% for $O_3$ and $\pm$ 5 ppb $\pm$ 5% for CO with a time resolution of 4 seconds and 30 seconds respectively. IAGOS took over from MOZAIC in 2011, including an overlap period between 2011 and 2014 (Petetin et al., 2016). The IAGOS European Research Infrastructure also includes the predecessor complementary program CARIBIC (Brenninkmeijer et al., 1999). The consistency between the MOZAIC, IAGOS and CARIBIC datasets are regularly checked

following the methodology of Nédélec et al. (2015) and Blot et al. (2021) to ensure the internal consistency of the CO and $O_3$ measurements since 1994.

As this study focuses on CO and $O_3$, the dataset used is from the start of the CO measurement (January 2002) to December 2019. This dense network of measurements allows an unprecedented number of pollution events to be sampled for an in-situ dataset with a higher vertical resolution than satellite datasets. In total, more than 43 000 flights were performed by the different

aircraft during this period. These flights were performed by 10 different airlines allowing the in situ measurements in several regions of the world. In addition, each flight takes two vertical profiles (during take-off and landing). In contrast with other in-situ datasets from field campaigns, IAGOS is not dedicated to the study of a single phenomenon but rather to the long-term sampling of the atmosphere. This makes the large and precise IAGOS data set particularly suitable for a thorough analysis of the variability of the CO anomalies (see section 2.3.2 for the formal definition) in the different parts of the troposphere.

## 2.2 SOFT-IO, The source-receptor link

Since IAGOS is not a research project focused on the study of one phenomenon of the atmosphere, but a global exploratory observing system sampling the atmosphere regardless of its current state, a tool was needed to get information on the type of source influencing the air mass (biomass burning or anthropogenic emissions). This is the main usage of the SOFT-IO model.

SOFT-IO is described in detail in Sauvage et al. (2017) and used in scientific studies (e.g. Petetin et al. (2018b); Lannuque et al. (2021); Cussac et al. (2020); Tsivlidou et al. (2022)) so only a broad description of the model is given here. SOFT-IO is a model based on FLEXPART (Stohl et al., 2005) and emission inventories of anthropogenic and biomass burning sources (described below) along the IAGOS flight tracks. A 20-day back trajectory is coupled to the emissions inventories to calculate the CO contributions from recent emissions. The model uses wind fields from ERA interim with a horizontal resolution of $1° \times 1°$ and 137 vertical levels.

The biomass burning inventory used in this version of SOFT-IO is the 1.2 version of the Global Fire Assimilation System (GFAS) (Kaiser et al., 2012). The horizontal resolution is $0.1° \times 0.1°$ with a daily temporal resolution. The emission top altitude is provided by GFAS (v1.2), and is calculated using the fire plume rise model (Paugam et al., 2015; Rémy et al., 2017). GFAS was chosen for its temporal resolution as well as its ability to model emission height. The anthropogenic emissions are from the Community Emissions Data System (CEDS) (McDuffie et al., 2020) with a resolution of $0.5° \times 0.5°$ and a monthly temporal resolution.

SOFT-IO models the CO source contributions and the geographical origin of the emitted CO. The geographical origin of the modelled CO is defined by the same 14 regions as defined in the GFED project (see Fig.1). These contributions cannot be directly compared with observations because SOFT-IO only models contributions from recent emissions (and not older contributions nor the background mixing ratio). SOFT-IO is therefore used here as a qualitative tool to assess whether the modelled contributions are mainly due to anthropogenic or biomass burning emissions and to label the corresponding observed plume as such.

Sauvage et al. (2017) and Tsivlidou et al. (2022) made a thorough statistical evaluation of SOFT-IO. The model had a really good score in the detection frequency of the CO anomalies (above 93% on average). Detection frequency was at its maximum in the LT as most anomalies are from local emissions at this altitude. In the MT and UT the scores were lower but remained above 80% as the simulation of horizontal and vertical transport could suffer some errors. It is important to note that the study presented here aims at using soft-io only as a qualitative tool to attribute a source type and a relative geographical origin to the emissions leading to the detected anomalies. SOFT-IO is a model which has already been used in several studies similar to the current study (e.g Petetin et al. (2018b); Cussac et al. (2020); Lannuque et al. (2021); Tsivlidou et al. (2022)).

In addition to the various observed parameters and to the SOFT-IO products, the IAGOS Data Centre provides some meteorological fields from the ECMWF operational analysis interpolated along the IAGOS flight track, as ancillary data (https://doi.org/10.25326/3) . Among these parameters (potential temperature, planetary boundary layer height potential vorticity), the potential vorticity (PV) is used to define whether the CO observations are above or below the dynamical tropopause (defined at 2PVU as in Thouret et al. (2006); Cohen et al. (2018)).

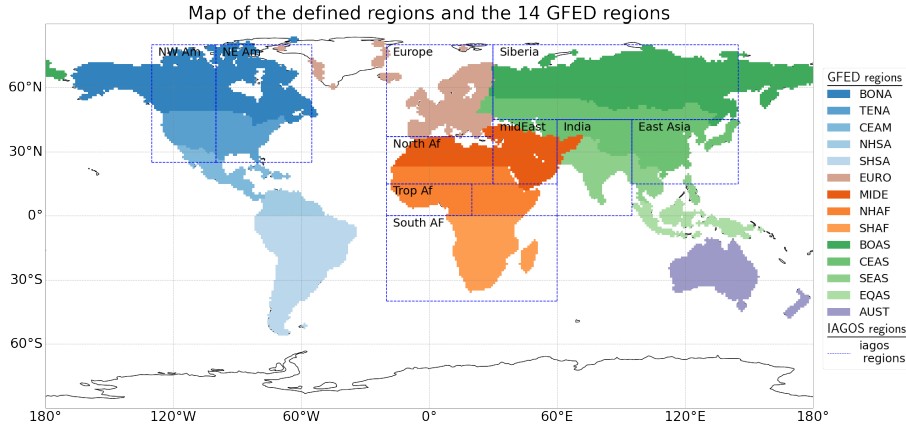

**Figure 1.** Map of the IAGOS regions (dotted lines) and the GFED defined regions (in colour) (see table A1 in the appendix for the full names of the acronyms)

## 2.3 Methods

### 2.3.1 Regions and seasons

In order to synthesise the seasonal and regional characteristics of the CO anomalies, the observations from IAGOS are split into different regions. These regions are defined to be characteristic of specific meteorological and chemical regimes (sources of precursors) similar to Cohen et al. (2018). This study is dedicated to the higher values of the CO distribution, so the sizes of the regions are larger here in order to increase the number of data points per region and not miss any extreme events. By nature, as IAGOS uses commercial aircraft to sample the atmosphere, the different regions are not sampled equally over the same time period (see Fig.B1), but a minimum of 1500 flights per region over the full sampling period is required.

Fig.1 shows the 10 regions defined and used in this study (dotted line). In addition, the colours of the map indicate the regions as defined by GFED which are used to assign an origin to the emitted CO (see 2.2 and see table A1 in the appendix for the full names of the acronyms). Fig.B2 in the appendix shows the IAGOS flight-tracks of the 19 years of data. It is important to keep in mind that we are studying data from aircraft measurements so they follow specific trajectories. Fig.B3 in the appendix shows the position of each of the visited airport by IAGOS aircraft. The lower and middle troposphere are sampled by IAGOS during landing and takeoff of the aircraft so in proximity of these airports. Note that the average horizontal distance between airport surface and the 8 km altitude is about 300 km (Petetin et al., 2018a). Fig.B1 in the appendix shows the availability of the data in each region. The number of flights is maximum over Europe due to the history of IAGOS and the dense traffic between the US and Europe. However, since 2006, regular flights from Europe to South Africa have been added. In addition, regular flights to eastern and equatorial Asia have been added since 2012.

In the northern hemisphere mid-latitudes (NW Am, NE Am, Eur, Sib and E Asia), four periods of three months are defined according to the meteorological seasons (DJF, MAM, JJA, SON). Note that this study focuses only on the boreal summer

and winter periods, one characterising the maximum of CO due to anthropogenic emissions in winter and the other one the
maximum of forest fire activity in summer (section 3.1). The two transitional periods are not presented here as we intend to
highlight the influence of the biomass burning emissions on the CO signal. In Africa, the seasons are defined according to
the shift of the ITCZ and the rainy seasons (as defined in Lannuque et al. (2021)) : DJFM and JJASO. The results for the
two transitional periods (April-May and November) are not presented here. For the Middle-East, seasons of interest for this
study are defined the same way as for Africa, because DJFM and JJASO there, also correspond to the maximum and minimum
of the CO seasonal cycle, respectively (Figs.C1 and C2). Furthermore, the Middle-East is connected to Africa in terms of
emissions as seen in Fig.1 (section 3.3). India (as defined Fig.1) is also an interesting region regarding the different influences
of biomass burning and anthropogenic emissions from the different continents. Differently from Northern mid-latitudes and
Northern Africa or Middle-East, the four seasons will be discussed here for India (section 3.2).

Finally, in order to characterise these CO extremes at different altitudes, the data sets are divided into three vertical layers.

– Lower Troposphere (LT): from the surface to 2000 m.

– Middle Troposphere (MT): from 2000 m to 8000 m.

– Upper Troposphere (UT): Above 8000 m and below the dynamical tropopause (defined as 2 PVU like in Thouret et al.
(2006); Cohen et al. (2018)).

IAGOS samples the lower and free troposphere during the landing and take-off. Petetin et al. (2018a) showed that close to the
surface, the IAGOS measurements are representative of urban areas and provide similar measurements to urban background
stations. At higher altitudes, in the free troposphere, the samples are less influenced by local emissions and therefore are
representative of regional background conditions following the flight tracks showed in Fig.B2 in the appendix.

### 2.3.2  Definition of the CO anomalies

Fig.2 illustrates the detection of two plumes in the IAGOS observations. The CO anomalies are defined as CO values exceeding
the threshold for three consecutive measurements (i.e. a distance of approximately 3 km during cruise phase). The chosen
threshold used in this study is the 95th percentile (q95) calculated for each region/season/altitude range (see table 1). The
number of observed anomalies per region can be found in the appendix (table A4).

Only data considered as CO anomalies are examined here. The selection process is repeated for each flight.

SOFT-IO is then used as a qualitative tool to assign a source type to each of the detected anomalies. This diagnostic is only
applied if the contributions modelled by SOFT-IO are above a detection threshold defined as 5 ppb. Several thresholds were
tested during this study and did not have a significant impact on the results. According to the method used in Petetin et al.
(2018b) the four categories are defined as follow :

– Anthropogenic: if the anthropogenic contributions calculated by SOFT-IO are at least twice those of the biomass burning.

– Biomass burning : if the biomass burning contributions calculated by SOFT-IO are at least twice the anthropogenic
contributions.

| | | LT | FT | UT |
|---|---|---|---|---|
| NW Am | DJF | 256 | 160 | 146 |
| | MAM | 255 | 170 | 171 |
| | JJA | 251 | 149 | 145 |
| | SON | 243 | 141 | 120 |
| NE Am | DJF | 264 | 159 | 126 |
| | MAM | 246 | 166 | 156 |
| | JJA | 241 | 156 | 132 |
| | SON | 241 | 140 | 112 |
| Eur | DJF | 332 | 158 | 126 |
| | MAM | 267 | 164 | 140 |
| | JJA | 200 | 140 | 123 |
| | SON | 253 | 138 | 109 |
| Sib | DJF | no data | no data | 127 |
| | MAM | no data | no data | 146 |
| | JJA | no data | no data | 181 |
| | SON | no data | no data | 123 |
| E Asia | DJF | 559 | 209 | 129 |
| | MAM | 504 | 265 | 185 |
| | JJA | 441 | 173 | 162 |
| | SON | 457 | 180 | 159 |

| | | LT | FT | UT |
|---|---|---|---|---|
| India | DJF | 424 | 157 | 132 |
| | MAM | 305 | 191 | 130 |
| | JJA | 267 | 134 | 131 |
| | SON | 470 | 150 | 150 |
| North Af | DJFM | no data | no data | 145 |
| | AM | no data | no data | 156 |
| | JJASO | no data | no data | 110 |
| | N | no data | no data | 124 |
| Middle E | DJFM | 253 | 148 | 135 |
| | AM | 272 | 143 | 131 |
| | JJASO | 300 | 129 | 113 |
| | N | 244 | 127 | 118 |
| Gulf of G | DJFM | 724 | 297 | 190 |
| | AM | 419 | 203 | 171 |
| | JJASO | 280 | 192 | 147 |
| | N | 383 | 199 | 155 |
| South Af | DJFM | 219 | 132 | 172 |
| | AM | 272 | 120 | 148 |
| | JJASO | 400 | 245 | 197 |
| | N | 247 | 150 | 153 |

**Table 1.** q95 values (in ppb) used as thresholds for the different regions for different seasons. For North Africa and Siberia no airports are visited by IAGOS aircrafts so there is no data available for the MT and LT layers.

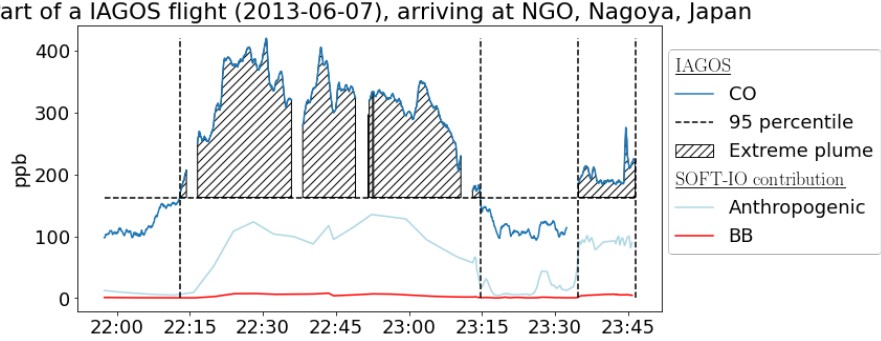

**Figure 2.** Illustration of the method used to define the CO anomalies applied to part of a IAGOS flight (data here are taken over Siberia at cruise altitude (around 250 hPa) on the day/year). The dark blue line represents the CO measured by IAGOS. The horizontal dashed line represents the seasonal and regional 95th percentile of the IAGOS dataset (181 ppb). It is used as the threshold for the CO anomalies in this region and season, at this altitude level (UT). The hatched area represents the defined anomalies. The light blue and red lines represent the modelled anthropogenic and biomass burning contributions modelled by SOFT-IO. The gaps are missing data.

– MIX sources: if none of the contributions, as calculated by SOFT-IO, is twice the other.

– Observed by IAGOS but undetected by SOFT-IO.

In Fig.2 both plumes of high CO mixing ratios are clearly attributed to anthropogenic sources. In addition, SOFT-IO provides information on the emitting region of the contributions (see section 2.2). This diagnosis is repeated for each plume detected. Thus, we can compute the main emitting region responsible for all detected plumes.

## 3   Results:

The first part of the results is dedicated to the five regions in the Northern hemisphere mid-latitudes (NW and NE America, Europe, Siberia and East Asia), then India, and Africa (North Africa, the Gulf of Guinea and South Africa) plus the Middle-East. Each vertical layer will be treated individually from the lower troposphere (LT) to the upper Troposphere (UT) (Siberia and North Africa are only sampled in the UT). The characteristics of the CO-extreme plumes will be given before presenting the associated $O_3$ distributions in such plumes.

### 3.1   Northern Hemisphere mid-latitudes

### 3.1.1   Lower troposphere (LT)

In this layer, our data are similar to urban background stations (Petetin et al., 2018a).

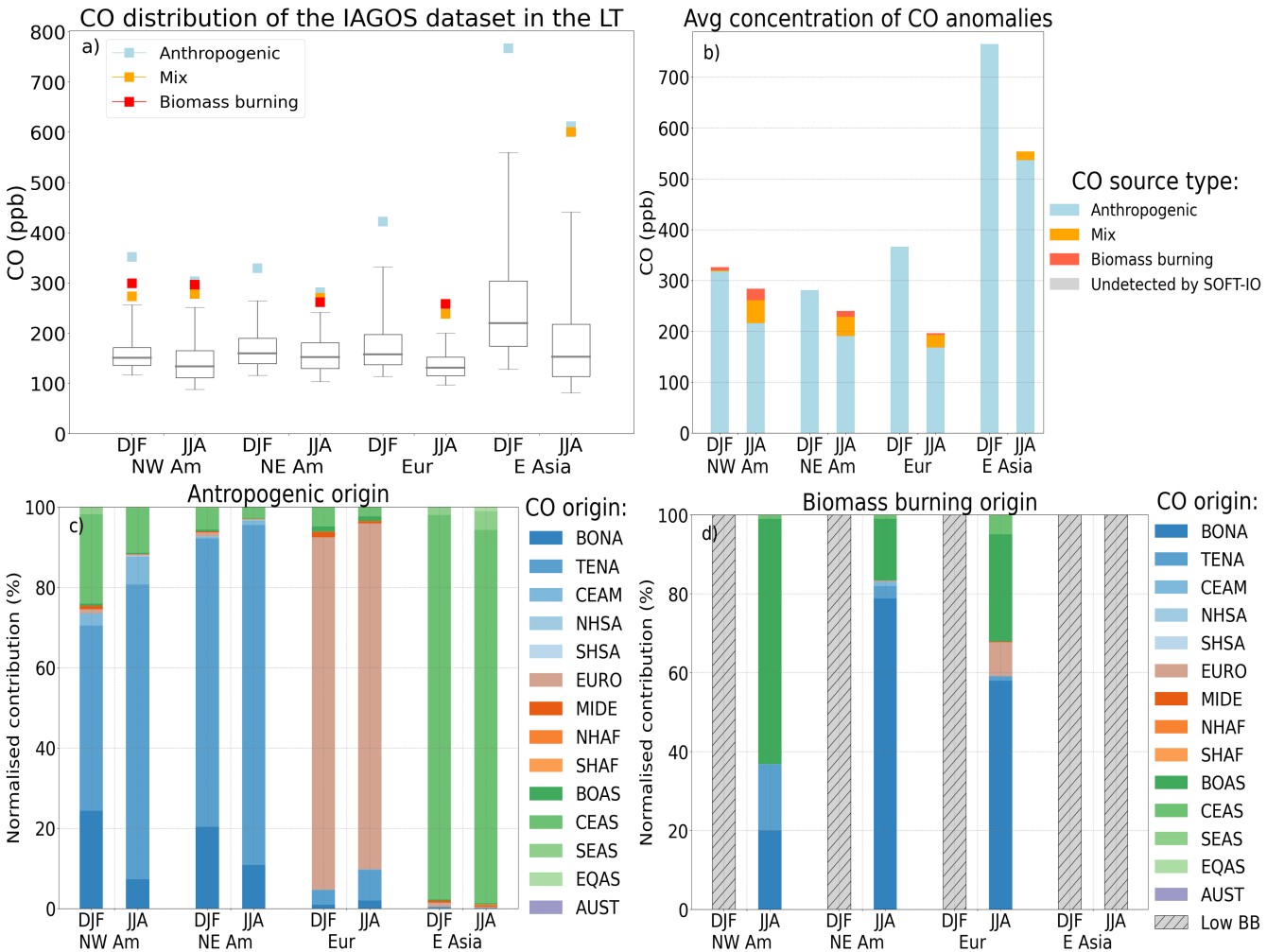

**Figure 3.** a) CO measured by IAGOS in the LT (below 2km). The box-plot represents the 5th, 25th, 50th, 75th and 95th percentiles of the CO distribution, while the coloured squares represent the mean values of CO inside the detected anomalies (each colour represents a type of CO anomaly attributed to a different source with SOFT-IO: red for biomass burning, blue for anthropogenic and orange for MIX sources). b) Bar-plot showing the averaged mixing ratios of CO in all the detected anomalies (>q95) in the LT in each region for JJA and DJF (given by the total height of the bar), and their proportion according to the different sources (blue for anthropogenic, red for biomass burning and orange for mix, the relative height of the coloured blocks represents the proportion of each type of anomalies). The proportion of plumes where no contribution is modelled by SOFT-IO are represented in grey (in this figure no anomalies are undetected by SOFT-IO over the 4804 observed). c) Regional origin (according to GFED regions, as in Fig. 1) of the anthropogenic contributions of the anomalies associated with MIX and anthropogenic sources in the LT in NH extra-tropics (the hatched part cover region/season with not enough anomalies attributed to the MIX or anthropogenic categories) d) Same for the origin of the biomass burning contributions associated with MIX and biomass burning anomalies. The Low BB patched (hatched grey patches) is applied if a regions has less than 3% of its plumes attributed to either MIX or BB sources.

In the LT (Fig.3) of most regions the distribution of CO is higher in DJF than in JJA due to the higher anthropogenic emissions during the winter months (e.g mean of the anthropogenic emissions from CEDS in Europe are 60% higher in DJF than in JJA).

The higher levels of CO near the surface in winter are also due to the weak convection and mixing in this season, which allows pollutants to accumulate in the boundary layer (Cohen et al., 2018), and its longer chemical lifetime due to the lower

photochemistry during this season (Novelli et al., 1998). As expected, anthropogenic contributions have a strong local influence in the LT (Fig. 3.c). For example, anthropogenic contributions are almost entirely from local sources in NW America, NE America and Europe in the LT.

It is in agreement with the fact that inter-continental transport impacts mostly the Free Troposphere because of the stronger prevailing winds there. Polluted airmass can also be transported for long distances at lower altitudes, or sink in the Boundary

layer (BL) after being transported at higher altitudes, but it generally requires a few additional days (Stohl et al., 2002) than the typical west to east intercontinental transport which generally needs no more than a few days in the middle troposphere of the Northern Hemisphere (Jaffe et al., 1999; Liang et al., 2007).

Most of the European pollution is exported via low altitude pathways, and can impact the concentration of CO into the LT of Eastern Asia North America and Northern Africa (Huntrieser and Schlager, 2004; Duncan et al., 2008; Li et al., 2002).

However those contributions in North America and East Asia are generally low compared with the mixing ratio of CO in the LT of those regions. Here, we are interested in the extreme values at the surface close to the major airports of the region (and therefore close to urbanised areas) so the low contributions from Europe are of minor importance but could have more impact in more remote parts of Asia.

In DJF, as there are almost no fires in the northern hemisphere mid-latitudes almost all of the CO anomalies are attributed to

anthropogenic emissions.

In JJA, even if they remain rare, some regions have a few of their anomalies attributed to Biomass Burning (BB) emissions, which are mostly from boreal regions. Even in Europe, where more than half of the BB contributions of the MIX and BB anomalies are from boreal North America.

At this altitude, the highest values of CO are found in Eastern Asia during both seasons. The anomalies can even reach a

mixing ratio over 700 ppb in DJF. Those extremely high values are due to the important emissions from local anthropogenic sources and especially from the industrial and residential sectors (Qu et al., 2022).

As outlined in the introduction, CO is an interesting tracer for surface influenced airmasses, but also because it is a precursor of $O_3$. It is therefore important to also analyse the $O_3$ mixing ratio within the detected CO plume. This is shown in Fig.4. The figure shows the seasonal distribution of $O_3$ measured in the 19 years of data to the values of $O_3$ measured in the different

types of CO anomalies.

In the LT in DJF our results are similar regardless of the region. We observe values of $O_3$ inside the CO anomalies close to the minima of the seasonal $O_3$ cycle. We can see that, in addition to the low photochemical activity linked to the boreal winter, we are seeing a cycle of $O_3$ destruction in the CO-rich fresh airmasses. These low values of $O_3$ in polluted urban airmass are often characteristic with NO titration (e.g. Yang et al. (2019)). In JJA, the mean $O_3$ mixing ratios in the CO anomalies are

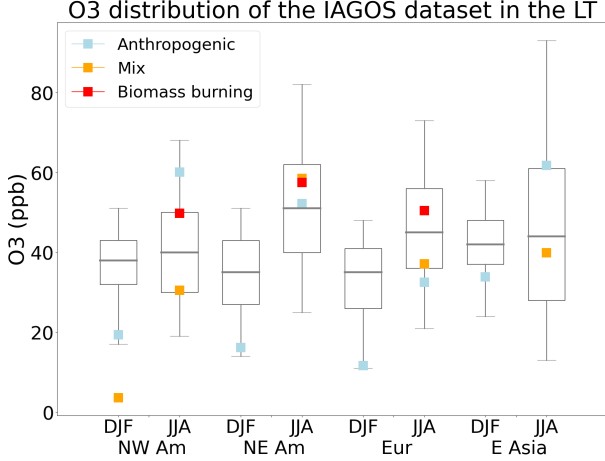

**Figure 4.** $O_3$ distribution measured by IAGOS in the lower troposphere (LT) (from the surface to 2000m). Coloured points represent $O_3$ mixing ratios inside the detected CO anomalies (each colour represents a type of CO anomaly attributed to a different source with SOFT-IO). The box plot represents the 5th, 25th, 50th, 75th and 95th percentiles of the $O_3$ distribution of the complete database (of these regions, seasons and vertical layer) during the studied period with the simultaneous CO records.

closer to the median. However, there are strong regional variations showing the important local influence at this altitude. East Asia is a region with important $O_3$ values and a region having frequent high $O_3$ episodes (Chang et al., 2017b; Lu et al., 2018). In this region anthropogenic CO anomalies are also associated with important $O_3$ values (20 ppb above the median).

### 3.1.2 Middle Troposphere (MT)

At higher altitudes, the measured CO is less influenced by the local conditions and emissions. This altitude layer is more impacted by long-range transport as the strong westerly winds present in the free troposphere (middle and upper troposphere) allow a rapid transport of the polluted airmasses across the northern hemisphere mid-latitudes. (Jaffe et al., 1999; Stohl et al., 2002; Liang et al., 2007).

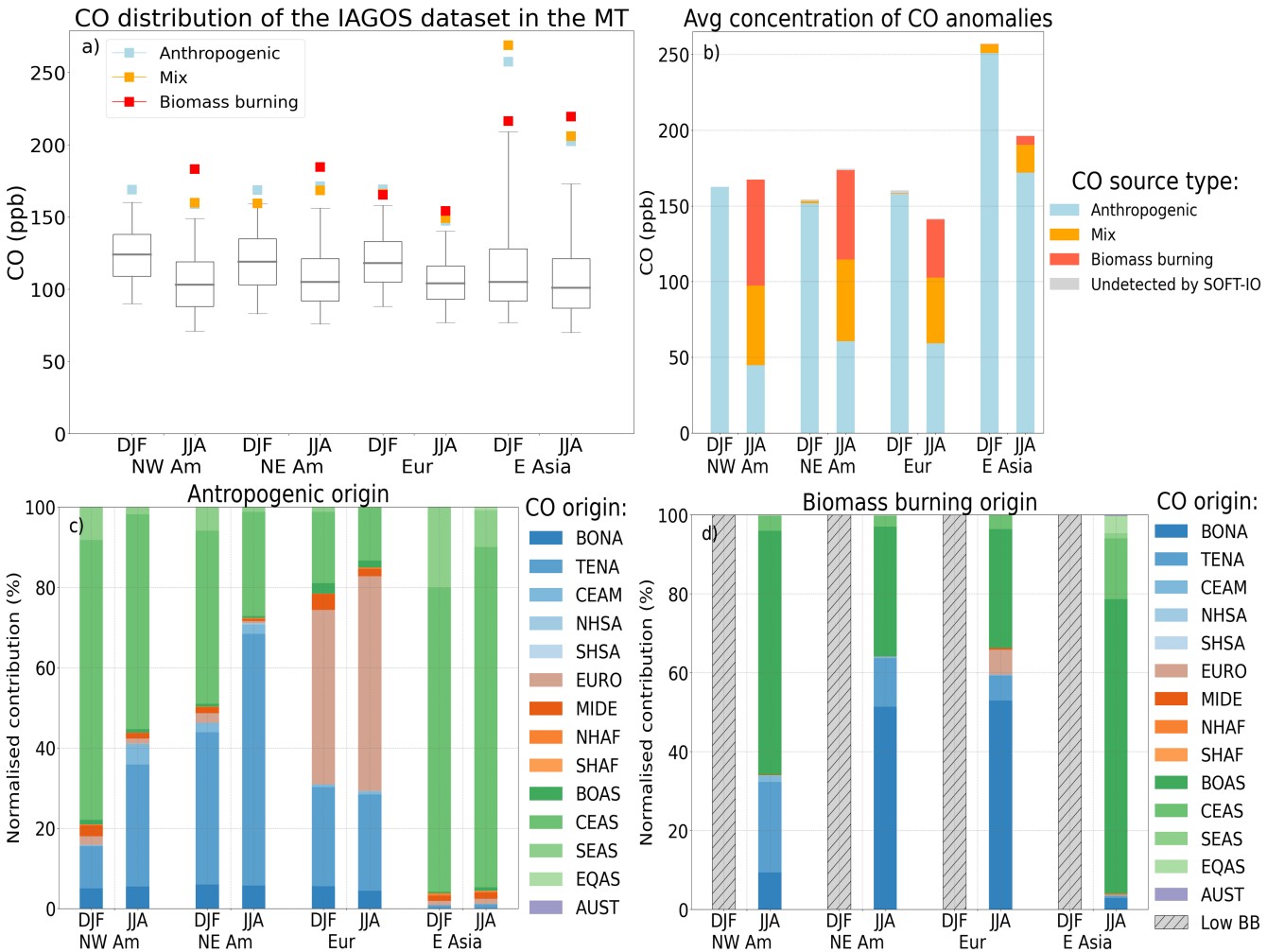

**Figure 5.** Same as Fig.3 but for the MT (between 2000m and 8000m). At this altitude 24 anomalies over the 5341 observed, are undetected by SOFT-IO, representing thus 0.4% (in grey on panel b).

In the MT (Fig.5), the CO distribution presents a maximum in DJF in the CO distribution. In this layer of the atmosphere, the local influence in the anthropogenic contributions (Fig.5.c) is still strong. Well known efficient processes for long-range transport of pollution are the Warm conveyor Belt (WCB) and frontal systems (e.g. (Cooper et al., 2004; Ding et al., 2009)) which can transport polluted surface airmasses to higher altitudes where important winds (e.g. jet stream at mid-latitude) can rapidly transport those airmasses to another continent. So, in general, there is important export of the pollutant from the regions at the western part of an ocean (start of the WCB) and the continent in the eastern part of the ocean will be the receptor (Europe and Western America) (Stohl et al., 2002; Huntrieser and Schlager, 2004; Cooper and Parrish, 2004). This feature is well captured by SOFT-IO where we can see that an important part of the contribution in NW America is coming from Eastern Asia. It is also true for Europe where more than half of the contributions are coming from either North America or Asia. We

can also see the lower contribution from long range transport in summer when the WCB is weaker (Cooper and Parrish, 2004). East Asia is mostly impacted by its own pollution during the two seasons. The upwind continent is Europe which is not known for having efficient vertical transport processes and so being prone to important export of its pollution (Stohl, 2001). East Asia
on the contrary is one of the regions with the most efficient vertical transport (Stohl et al., 2002).

In JJA, BB contributions come mostly from boreal America and Asia. Most of the time, the airports sampled by IAGOS are further south than most of the intense boreal fires. So, it is not surprising that little influence of the BB is detected in the LT. However, the influence from BB grows with altitude. In the MT, we observe an increased number of episodes attributed to either BB emissions or MIX sources in the MT of America and Europe in JJA (Fig.5.b). Fig.5.a shows that the plumes
attributed to BB emissions are the most intense in JJA.

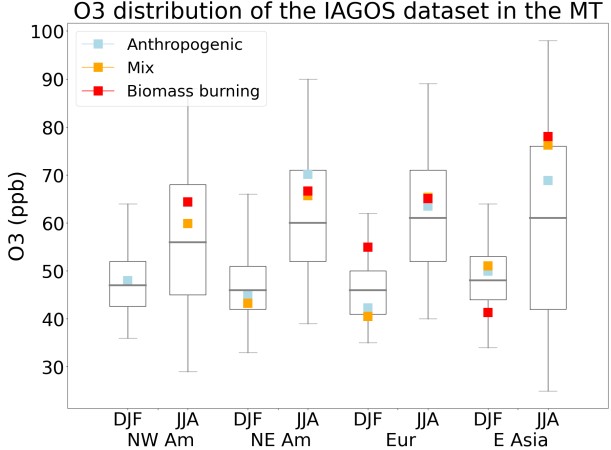

**Figure 6.** Same as Fig.4 for the MT

Fig.6 shows the mixing ratio of $O_3$ associated with high values of CO. In the MT there is almost no signal during the winter months (mixing ratio of $O_3$ inside CO anomalies is close or below the median) because of the relatively weak photochemical activity. In JJA, the $O_3$ mixing ratio within the CO anomalies is between the median and the 75th percentile of the total $O_3$ distribution, so the mixing ratio of $O_3$ in the CO plume are on average 5 to 10 ppb higher than the median values depending
on the region. In East Asia, BB (and MIX) plumes are rare and mostly come from boreal North Asia. $O_3$ values within those plumes are 20 ppb higher than the median and 10 ppb higher than the plumes from anthropogenic emissions.

### 3.1.3   Upper troposphere

To reach the UT, a polluted airmass needs to meet with an intense vertical transport episode, and not every WCB or deep convection episode brings airmasses from the surface to the UT. When in the UT, those airmasses can rapidly cross the entire
hemisphere.

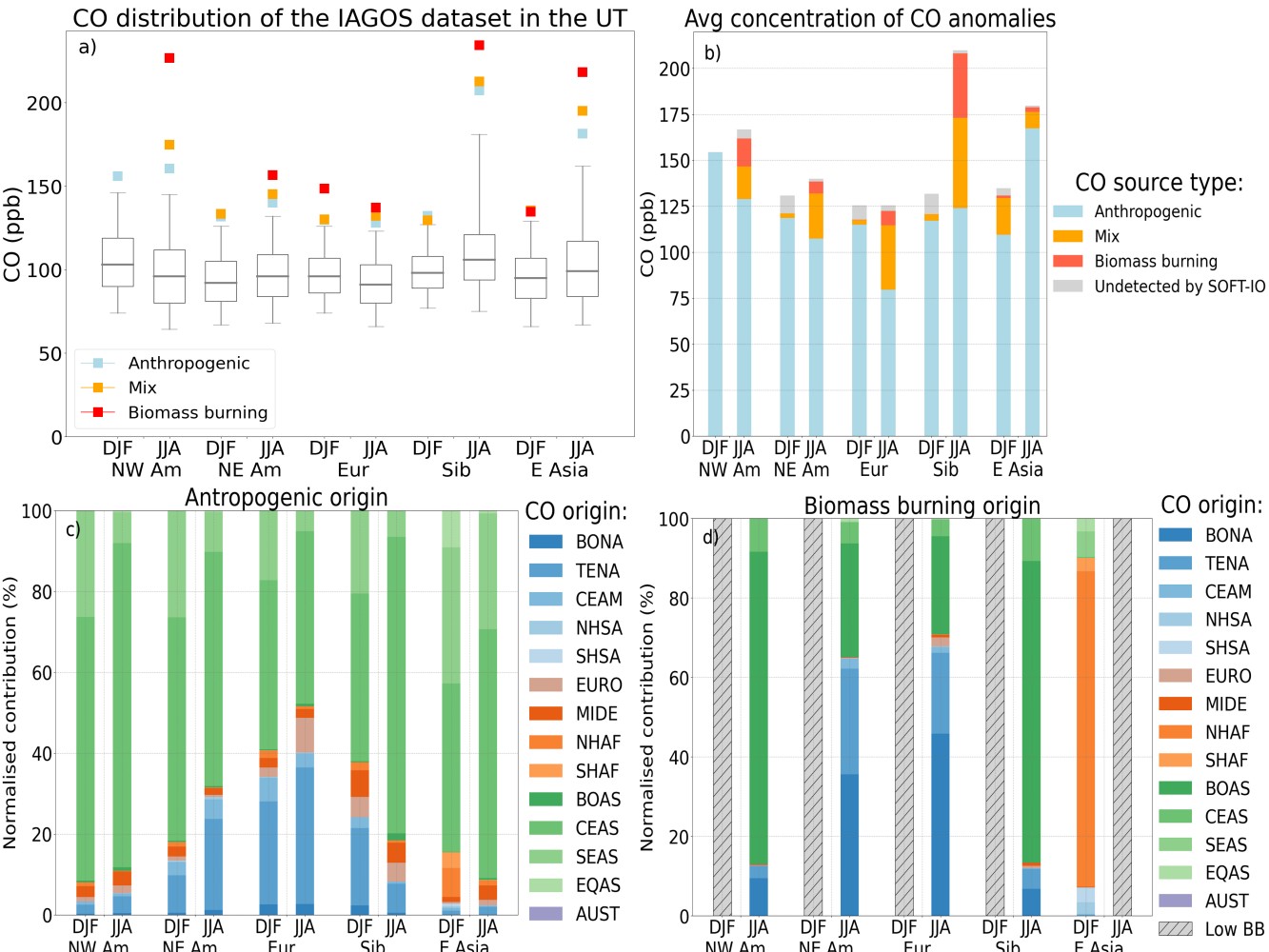

**Figure 7.** Same as Fig.3 but for the UT (between 8000m and the dynamical tropopause (2PVU)). At this altitude 223 anomalies over the 7865 observed are undetected by SOFT-IO, representing thus 2.8% percent (in grey on panel b).

Fig.7 is the same as Fig.3 for the upper tropospheric layer. Some regions like Europe do not show important variations of their 95th percentile between JJA and DJF while other regions like Siberia and East Asia present a drastic increase during JJA.

We can see in Fig.7.b that in DJF, the majority of the plumes are explained by anthropogenic emissions. In JJA, the number of anomalies attributed to BB increases with the onset of the northern hemisphere fire season. However, a higher number of anomalies are still explained by anthropogenic emissions, which is different from what we observe in the MT of America and Europe. This is because the most intense pyroconvection episodes from boreal fires rarely reach the UT directly (Labonne et al., 2007). Thus, regardless of the emission intensity, vertical transport is required for a CO plume to reach the UT. Anthropogenic emissions continuously inject CO into the boundary layer. Consequently, episodes of significant vertical transport of airmasses from the surface to the UT may cause a drastic increase in the upper tropospheric CO mixing ratios, even if local emissions

are not higher than usual. However, due to the intensity of BB emissions, when these plumes reach the UT they often are the most intense CO anomalies (Fig. 7.a). The most intense anomalies are detected in north-western America, Siberia (and East Asia in small proportion) and they are attributed to emissions of biomass burning from boreal Asia. In the UT, those anomalies even if not the most frequent are extremely intense. Siberia and East Asia both present two of the most important amplitudes of the CO seasonal cycle. The large increase of CO in JJA in Siberia with respect to DJF can be partly attributed to the local fires. However, approximately 60% of the episodes are still related to anthropogenic emissions and around 25% are due to MIX sources. The mean mixing ratio of these episodes during the summer months also increased significantly compared with their winter values. Furthermore, East Asia shows a similar summertime increase in the mixing ratio of its extremes without a significant number of BB plumes.

In DJF in Siberia, the anthropogenic contributions are small and there is no clear signal. In JJA however, there is a 50% increase in the anthropogenic contribution, of which 70% comes from CEAS. The low mixing ratio and contribution in winter are partly explained by the presence of the Siberian high, which prevents the export of polluted surface airmasses from the eastern part of Asia (Pochanart et al., 2004).

However, the wind direction changes with the onset of the east asian summer monsoon. In JJA, there are strong southeasterly ascending winds that transport pollution and moisture into the upper troposphere of East Asia and these airmasses can even reach the northern part of Siberia, which can explain the important contribution of CO from CEAS during this season in Siberia. It can explain the very low number of episodes associated with fire emissions (MIX and BB) in East Asia as heavy rainfall prevents fires in this region and the prevailing winds from the Pacific ocean are less likely to bring airmasses polluted by Siberian fires (Pochanart et al., 2004).

In the other regions (North America and Europe), the most intense anomalies remain those attributed to BB emissions and they represent around 5 to 10% of the number of anomalies. As we said previously, the BB anomalies in NW America are attributed to emissions from boreal Asian fires. In NE America and Europe those anomalies are less intense and they are attributed to fires from boreal America, boreal Asia and Temperate North America. Most of the BB contributions are from the two boreal regions (boreal America and boreal Asia), which is probably due to the higher emissions height of those fires increasing the probability of the emitted CO reaching the UT (Dentener et al., 2006).

In the two regions of North America, the main anthropogenic contributions to CO anomalies come from CEAS, but it can be seen that the influence of American emissions is greater in the eastern part, while the contributions in the western part originate almost entirely from CEAS. Europe's anthropogenic contributions come from Asia and North America. Only a small fraction is emitted locally, which is not surprising given the relatively weak convective activity in the region (Stohl et al., 2002).

The most polluted airmasses in the UT are often rapidly transported upward after their emission (Huang et al., 2012). Among the emitting regions, Eastern Asia is one of the more prone to vertical uplift of its pollutants because of the important convective activity of the regions (WCB, east asian monsoon...) (Stohl et al., 2002) and the presence of the Tibetan plateau, which can play an important role by lofting polluted airmass into the upper part of the troposphere (Bergman et al., 2013; Pan et al., 2016). Once in the UT, those airmasses can be transported around the hemisphere, which can be seen by the anthropogenic contribution from SOFT-IO where CEAS alone accounts for at least 40% of the anthropogenic contribution in the different

regions and even reaches 79% in NW Am. The total emissions of CEAS during this period account for about half of the northern hemisphere emissions.There is important vertical uplift also in North America because of frequent deep convection episodes and the mid-latitude cyclone starting in the regions (Cooper and Parrish, 2004). The European region in contrast, is identified to have few vertical uplift pathways (Huntrieser and Schlager, 2004). It means that "high level" of emissions are not the only parameter to take into account, but there is also the fact that the east asian atmosphere is characterised by strong

convective activity (e.g Stohl et al. (2002)), which allows the polluted air to be quickly transported in the MT or UT where it can be distributed all over the Northern hemisphere.

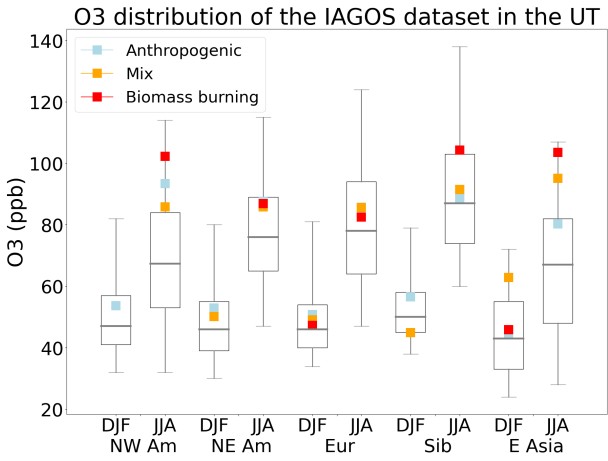

**Figure 8.** Same as Fig.4 for the UT.

Mixing ratios of $O_3$ within CO anomalies are shown in Fig. 8. As explained in sect. 2, the UT is defined as being below the 2 PVU surface. It may therefore include some stratospheric air or at least part of the mixing layer. The $O_3$ values measured within CO anomalies shown in Fig. 8 are mostly typical of tropospheric values, but it is possible that a small fraction of them

are contaminated by some stratospheric airmasses. Obviously, $O_3$ presents a stronger seasonal cycle than CO (Fig. 8). In DJF, the $O_3$ mixing ratio is at its minimum and values within the anomalies of CO are slightly lower than the 75th percentile in most regions. However, during the summer months the regional variations are more important, some regions show values of $O_3$ between the 75th and 95th percentile inside the CO anomalies whereas in Europe for example the $O_3$ values are just above median level.

Previous studies already noticed the $O_3$ maximum over Siberia (Gaudel et al., 2018). (Cohen et al., 2018) suggested that this maximum could be due to a higher stratospheric influence over the region. In the anthropogenic CO anomalies, the $O_3$ values are close to the background. However, as demonstrated in Fig.7 a significant portion of polluted airmasses are transported from the surface of East Asia to the UT of Siberia via the East Asian summer monsoon, which could potentially influence the production of $O_3$.

On average for the other regions, $O_3$ mixing ratios in CO anomalies are 13 ppb ppb higher than their respective median and this difference can reach 21 ppb for the CO anomalies associated with Biomass Burning emissions. The CO anomalies with

the highest values of $O_3$ are the anomalies associated with BB emissions from boreal Asia and detected in NW America, East Asia and Siberia. One reason for these high ozone levels is that those anomalies all come from Siberia, a region with very high ozone levels. Moreover, we saw that those fires were responsible for particularly intense CO anomalies and so probably emitted not only CO but also many reactive compounds such as VOCs and NOx which are other precursors of $O_3$. As for episodes associated with anthropogenic emissions, their associated mixing ratios of $O_3$ are often above median levels but a lot of variation can be observed depending on the region. In NW America, in JJA levels of $O_3$ inside the anthropogenic CO anomalies are high (93ppb so 10ppb above its 75th percentile), those anomalies are associated with emission from CEAS, so the airmass rich in pollutants had the time to produce an important quantity of $O_3$ before reaching the American continent. Production or elevated values of $O_3$ during the transport of polluted plumes from East Asia have already been observed during the Intercontinental Transport and Chemical Transformation 2002 campaign (ITCT 2K2) (Nowak et al., 2004; Hudman et al., 2004), so similar processes could be at play here.

## 3.2 India

The seasonal cycle of CO over India is characterised by a minimum in JJA in the LT and MT, and a maximum in SON-DJF in the LT superposed by a maximum in MAM in the MT (Fig.D1 and D2 in the appendix). The Asian monsoon has a strong influence in this part of the world on the redistribution of the pollutants emitted at the surface (Lawrence, 2004b). Interesting and specific features appear in all four seasons in the UT as highlighted in Fig.9. DJF and MAM have a similar signal in the UT as the same sources are at the origin of most of the CO anomalies. We can see on Fig.9.b that half of the CO anomalies are linked to BB emissions (pure BB and MIX sources) and half are pure anthropogenic anomalies. From December until late March, it is the fire season in the Northern Hemisphere of Africa, and we can see on Fig.9.d that those emissions can reach the UT of India. It is also the period of the winter monsoon in Southern Asia, this season is characterised by week convective activity and Northern prevailing wind transporting pollution at low altitude toward the Indian ocean (Lelieveld et al., 2001; Lawrence and Lelieveld, 2010) and explaining the rather high values of CO in the LT and MT during this period (Fig.D1 and D2 in the appendix) and the low contribution from SEAS in the UT, at this altitude the anthropogenic CO anomalies receive an influence from CEAS and SEAS but also from NHAF. In JJA, it is the wet phase of the monsoon in India so the important convective activity and precipitation associated with this period (Kar et al., 2004) leads to rapid transport of the South-Asian emission to the UT while preventing BB: almost all the CO anomalies are caused by anthropogenic emissions from India or the close proximity (SEAS and CEAS). In SON, the CO anomalies are at their maximum and are caused by anthropogenic emissions from SEAS and CEAS but also by BB emissions from EQAS. The BB anomalies are clearly the most intense during this season. It is interesting to note that in the vast majority the BB anomalies recorded by IAGOS during SON were from 2015. This year was hit by an important El Niño phenomenon characterised by especially intense fires over the Equatorial part of Asia (Field et al., 2016). According to Kar et al. (2004), during this season in 2002 there was also important transport of CO from tropical fires.

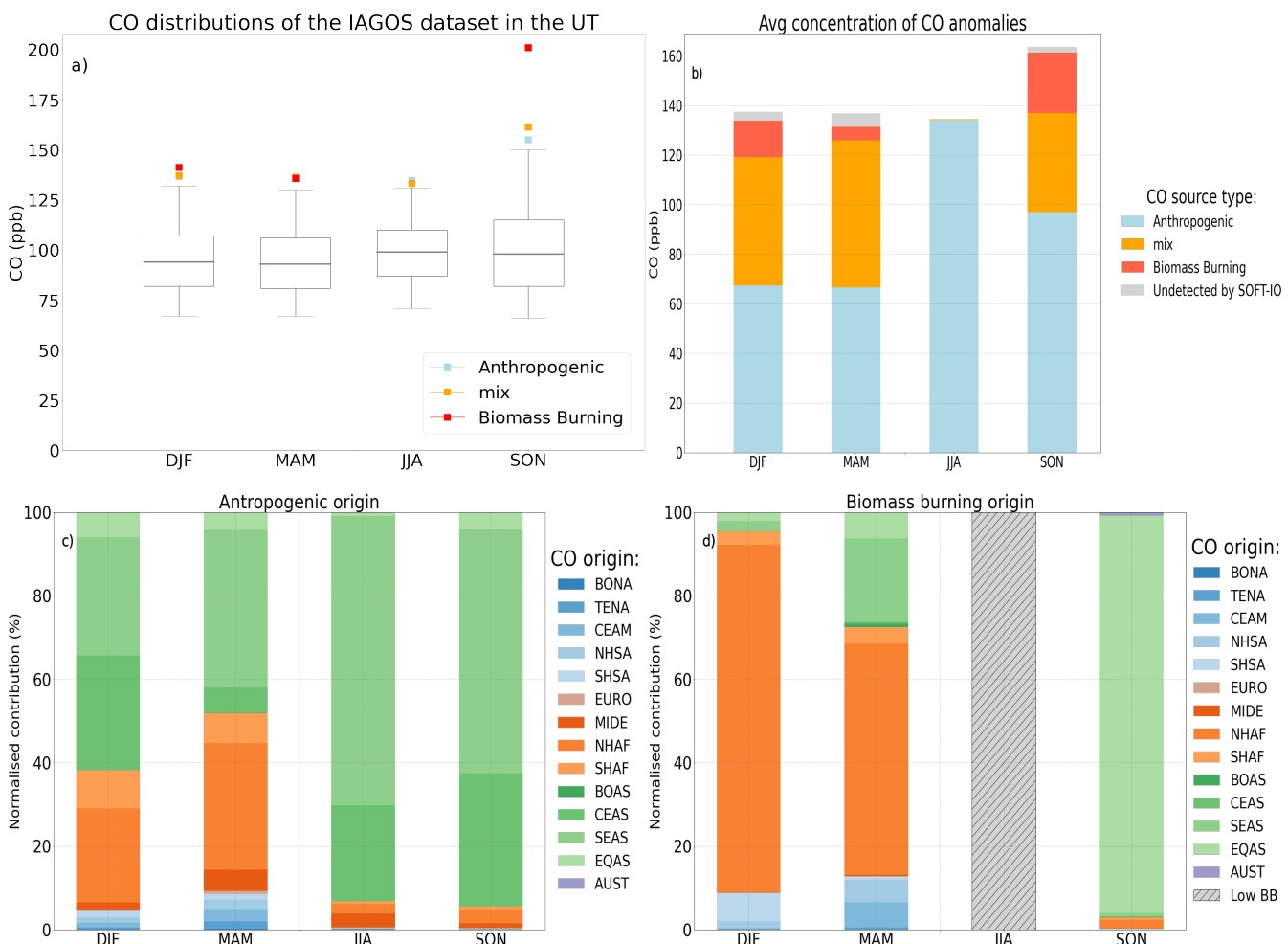

**Figure 9.** Same as Fig.3 but only for the Indian region during the four seasons for the UT (between 8000m and the dynamical tropopause). (In MAM in panel a) the other squares are superposed below the one from BB origins). At this altitude 37 anomalies over the 2228 observed are undetected by SOFT-IO, representing thus 1.7% (in grey on panel b).

The $O_3$ cycle shown here is similar to the cycle described in Lal et al. (2014) and obtained by a radiosonde, here the focus is on the O3 measured in the CO anomalies. In the LT (see Fig.D3 in the appendix), the minimum values of $O_3$ are reached during the summer monsoon in JJA. The low values can be explained by the increased marine influence during this period (Lawrence and Lelieveld, 2010). At this altitude the $O_3$ values recorded simultaneously as the CO anomalies are low and show the low $O_3$ production in those plumes.

In the MT (see Fig.D4 in the appendix) and UT (see Fig.10), the maximum of the $O_3$ is reached during MAM, and the minimum is reached during DJF. In the UT, in DJF and MAM an important part of the CO anomalies come from northern African BB. Those plumes are associated with higher values of $O_3$ (11 and 10 ppb above the median respectively for DJF

and MAM). CO anomalies in JJA are caused by the local emission of anthropogenic CO rapidly transported to the UT by the important convective activity of the South Asian summer monsoon. This rapid transport could explain that the associated values of $O_3$ are close to the median (65 ppb). In the post monsoon season (SON) BB anomalies from Equatorial Asia are added to the local anthropogenic anomalies. The values of $O_3$ in the BB plumes are low and close to the 25th percentile (44 ppb) which is explained by the lower background values of $O_3$ in Equatorial Asia compared to India (Cohen et al., 2018).

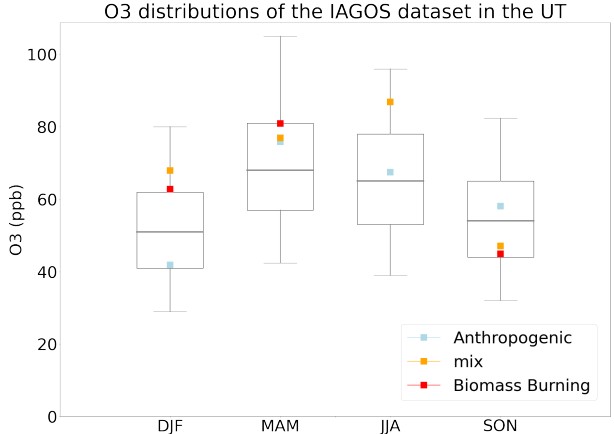

**Figure 10.** Same as Fig.4 for the UT in the Indian region during the four seasons.

## 3.3 Africa and Middle East

### 3.3.1 Lower and Middle Troposphere

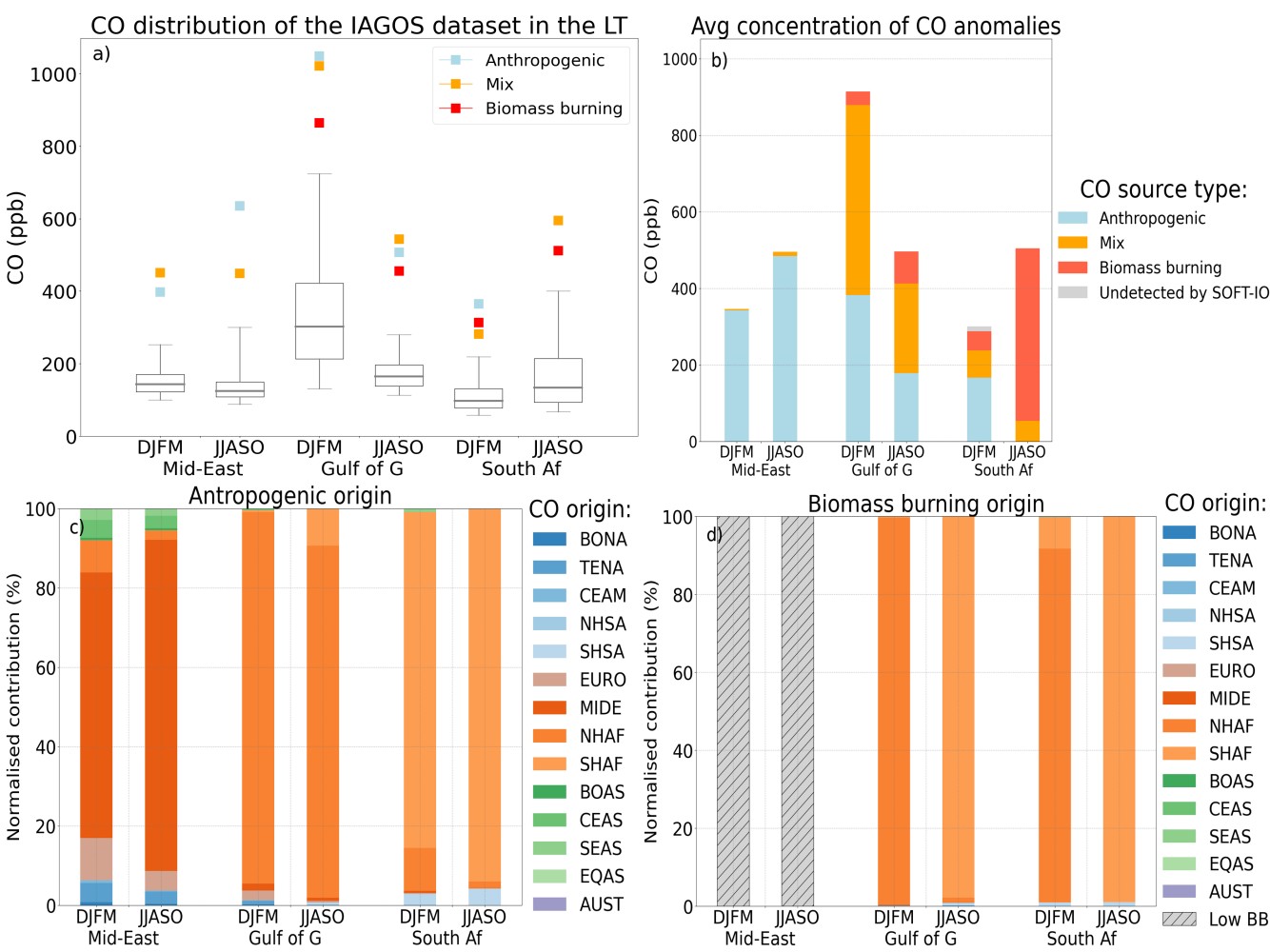

**Figure 11.** Same as Fig.3 but for the LT (below 2 000m) in Africa and Middle East. At this altitude 3 anomalies over the 1449 observed are undetected by SOFT-IO, representing thus 0.2% (in grey on panel b).

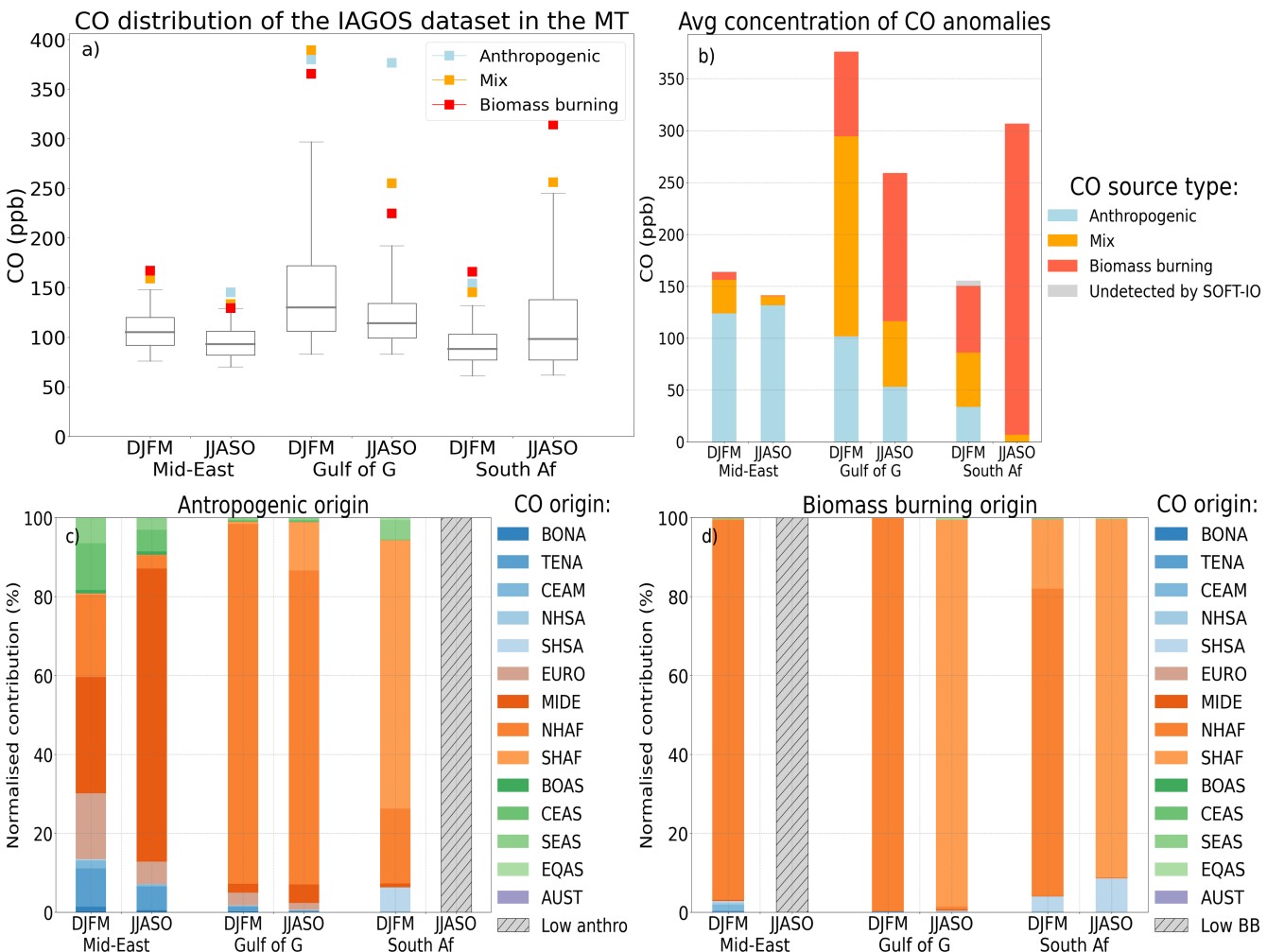

**Figure 12.** Same as Fig.3 but for the MT (between 2 000m and 8 000m) in Africa and Middle East. At this altitude 8 anomalies over the 1528 observed are undetected by SOFT-IO, representing thus 0.5% (in grey on panel b).

This section is focused on the CO anomalies detected over Africa and Middle East. As the result in the LT and in the MT present similar characteristics they are treated simultaneously.

Fig.11.a and Fig.12.a show the CO distribution in the two regions of Africa (Gulf of Guinea and Southern Africa) and the Middle East in the LT and the MT. Both layers present a maximum in DJFM in the Gulf of Guinea with a 95th percentile above 724 ppb in the LT and 297ppb in the MT. DJFM is the dry season in the Northern part of Africa, which causes high levels of CO from biomass burning emissions (see Fig. 11.c). In the Gulf of Guinea, the maximum values of CO are reached during DJFM, which come from the important Biomass burning episode of the region during this season. There is also a large population which explains the important anthropogenic contribution. The accumulation of the pollution observed in the LT during this season has already been characterised in Sauvage et al. (2005) and is caused by the Harmattan winds bringing rich

CO airmasses caused by the upwind fires to the southwest of the Gulf of Guinea where there are most of the airports visited by IAGOS aircrafts (see Fig.B3 in the appendix).

In JJASO, the southwesterly trade winds bring airmasses from the Atlantic ocean. These airmasses are cleaner with respect to anthropogenic pollution but can bring BB plumes from Southern Africa.

The proportion of BB sources increased in the MT. The contribution of anthropogenic emissions maximise near the surface, especially over the Gulf of Guinea, one of the largest populated and polluted urban areas of the continent. In the mid-troposphere (MT), the intensity of the CO anomalies attributed to anthropogenic sources decreases in favour of those from BB and MIX sources.

The changes in origins of the BB contributions in DJFM and JJASO follows the shift of the Biomass Burning season from the northern hemisphere to the southern hemisphere.

In JJASO, during the dry season in southern Africa, the anomalies are the most intense there. The MT 95th percentile is just below 250 ppb in JJASO and most of the detected anomalies of Southern Africa are attributed to emissions from southern hemisphere fires.

The Middle East plumes have a high contribution from anthropogenic emissions in both seasons in the LT and the MT. The Middle East has been identified in previous studies as receiving the pollution of multiple regions (Li et al., 2001; Stohl et al., 2002; Duncan et al., 2008). Europe is mostly exporting its pollution via low altitude pathways and we can see on Fig 11.c and Fig 12.c that up to 20 % of the anthropogenic contributions can come from Europe. There are also contributions from Temperate North America and South and East Asia, but contrary to the European contributions these probably followed higher altitude pathways before sinking to the MT or LT (Li et al., 2001; Stohl et al., 2002). We can also see important differences in the provenance of the anthropogenic contributions between DJFM and JJASO.

In JJASO, we are seeing contributions mostly from the local regions (MIDE) similarly to the contributions in the LT. According to previous studies the Planetary Boundary Layer in this region can reach 4000 or 5000 meters in JJA (Gamo, 1996; Ntoumos et al., 2023). So, this differences in the origins of the contributions between DJFM and JJASO may be caused by the higher PBL height in JJASO.

Fig.13 and Fig.14 show the $O_3$ distribution measured by IAGOS as well as its mixing ratio inside the detected CO anomalies. In the following paragraph, if not mentioned otherwise, the $O_3$ mixing ratio refers to the mixing ratio inside the CO anomalies. The box plot Fig.13 shows that the lower part of the troposphere presents important variability between regions and seasons.

In the Middle East, $O_3$ values are among the highest in JJASO in the LT and MT. The summertime median is also higher than the median from East Asia (see Fig.4 and Fig.6) which is a region with identified extreme $O_3$ values (Chang et al., 2017a; Lu et al., 2018). Li et al. (2001) suggested that the important tropospheric $O_3$ in Middle East were due to the constant import of pollution from different regions trapped in the upper level anticyclone and the strong subsidence associated to it cause an accumulation in the region. Here the CO anomalies detected are mostly caused by emissions from the Middle east rather than from long range transport. In the Middle East LT, values of $O_3$ inside CO anomalies attributed to anthropogenic emissions are lower than the 25th percentile, which is similar to the observation made on the northern hemisphere mid-latitudes. In the MT, the anthropogenic anomalies are close to the median during both seasons.

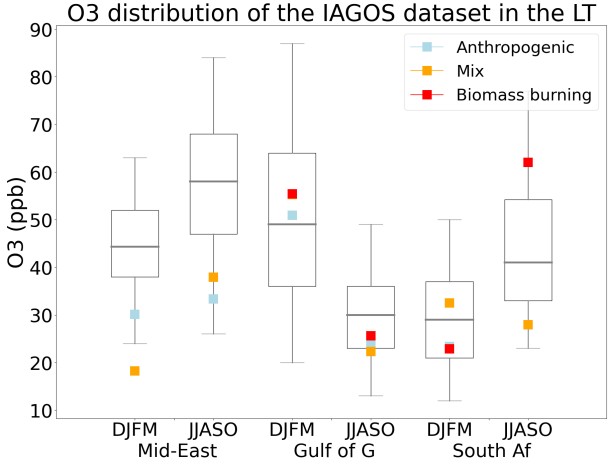

**Figure 13.** Same as Fig.4 for the LT in Africa and Middle East

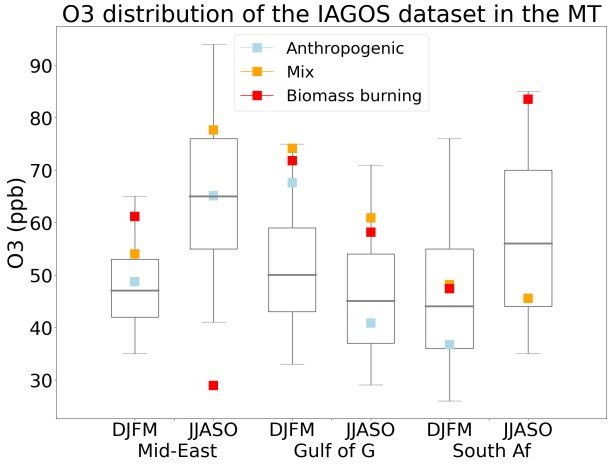

**Figure 14.** Same as Fig.4 for the MT in Africa and Middle East

In DJFM, in the Gulf of Guinea, values of $O_3$ associated with the CO anomalies are just above the median in the LT, whereas they are almost as high as the 95th percentile in the MT. In JJASO, values of $O_3$ in the Gulf of Guinea exceed the median only during the MIX and BB anomalies in the MT. South Africa presents low values of $O_3$ in DJFM but much higher values during the BB season of the southern hemisphere. As in the Northern Hemisphere, the mixing ratio of $O_3$ inside plumes of CO influenced by BB is higher than the median. The important mixing ratio of $O_3$ in the BB anomalies were already discussed previously for the NH mid latitude CO anomalies and can be caused by the important quantity of reactive gases acting as $O_3$ precursors emitted by biomass burning (e.g. Galanter et al. (2000); Mauzerall et al. (1998)).

 ### 3.3.2 Upper troposphere

As the seasonality of CO in the African upper troposphere has already been described in Lannuque et al. (2021), this section will emphasise the differences between the seasonal cycle of CO presented in Lannuque et al. (2021) and the extreme values of CO presented here.

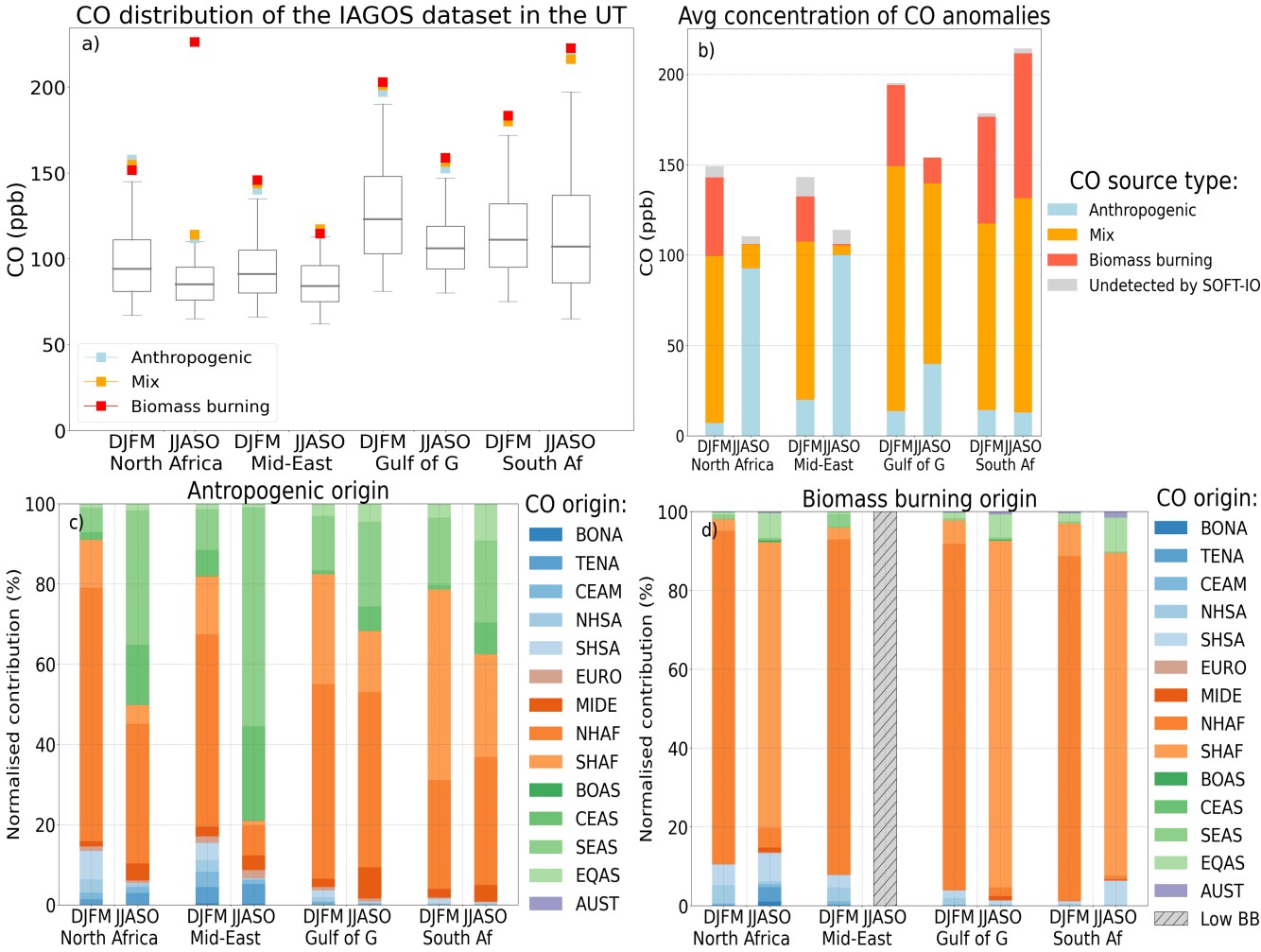

**Figure 15.** Same as Fig.3 but for the UT (between 8000m and the dynamical tropopause) in Africa and Middle East. At this altitude 268 anomalies over the 5859 observed are undetected by SOFT-IO, representing thus 4.6% (in grey on panel b).

Similar to the study of the CO seasonal cycle from Lannuque et al. (2021), the anomalies are the most intense in the hemisphere of the strongest Hadley cell. In DJFM, the dry season is in the African Northern hemisphere causing important fires emitting a lot of CO, whereas in JJASO, the dry season is in the southern hemisphere and the most intense CO anomalies are detected in South Africa.

The Middle East and the North of Africa, similarly to the Gulf of Guinea show an important seasonal variation with a maximum in DJFM. Fig.15.b shows that the DJFM maximum reached in the northern hemisphere regions are mostly caused by important biomass burning plumes from NHAF (Fig. 15.d). In JJASO, it is the wet season in NHAF, so BB emissions are drastically reduced in the region. Furthermore, as Lannuque et al. (2021) showed, anthropogenic CO is transported from SEAS (Fig. 15.c). There are significant anthropogenic emissions in the Indian subcontinent, and the active convection brought by the Asian Summer Monsoon (ASM) allows the emitted CO to be rapidly transported from the surface to the UT. There, it is trapped in the Asian Monsoon Anticyclone (AMA) (Park et al., 2008; Barret et al., 2008; Tsivlidou et al., 2022).

Anomalies in the Gulf of Guinea and in South Africa in the UT are heavily influenced by BB emissions, only a small fraction of the plumes in these two regions are solely caused by anthropogenic emissions. The others are caused by either pure BB emissions or MIX sources. The BB contribution comes from either NHAF or SHAF depending on the season. In DJFM, in Southern Africa it is interesting to note that the CO mixing ratio are higher in the UT than in the MT (see Figs. 12 and 15 and table 1). It shows the importance of the Hadley cell circulation for the distribution of the pollutant in the UT.

The signal of the climatologies studied in Lannuque et al. (2021) and of the extreme studied here, is similar. The main differences are the increased BB proportion from NHAF in DJFM in the four regions. The small contribution from North America observed by Lannuque et al. (2021) is barely visible here as the airmasses transported from there are probably too diluted (i.e. close to the median) to contribute to any important anomalies of CO.

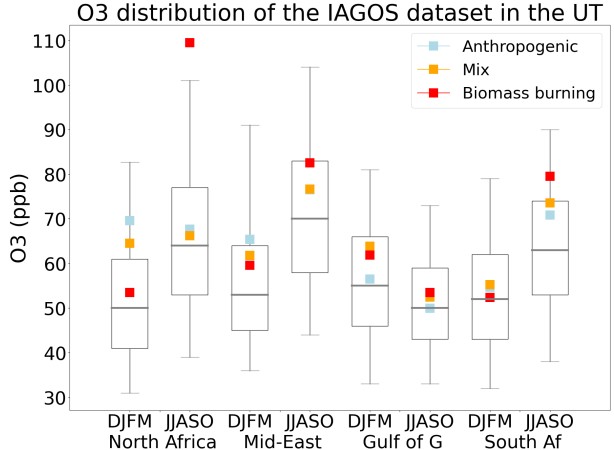

**Figure 16.** Same as Fig.4 for the UT in Africa and Middle East

$O_3$ mixing ratio associated with the observed CO anomalies is shown in Fig.16. The upper troposphere signal is not as clear as the one from the northern hemisphere. However, like in the MT in middle East and northern Africa, we can see that the $O_3$ mixing ratio in the BB anomalies are higher than the median of $O_3$ and can even reach the 75th percentile of $O_3$ in DJFM. Middle East shows the highest values of $O_3$ during JJASO. At this altitude layer CO anomalies are mostly from anthropogenic emissions originating from SEAS. Those anomalies show a 7 ppb enhancement compared with the median of 70 ppb. This is in agreement with a previous study from Li et al. (2001) showing elevated $O_3$ values in Middle East due to

important import of anthropogenic pollution from polluted regions and very little from stratospheric intrusion. Middle East meteorological conditions are favourable for $O_3$ production (Duncan et al., 2008) as well as a constant important of pollutant from Asian emissions (Stohl et al., 2002) and an influx of NOx produced by lightning during the Asian monsoon Li et al. (2001).

In the Gulf of Guinea, the overall distribution of $O_3$ measured by IAGOS is lower than in the rest of Africa. Its maximum values of $O_3$ are observed in DJFM during the fire seasons of North Africa. The $O_3$ values observed in the CO anomalies are a bit lower than the 75th percentile in DJFM and close to the median in JJASO. In Southern Africa, in DFJM, during the raining season, no clear signal is observed. In JJASO during the fire seasons $O_3$ is at its maximum and high values are observed in the CO anomalies.

## 4   Conclusions

This study is based on the in-situ IAGOS CO anomalies defined as the observations above the 95th percentile of each individual altitude range (LT, MT, or UT), region and season. IAGOS is a research infrastructure which uses commercial aircraft to measure atmospheric composition. In total, over the 18 years of measurements, more than 43,000 equipped IAGOS flights were made. In addition, SOFT-IO allows us to give a diagnosis on the main type of source as well as the region of emission responsible for the detected CO anomalies. SOFT-IO is based on FLEXPART retro-plumes initiated at each IAGOS measurement point. The back trajectory ensembles are then coupled to two CO emissions inventories : GFAS for the biomass burning and CEDS for anthropogenic emissions. The conclusions below relate only to CO anomalies (consecutive values of CO above the 95th percentile of the region/season/altitude).

In the northern mid-latitudes, anthropogenic emissions peak in winter, and biomass burning emissions peak in summer. The anomalies in the LT are very sensitive to local emissions which are highest during these winter months, and because of the weak convection and low photochemical activity in the northern hemisphere, these emissions accumulate until spring. $O_3$ values in the CO anomalies of this season are 17 ppb lower than the median on average. In summer, at this altitude, there is important regional variations which probably highlight the local environment more or less prone to $O_3$ production/destruction.

In the middle troposphere, the high CO plumes over NW and NA America, Europe in JJA are mainly due to boreal fire emissions. Those fires originate either from boreal Asia or boreal America. High CO plumes from anthropogenic origins still account for a significant proportion of the anomalies, but unlike the lower troposphere, the origins of these emissions are split between a local and a long-range influence. $O_3$ in the MT in JJA shows higher values than normal in the CO anomalies (from 7 to 9 ppb higher than the median)

A large proportion of the plumes over NW America originate from emissions in East Asia. In Europe, the majority of CO anomalies attributed to BB result from emissions in either boreal America or boreal Asia, with only 10% being due to fires on the European continent. East Asia continues to be dominated by anthropogenic pollution throughout the year, due to several factors. This is due to the high levels of anthropogenic pollution emitted in the region, as well as the favorable conditions for these emissions to rise out of the lower troposphere due to the high level of convective activity. Furthermore, the summer

monsoon winds from the southeast prevent the transport of Asian boreal fire emissions towards East Asia. Northern America also has even if in lower proportion large emissions and important export pathways allowing to transport polluted plumes towards Europe. European emissions are not negligible either but the synoptical conditions of the regions reduce the possibility to export its pollution with vertical transport. A low altitude transport of its pollution is thought to contribute to the surface pollution of multiple regions but they might be too diluted to strongly influence the most intense plume of CO studied here.

In the UT, in northern mid-latitudes, anomalies caused by BB emissions remain less frequent than those from anthropogenic emissions, but they are consistently the most intense during the boreal fire season (JJA). During the summer months, the upper values of the CO distribution (75th percentile and higher) in Siberian and East Asian regions experienced a significant increase. Around one third of the plumes identified above Siberia are associated with fire emissions. The rest are anomalies due to anthropogenic emissions from East Asia. This transport of pollution to Northeast Siberia is partly due to the East Asian monsoon, which can bring airmasses from Southeast Asia.

The $O_3$ associated with anthropogenic CO plumes are regularly higher than normal (10 to 20 ppb higher than the median depending on the region). $O_3$ reaches its maximum over Siberia so, the exported CO plumes from this region will cause important anomalies of $O_3$ in the regions with lower $O_3$ environment like Eastern Asia and NW America. The $O_3$ measured within the BB anomalies in Siberia are 15 ppb higher than the median but no no elevated values are measured in the anthropogenic anomalies coming from Eastern Asia. Further investigations are needed to explain the extremely high values of $O_3$ measured in the UT of Siberia in JJA.

The Indian pollution pattern is dominated by the two phases of its monsoon and two transition seasons. In DJF, it is the winter phase of the monsoon season and dominant north-easterly surface wind keeps the pollution at the surface and causes important pollution events in the LT. The values of CO remain high during the two transition periods (MAM and SON) . In the MT, the maximum is reached during MAM with the more active vertical mixing. At those altitudes (LT and MT), there are only few BB pollution events and the anthropogenic contributions are from local emissions (SEAS). In the UT, in DJF and MAM while vertical transport is still low, a lot of CO is coming from Northern Africa from anthropogenic and BB sources. In JJA, and during the Indian monsoon, strong convective activity favours the export of local pollution to the UT and most plumes are therefore attributed to local anthropogenic pollution. Finally, during the SON months, the plumes are linked both to anthropogenic pollution from South and East Asia and to pollution linked to fire emissions from equatorial Asia. The El Niño episode in 2015 and the major fires that took place during the year in equatorial Asia had a major influence on our measurements, since most of the fire plumes seen during the SON months in India were detected during that year.

CO anomalies in the African troposphere follow a different regime. Fires are much more frequent and are responsible for a large proportion of the CO anomalies, even in the lower layers of the troposphere. In the DJFM, fires occur in the tropical northern side of the continent, so the Gulf of Guinea is strongly affected. In addition, this region is a major emitter of anthropogenic CO and the mixing ratios of the anomalies are very high during this season. In Southern Africa in DJFM, CO anomalies are more intense in the UT than in the MT, highlighting the importance of the upper branch circulation of the Hadley cell for the transport of pollutants in this region. In JJASO, it is mainly the southern part that is affected by fires. In

fact, fire emissions are responsible for almost all the CO anomalies detected in this region/season. The Middle East is relatively unaffected by fires in the LT and MT, and the anthropogenic contribution are essentially locally emitted in summer.

In the UT in North Africa and Middle East, CO is in the vast majority coming from Northern Africa, the important anthropogenic and BB emissions are transported upward at the ITCZ and transported northward by the upper branch of the Hadley cell. In JJASO, these two regions are spared by BB emissions from Southern Hemisphere Africa and the CO mixing ratio decreases importantly during this season. Important part of the CO there is coming from Southern Asia where it has been transported upward by the South Asian monsoon and trapped in the SAMA before being transported westwards. and its anthropogenic pollution is essentially local. However, the upper part of the Northern African and Middle Eastern troposphere can be strongly polluted by African fires during DJFM.

To summarise, the CO anomalies observed throughout the troposphere over Africa are deeply influenced by the intensity of the emissions (both anthropogenic and BB) and the active convective activity from the Tropics. In the LT, the anomalies are the most intense and are linked with local emissions. Higher up, the anomalies are caused by emissions from further away and are deeply influenced by the ITCZ shift and the variation of wet/dry season.

$O_3$ mixing ratios in the CO anomalies vary considerably from region to region. For example, in the lower troposphere over the Middle East, where the plumes are mainly due to local anthropogenic emissions, $O_3$ levels are below the 25th percentile. In the Gulf of Guinea and South Africa we observe $O_3$ levels above background (even above 75th percentile for South Africa) in the fire plumes during their respective dry seasons, and lower or near background levels during their wet seasons. The signal described above is further enhanced in the middle troposphere and $O_3$ values in the CO anomalies can reach the 95th percentile during the respective dry seasons of the Gulf of Guinea and South Africa. The fact that these high values are mainly observed in the mid-troposphere and not in the LT, where emissions occur, confirms that a minimum age of airmasses must be respected to allow time for photochemistry to take place. For these two regions, the signal in the UT is similar, although attenuated ($O_3$ levels no longer reach the 75th percentile in the Gulf of Guinea, but remain high in South Africa). In the lower and middle troposphere, the maximum $O_3$ values are found in Middle East. Previous studies assumed that the high $O_3$ in the regions were due to long range transport of polluted airmasses followed by chemical production in the regions (Li et al., 2001; Duncan et al., 2008). In the LT and MT most of the detected CO anomalies are from local anthropogenic emissions which either show low values of $O_3$ or values close to the median. In the UT, in JJASO CO anomalies are mostly from anthropogenic emissions from South East Asia. Those anomalies show enhanced values of $O_3$.

Our study is based on extreme CO mixing ratios, which we have defined as observations above the regional and seasonal 95th percentile. In order to ensure the robustness of the results with respect to this parameter, we performed a sensitivity test to check whether any major changes in the features could be observed with a threshold defined as the 75th or 99th percentile. Overall, the same characteristics were observed with just a few differences. In the Northern Hemisphere, increasing the threshold causes a slight increase in the proportion of fire-related plumes (diagnosed as BB or Mix), which is not surprising as we have seen that these plumes are the most intense most of the time. In addition, we observed that our anthropogenic plumes in the UT having an East Asian origin are proportionally more numerous at the highest percentile. In the UT, the African plumes appear to be even less sensitive to a change in the threshold.

Our study focused on CO anomalies measured between 2002 to 2019, but important trends in CO and $O_3$ in the atmosphere have been observed in several of the studied regions (e.g. Novelli et al. (1998); Kim et al. (2023); Gaudel et al. (2020)). So, we performed the same analysis with only the last 10 years of the IAGOS measurements. Several regions, showed a decreased 95th percentile in this datasets (see tables A2 and A3 in the appendix). However, the origins and sources of the anomalies remain similar in regions with sufficient number of data. The conclusion of the study remained largely unchanged for the CO anomalies of the last 10 years.

This study provides useful diagnostics to characterise the high levels of CO in the troposphere at northern mid-latitudes and over India, Africa and Middle-East. The characteristics of those plumes of CO are described for the seasons of (i) the most intense fire activity and (ii) the maximum of anthropogenic emissions with respect to the effect of those emissions on the CO distributions in the troposphere. Thanks to the simultaneously recorded $O_3$ mixing ratios, the diagnostics provided by this study include a first assessment of the $O_3$ levels within the extreme CO anomalies.

These $O_3$ values give information on its possible production in polluted plumes. However, without the measurements of additional chemical compounds (like VOCs and NOx for example) it is difficult to draw robust conclusions. To go further into the analysis on the $O_3$ in pollution plumes, information on more chemical compounds are required. The current perspective is to carry a similar study with a Chemistry Transport Model in order to get further information on the provenance of $O_3$ values but also on the amount of $O_3$ productions in polluted plumes, especially in regions with elevated values of $O_3$ like Siberia and the Middle East.

We have presented a detailed analysis of the characteristics of high carbon monoxide plumes and their associated ozone anomalies in different regions of the world. It is important for the IAGOS infrastructure to continue those measurements and to expand the regions sampled by the research infrastructure in order to provide these diagnostics in additional regions. This is particularly important in tropical regions, where anthropogenic emissions are increasing and impact on the $O_3$ trend globally (Zhang et al., 2016). Increased number and sampling frequency of measurements of NOx and aerosols by IAGOS will be available and valuable for future analysis focusing on $O_3$ photochemical production or air quality.

*Data availability.* The IAGOS data (IAGOS, 2022) are available at the IAGOS data portal (https://doi.org/10.25326/20) and more precisely, the time series data are found at https://doi.org/10.25326/06 (Boulanger et al., 2018).

| Acronym | Full name |
|---------|-----------|
| BONA | Boreal North America |
| TENA | TEmperate North AMerica |
| CEAM | CEntral AMerica |
| NHSA | North Hemisphere South America |
| SHSA | South Hemisphere South America |
| EURO | Europe |
| MIDE | MIDdle East |
| NHAF | Northern Hemisphere AFrica |
| SHAF | South Hemisphere AFrica |
| BOAS | BOreal ASia |
| CEAS | CEntral Asia |
| SEAS | South East Asia |
| EQAS | Equatorial Asia |
| AUST | AUSTralia |

**Table A1.** Table GFED acronym:

| | | LT | FT | UT |
|-------|-----|------|------|------|
| NW Am | DJF | 256 | 160 | 146 |
| | JJA | 251 | 149 | 145 |
| NE Am | DJF | 264 | 159 | 126 |
| | JJA | 241 | 156 | 132 |
| Eur | DJF | 332 | 158 | 126 |
| | JJA | 200 | 140 | 123 |
| Sib | DJF | no data | no data | 127 |
| | JJA | no data | no data | 181 |
| E Asia | DJF | 559 | 209 | 129 |
| | JJA | 441 | 173 | 162 |

| | | LT | FT | UT |
|-------|-----|------|------|------|
| NW Am | DJF | 224 | 155 | 142 |
| | JJA | 227 | 168 | 140 |
| NE Am | DJF | 230 | 148 | 112 |
| | JJA | 225 | 156 | 126 |
| Eur | DJF | 315 | 150 | 117 |
| | JJA | 187 | 135 | 118 |
| Sib | DJF | no data | no data | 119 |
| | JJA | no data | no data | 168 |
| E Asia | DJF | 550 | 205 | 128 |
| | JJA | 403 | 160 | 153 |

**Table A2.** q95 values (in ppb) used as thresholds for the different regions using data from 2002 to 2019 on the left and using data from 2010 to 2019 on the right

| | | LT | FT | UT |
|---|---|---|---|---|
| India | DJF | 424 | 157 | 132 |
| | MAM | 305 | 191 | 130 |
| | JJA | 267 | 134 | 131 |
| | SON | 470 | 150 | 150 |
| North Af | DJFM | no data | no data | 145 |
| | JJASO | no data | no data | 110 |
| Middle E | DJFM | 253 | 148 | 135 |
| | JJASO | 300 | 129 | 113 |
| Gulf of G | DJFM | 724 | 297 | 190 |
| | JJASO | 280 | 192 | 147 |
| South Af | DJFM | 219 | 132 | 172 |
| | JJASO | 400 | 245 | 197 |

| | | LT | FT | UT |
|---|---|---|---|---|
| India | DJF | 399 | 155 | 131 |
| | MAM | 310 | 194 | 130 |
| | JJA | 237 | 132 | 132 |
| | SON | 468 | 140 | 155 |
| North Af | DJFM | no data | no data | 137 |
| | JJASO | no data | no data | 110 |
| Middle E | DJFM | 238 | 143 | 140 |
| | JJASO | 239 | 125 | 115 |
| Gulf of G | DJFM | 708 | 283 | 183 |
| | JJASO | 289 | 196 | 146 |
| South Af | DJFM | 252 | 165 | 162 |
| | JJASO | 457 | 263 | 195 |

**Table A3.** q95 values (in ppb) used as thresholds for the different regions using data from 2002 to 2019 on the left and using data from 2010 to 2019 on the right

| | | LT | FT | UT |
|---|---|---|---|---|
| NW Am | DJF | 168 | 137 | 88 |
| | JJA | 66 | 87 | 133 |
| NE Am | DJF | 349 | 323 | 337 |
| | JJA | 409 | 589 | 1207 |
| Eur | DJF | 1192 | 1032 | 1180 |
| | JJA | 1701 | 1493 | 2186 |
| Sib | DJF | no data | no data | 181 |
| | JJA | no data | no data | 470 |
| E Asia | DJF | 480 | 944 | 1146 |
| | JJA | 415 | 711 | 937 |

| | | LT | FT | UT |
|---|---|---|---|---|
| India | DJF | 150 | 164 | 414 |
| | MAM | 128 | 121 | 507 |
| | JJA | 155 | 141 | 890 |
| | SON | 123 | 155 | 417 |
| North Af | DJFM | no data | no data | 433 |
| | JJASO | no data | no data | 1285 |
| Middle E | DJFM | 404 | 275 | 338 |
| | JJASO | 432 | 330 | 1282 |
| Gulf of G | DJFM | 144 | 303 | 484 |
| | JJASO | 328 | 269 | 756 |
| South Af | DJFM | 79 | 148 | 367 |
| | JJASO | 49 | 179 | 713 |

**Table A4.** Number of observed anomalies for the different regions and seasons.

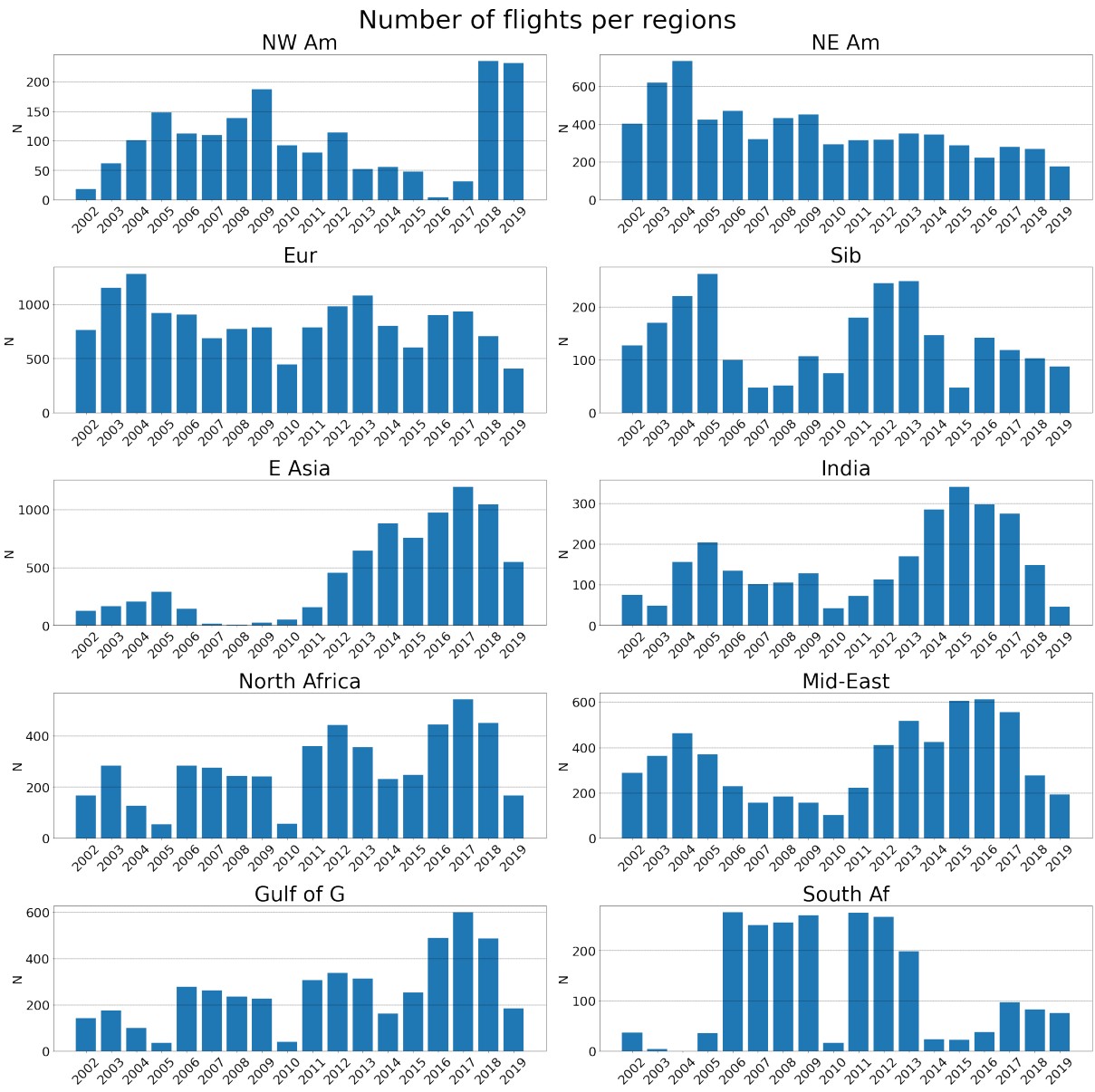

**Figure B1.** Data availability (number of measured flight per regions)

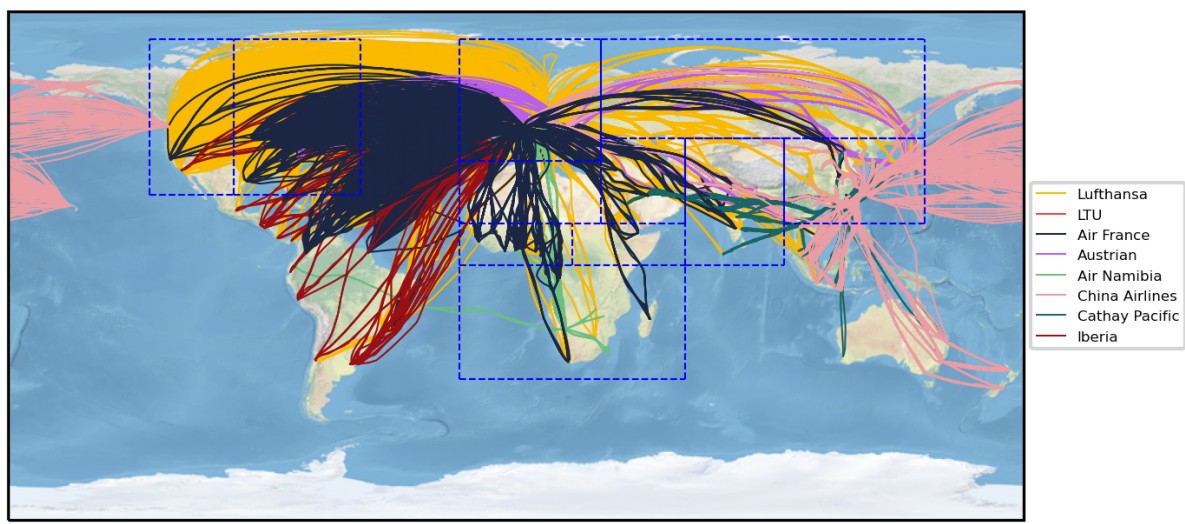

**Figure B2.** Trajectories of every IAGOS flights

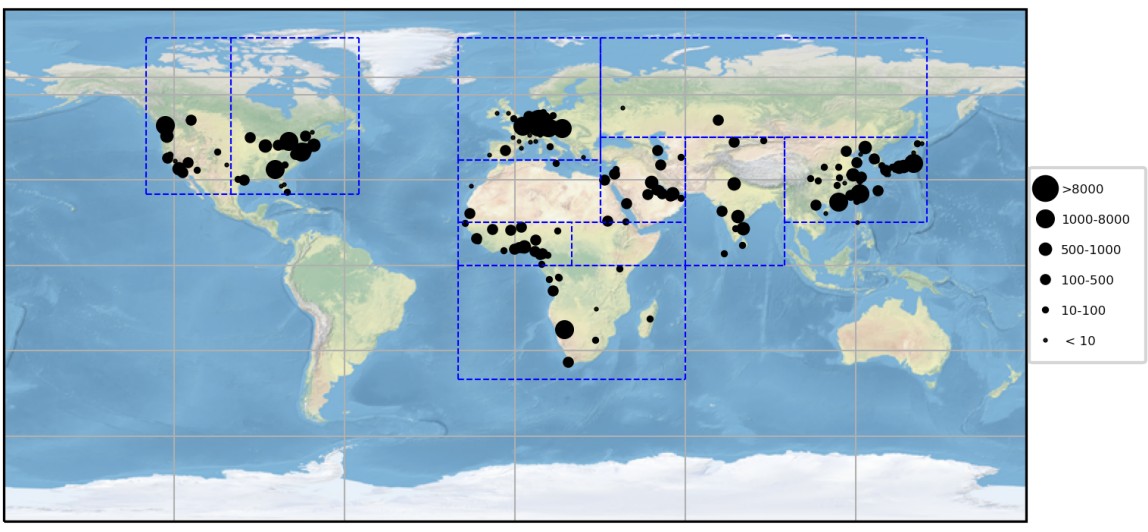

**Figure B3.** Map of the visited airports by IAGOS aircrafts

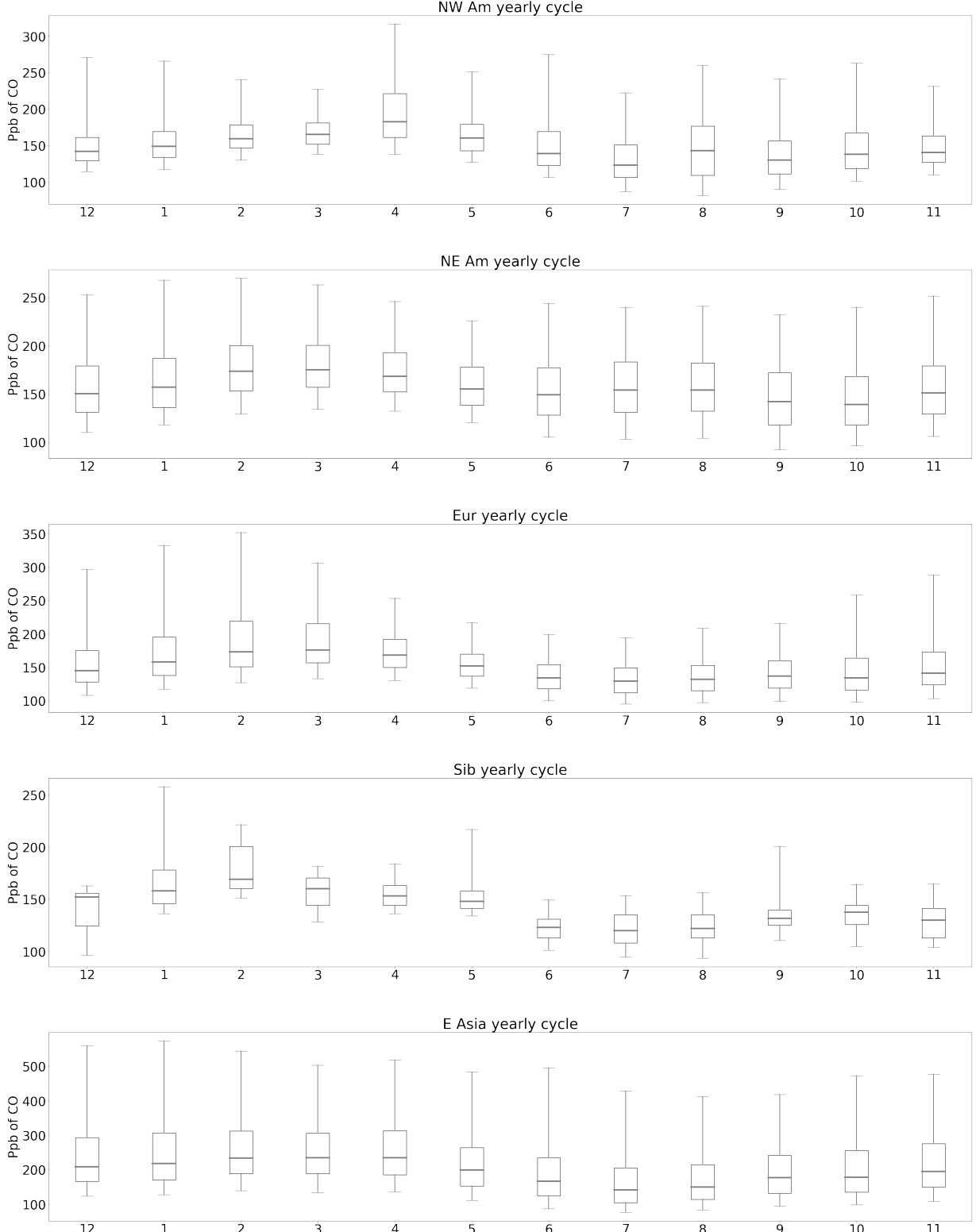

**Figure C1.** CO yearly cycle

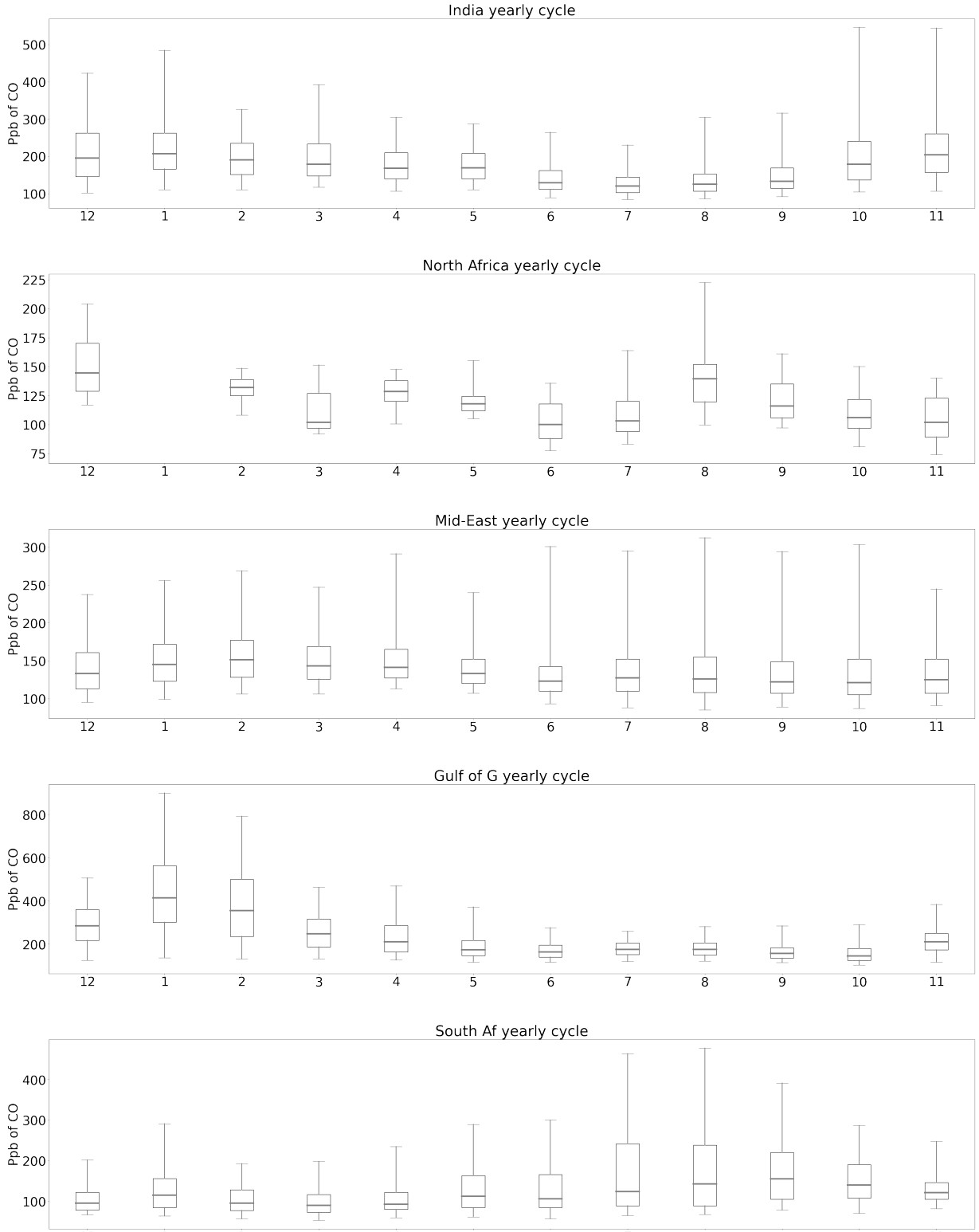

**Figure C2.** CO yearly cycle

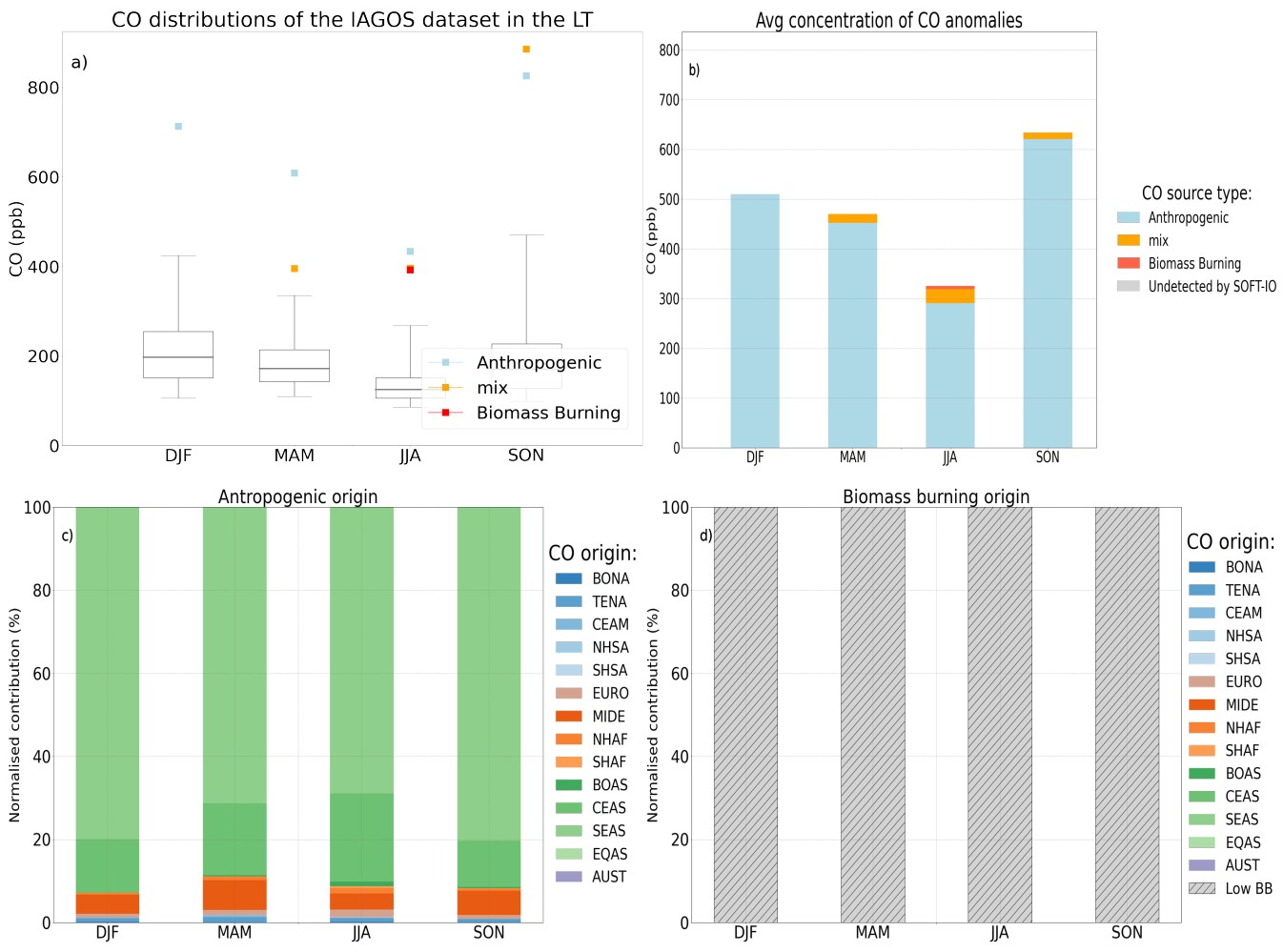

**Figure D1.** Same as Fig.3 but only for the Indian region during the four seasons for the LT (below 2000m). At this altitude no anomalies are undetected by SOFT-IO (in grey on panel b).

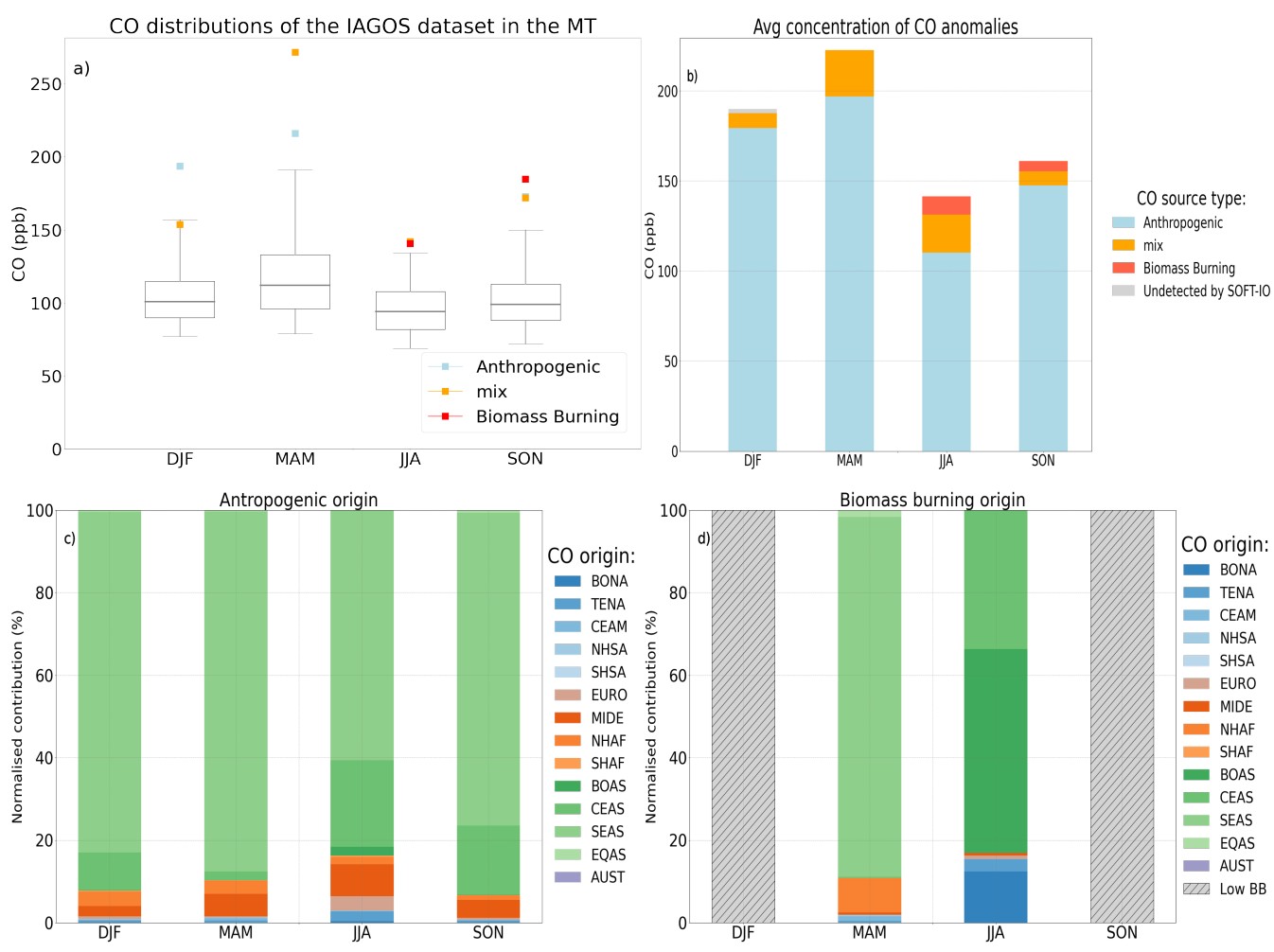

**Figure D2.** Same as Fig.3 but only for the Indian region during the four seasons for the MT (between 2000m and 8000m). At this altitude 2 anomalies are undetected by SOFT-IO (in grey on panel b).

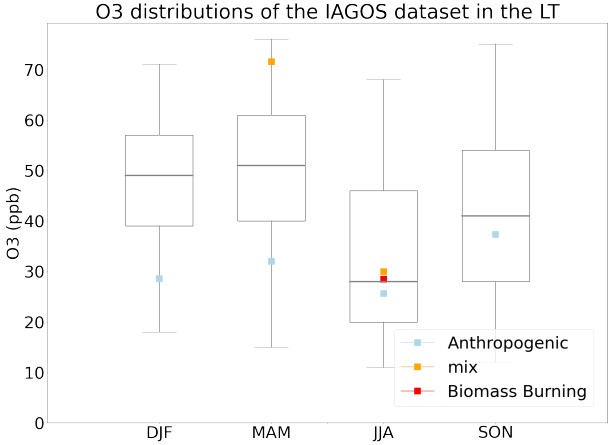

**Figure D3.** Same as Fig.4 for the LT in the Indian region during the four seasons.

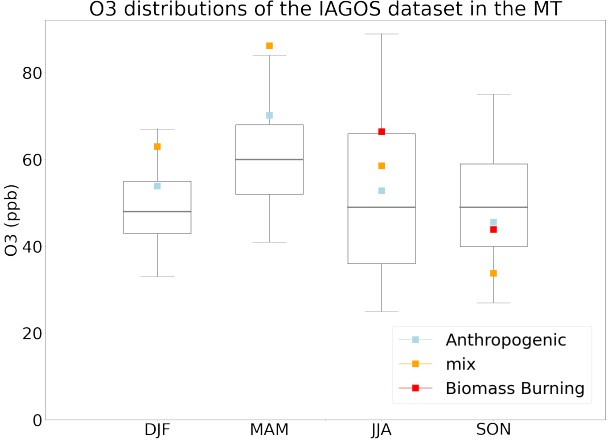

**Figure D4.** Same as Fig.4 for the MT in the Indian region during the four seasons.

*Author contributions.* TL,BS, BJ VM and VT designed the study. The IAGOS and SOFT-IO data were provided by BS, PW, YB, RB, DB, HC, JMC, PN and VT. The paper was written by TL and reviewed by BS, BJ and VT, and edited and approved by all the authors.

*Competing interests.* The authors declare that they have no conflict of interest.

*Acknowledgements.* We acknowledge the strong support of the European Commission, Airbus and the airlines (Deutsche Lufthansa, Air France, Austrian, Air Namimbia, Cathay Pacific, Iberia, China Airlines, Hawaiian Airlines, and Air Canada so far) that have carried the MOZAIC or IAGOS equipment and performed the maintenance since 1994. IAGOS has been funded by the European Union projects IAGOS–DS and IAGOS–ERI. Additionally, IAGOS has been funded by INSU-CNRS (France), Météo-France, Université Paul Sabatier (Toulouse, France) and Research Center Jülich (FZJ, Jülich, Germany). The IAGOS database is supported in France by AERIS (https://www.aeris-
data.fr)

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
