# Peer review of "Seasonal, regional and vertical characteristics of high carbon monoxide plumes along with their associated ozone anomalies as seen by IAGOS between 2002 and 2019"

_EGUsphere, 2023_

## Referee Comment (RC1)

Referee comment on "Seasonal, regional and vertical characteristics of high carbon monoxide plumes along with their associated ozone anomalies as seen by IAGOS between 2002 and 2019" by Lebourgeois et al.

This manuscript provides a statistical analysis of extreme CO values of IAGOS database for different regions, seasons and vertical layers (lower/middle/high troposphere), both in terms of origin (using SOFT-IO software) and in terms of impact on O3 production.

I found the manuscript to be well organised and clearly written. The methods used are scientifically sound and the figures chosen appropriately support the discussion and conclusions. The results provide an important overview of the CO plumes observed over 18 years of in situ measurements.

However, I find that there could be more links to the main processes involved, as well as a fuller discussion of how the results align with recent literature, including key publications using the IAGOS datasets that are cited in the article (but not only).

A discussion of how representative the in situ data used is of each major region should also be added, since I don't think that the data are randomly distributed within each region/layer.

**General comments**

1. Introduction.

I don't really see the relevance of the paragraph on satellite observations to this paper. However, a more precise description of previous results using CO and O3 observation datasets (in particular IAGOS) would be useful to better put the main results of this paper into perspective (in introduction and to discuss the results).

2. Methods

2.2: The description of the SOFT-IO software should include a paragraph on performance and uncertainties (in emission inventories and attribution using back trajectories). The important warning in § 122-127 could be written more clearly, and discussion of the performance in attribution in past studies could be helpful.

2.3.1: The data is divided into large regions for analysis. It would be important to discuss the location of the data analysed within each region and for each layer. If I understand the data correctly, the profiles correspond to specific airports and the flight paths also follow specific routes. I think it would be important to better discuss the representativeness of the data set for each region/layer.

2.3.2: For the type of source attributed to the anomalies detected (l.175-180) , why did you choose to use the main characteristic and why not add the fractional contributions? Is it because of too large uncertainties in attribution with SOFT-IO?

3. Results

For all regions and layers, I was wondering what fraction of the detected anomalies are successfully attributed to a main source using SOFT-IO?

The seasonal variations in the LT are mainly attributed to variations in the local sources. Does that mean that seasonal variations in the background levels has little impact? Even if anomalies are selected, the final concentration is an enhancement above background, both for CO and for O3.

l. 207: The authors mention "a cycle of O3 destruction in CO-rich air masses": O3 is then lower than background in the corresponding area? It would be helpful to add some detail on the corresponding chemical processes (same comment for all regions). What could be the impact of other co-emitted compound such as aerosols? I understand that this is beyond of the scope of this paper but for each region, it would be helpful to have some reference to the literature on the subject.

As mentioned in the general comments, it would be important to better discuss the results obtained in light of the literature, in terms of source contributions but also in terms of O3 enhancement in CO enriched air masses.

Although carefully conducted, the analysis reads a bit like a list. Perhaps a summary scatterplot of O3 versus CO could be used to get a more general view of the data set? A colour code could be used, for example, to differentiate regions / layers, etc.

**Specific comments**

l. 51: Other O3 precursors have a long lifetime, CH4 for instance.

l. 101: The paragraph on SOFT-IO should be included in the next section which is dedicated to the software.

l. 213: need reference for the larger number of convective events during summer.

l. 218: 'increased number of episodes…' increased compared to what?

l. 244-245: again, increased compared to? Large increase in East Asia vs Siberia are attributed to quite different sources. What processes are at play? What transport pathways? Does that hold if not only the main features are kept, but the fractional contribution from different sources? Is the situation still that contrasted?

l. 259-260: "which is probably due to the higher emission height…" Could this statement be checked? I agree that injection heights may be important here. What height is considered in SOFT-IO for source attributions?

l. 315: Why is O3 particularly low in BB plumes for this region? Has this event (2015) been analysed in past publications (even using other methods)?

In fact I also have the same question for other regions, such as African/ME BL, etc.

l. 367: Can BB be called 'wildfires' in this region? I would think there is significant contribution from agricultural burning as well. The use of the term 'wildfire' should probably be reviewed throughout the manuscript.

l. 462: same phrases repeated.

All figures for the statistical analyses: mention the total number of points in the subsets.

Figures O3 (Fig 4, etc): What do colored dots represent? Average value? Why not show the full boxplot for each source? Not enough data?

---

## Community Comment (CC1)

Comments by Owen R. Cooper (TOAR Scientific Coordinator of the Community Special Issue) on:

**Seasonal, regional and vertical characteristics of high carbon monoxide plumes along with their associated ozone anomalies as seen by IAGOS between 2002 and 2019**

Lebourgeois, T., B. Sauvage, P. Wolff, B. Josse, V. Marécal, Y. Bennouna, R. Blot, D. Boulanger, H. Clark, J-M Cousin, P. Nedelec, and V. Thouret (2024),

This manuscript was submitted to ACP as part of the TOAR-II Community Special Issue:
This review is by Owen Cooper, TOAR Scientific Coordinator of the TOAR-II Community Special Issue. I, or a member of the TOAR-II Steering Committee, will post comments on all papers submitted to the TOAR-II Community Special Issue, which is an inter-journal special issue accommodating submissions to six Copernicus journals:  ACP (lead journal), AMT, GMD, ESSD, ASCMO and BG. The primary purpose of these reviews is to identify any discrepancies across the TOAR-II submissions, and to allow the author teams time to address the discrepancies.  Additional comments may be included with the reviews. While O. Cooper and members of the TOAR Steering Committee may post open comments on papers submitted to the TOAR-II Community Special Issue, they are not involved with the decision to accept or reject a paper for publication, which is entirely handled by the journal's editorial team.

**General Comments:**

TOAR-II has produced two guidance documents to help authors develop their manuscripts so that results can be consistently compared across the wide range of studies that will be written for the TOAR-II Community Special Issue.  Both guidance documents can be found on the TOAR-II webpage:
https://igacproject.org/activities/TOAR/TOAR-II

*The TOAR-II Community Special Issue Guidelines*:   In the spirit of collaboration and to allow TOAR-II findings to be directly comparable across publications, the TOAR-II Steering Committee has issued this set of guidelines regarding style, units, plotting scales, regional and tropospheric column comparisons, tropopause definitions and best statistical practices.

*The TOAR-II Recommendations for Statistical Analyses*:  The aim of this guidance note is to provide recommendations on best statistical practices and to ensure consistent communication of statistical analysis and associated uncertainty across TOAR publications. The scope includes approaches for reporting trends, a discussion of strengths and weaknesses of commonly used techniques, and calibrated language for the communication of uncertainty. Table 3 of the TOAR-II statistical guidelines provides calibrated language for describing trends and uncertainty, similar to the approach of IPCC, which allows trends to be discussed without having to use the problematic expression, "statistically significant".

**Major Comments:**

This manuscript provides a detailed analysis of the observed carbon monoxide mixing ratios above several regions sampled by the IAGOS program, and their associated emissions source regions.  I find the analysis to be scientifically sound and the conclusions are consistent with the other papers submitted so far to the TOAR-II Community Special Issue.  However, to increase this papers relevance to the TOAR-II effort I would like to recommend four areas for further analysis and discussion:

1) There are some previous TOAR papers and other peer-reviewed studies that are relevant to this work and they should be cited (see papers referenced below).

2) This study touches on two important topics that have received little attention in the peer-reviewed literature, and the authors have an excellent opportunity to expand upon these topics. Specifically, these topics are the exceptionally high ozone mixing ratios in the UT above Siberia, and the high ozone levels in the lower, mid- and upper troposphere above the Middle East. Figure 5 (top left panel) of Gaudel et al. (2018) shows that the highest ozone values in the upper troposphere in June-July-August are found above Siberia. Figure 3c of Gaudel et al. (2020) shows that the ozone above the Middle East is even greater than ozone above China. These two regions have received relatively little attention in the peer-reviewed literature (an exception is Li et al., 2001). It would be helpful if these two regions can be given greater attention and highlighted in the abstract and conclusions as regions with exceptionally high ozone. Please elaborate on the potential contribution of biomass burning and anthropogenic emissions to the observed high ozone levels. The authors mention a potential contribution of the Asian summer monsoon to the high ozone levels above Siberia (i.e. the monsoon transports pollution from South and East Asia to Siberia). This is an excellent hypothesis and I think it should receive further attention.

3) Previous studies (Nowak et al., 2004, Figure 3; Cooper et al., 2002, Figure 8) have shown that scatter plots of ozone vs. CO are an effective way to highlight air pollution episodes and stratospheric intrusions (or air in the UTLS that is of stratospheric origin). These types of plots would be helpful for this study. For example, on line 277 the authors speculate that some of the high ozone values may be due to stratospheric influence. A scatter plot ozone vs CO could indicate instances of stratospheric intrusions.

4) Many of the study regions have well known trends in ozone and CO since 2000, but these trends are not addressed. The plots of average ozone and CO can therefore be misleading for certain regions. For example, during summertime, lower tropospheric ozone has decreased strongly in eastern North America since 2000, while it has increased in wintertime (see Figures 14 and 15 of Gaudel et al.; also see Chang et al., 2017). Lower tropospheric ozone has also increased in East Asia (Lu et al., 2018; Kim et al., 2023). Globally, CO has decreased since the 1990s. For example, trends of global CO levels can be found at the bottom of this webpage maintained by the NOAA Global Monitoring Laboratory: https://gml.noaa.gov/ccgg/figures/ (The figure called co_tr_global.pdf shows a decrease of CO since the year 2000). Also, extreme CO air pollution events have decreased across the USA since 2000: https://www.epa.gov/air-trends/carbon-monoxide-trends
Because ozone and CO have changed in many regions since the year 2000 it would be helpful to compare regions using data from the most recent years, when possible, rather than using 20-year averages.

**Minor Comments:**

The paper contains many grammatical errors that should be corrected. A co-author with excellent English skills should carefully proofread the entire text.

Line 191
In addition to increased CO emissions in winter, there is also a well-known increase in CO lifetime in winter (due to less ozone and OH), which also explains the higher wintertime concentrations (Novelli et al., 1998).

Discussion of transport processes impacting India should reference previous work on this topic, e.g. Lal et al. 2014; Lawrence and Lelieveld, 2010; Lelieveld et al., 2001.

**References:**

Chang, K-L, I. Petropavlovskikh, O. R. Cooper, M. G. Schultz and T. Wang (2017), Regional trend analysis of surface ozone observations from monitoring networks in eastern North America, Europe and East Asia, *Elem Sci Anth., 5:50*, DOI: http://doi.org/10.1525/elementa.243

Cooper, O. R., J. L. Moody, D. D. Parrish, M. Trainer, T. B. Ryerson, J. S. Holloway, G. Hübler, F. C. Fehsenfeld, and M. J. Evans (2002), Trace gas composition of midlatitude cyclones over the western North Atlantic Ocean: A conceptual model, *J. Geophys. Res., 107*(D7), 4056, doi:10.1029/2001JD000901.

Gaudel, A, et al. 2018. Tropospheric Ozone Assessment Report: Present-day distribution and trends of tropospheric ozone relevant to climate and global atmospheric chemistry model evaluation. Elem Sci Anth, 6: 39. DOI: https://doi.org/10.1525/elementa.291

Gaudel, A., O. R. Cooper, K.-L. Chang, I. Bourgeois, J. R. Ziemke, S. A. Strode, L. D. Oman, P. Sellitto, P. Nédélec, R. Blot, V. Thouret, C. Granier (2020), Aircraft observations since the 1990s reveal increases of tropospheric ozone at multiple locations across the Northern Hemisphere. Sci. Adv. 6, eaba8272, DOI: 10.1126/sciadv.aba8272

Kim, S.-W., Kim, K.-M., Jeong, Y., Seo, S., Park, Y., and Kim, J.: Changes in surface ozone in South Korea on diurnal to decadal timescales for the period of 2001–2021, Atmos. Chem. Phys., 23, 12867–12886, https://doi.org/10.5194/acp-23-12867-2023, 2023.

Lal, S., S. Venkataramani, N. Chandra, O. R. Cooper, J. Brioude, and M. Naja (2014), Transport effects on the vertical distribution of tropospheric ozone over western India, J. Geophys. Res. Atmos., 119, doi:10.1002/2014JD021854.

Lawrence, M. G. and Lelieveld, J.: Atmospheric pollutant outflow from southern Asia: a review, Atmos. Chem. Phys., 10, 11017–11096, https://doi.org/10.5194/acp-10-11017-2010, 2010.
Lelieveld, et al. (2001), The Indian Ocean experiment: widespread air pollution from South and Southeast Asia, Science, 291, DOI: 10.1126/science.1057103

Li, Q, et al. 2001. A tropospheric ozone maximum over the Middle East. Geophysical Research Letters 28: 3235–3238. DOI: https://doi.org/10.1029/2001GL013134

Lu, X., J. Hong, L. Zhang, O. R. Cooper, M. G. Schultz, X. Xu, T. Wang, M. Gao, Y. Zhao, Y. Zhang (2018), Severe surface ozone pollution in China: a global perspective, Environ. Sci. Technol. Lett. 5, 487-494.

Novelli, P. C., K. A. Masarie, and P. M. Lang (1998), Distributions and recent changes of carbon monoxide in the lower troposphere, J. Geophys. Res., 103(D15), 19015–19033, doi:10.1029/98JD01366.

Nowak, J. B., et al. (2004), Gas-phase chemical characteristics of Asian emission plumes observed during ITCT 2K2 over the eastern North Pacific Ocean, J. Geophys. Res., 109, D23S19, doi:10.1029/2003JD004488.

---

## Author Comment (AC1)

**Review response 1**

**June 2024**

*Referee comment on "Seasonal, regional and vertical characteristics of high carbon monoxide plumes along with their associated ozone anomalies as seen by IAGOS between 2002 and 2019" by Lebourgeois et al. This manuscript provides a statistical analysis of extreme CO values of IAGOS database for different regions, seasons and vertical layers (lower/middle/high troposphere), both in terms of origin (using SOFT-IO software) and in terms of impact on O3 production. I found the manuscript to be well organised and clearly written. The methods used are scientifically sound and the figures chosen appropriately support the discussion and conclusions. The results provide an important overview of the CO plumes observed over 18 years of in situ measurements. However, I find that there could be more links to the main processes involved, as well as a fuller discussion of how the results align with recent literature, including key publications using the IAGOS datasets that are cited in the article (but not only). A discussion of how representative the in situ data used is of each major region should also be added, since I don't think that the data are randomly distributed within each region/layer.*

We thank the reviewer for her/his comments that will help improving our study. We respond below to each specific point.

**1 Introduction**

*I don't really see the relevance of the paragraph on satellite observations to this paper. However, a more precise description of previous results using CO and O3 observation datasets (in particular IAGOS) would be useful to better put the main results of this paper into perspective (in introduction and to discuss the results).*

We thank the reviewer for this comment and as suggested, the following paragraphs in blue have been added in the introduction section (lines 71-83 in the revised manuscript):

Some studies have used the IAGOS database to study the characteristics of CO and $O_3$ values in the troposphere and lower stratosphere. This is the case for Cohen et al. [2018], which used this dataset to study the climatology and trends in O3 and CO in the UTLS. Petetin et al. [2018b], Lannuque et al. [2021], Tsivlidou et al. [2022] used IAGOS to study the characteristics of CO in different regions or altitude layers of the world. Tsivlidou et al. [2022] studied CO and O3 characteristics in the tropical regions. She highlighted the origins of the CO in the different regions of the tropics. She specifically showed the importance of the Anthropogenic emissions to

explain the values of CO in the tropical troposphere. Lannuque et al. [2021] studied the meridional distribution of O3 and CO over Africa using IAGOS and the satellite IASI (Infrared Atmospheric Sounding Interferometer). They showed the importance of the ITCZ and the upper branch of the Hadley cell for the redistribution of the pollutants over Africa. The Pollutant emitted at the surface is transported by trade winds toward the ITCZ where it is transported to the UT and redistributed to higher latitude by the Hadley cell. Petetin et al. [2018b] studied the CO vertical profile over different airport clusters. They characterised their seasonal profile as well as the seasonality of the highest CO anomalies 95 and 99 percentile. They showed a strong seasonal variability of the most extreme anomalies in northern America which were due to BB emissions. He also looked at the origins of the CO responsible for the CO anomalies at the different airport clusters.

**2 Methods**

*2.2: The description of the SOFT-IO software should include a paragraph on performance and uncertainties (in emission inventories and attribution using back trajectories). The important warning in § 122-127 could be written more clearly, and discussion of the performance in attribution in past studies could be helpful.*

Thank you for this comment that will improve the manuscript. The performance and uncertainties of SOFT-IO are described in Sauvage et al. [2017], Tsivlidou et al. [2022]. In order to provide key information in this study, the following paragraph has been added in the revised version, lines 142-148 :

Sauvage et al. [2017] and Tsivlidou et al. [2022] made a thorough statistical evaluation of SOFT-IO. The model had a really good score in the detection frequency of the CO anomalies (above 93% on average). Detection frequency was at its maximum in the LT as most anomalies are from local emissions at this altitude. In the MT and UT the scores were lower but remained above 80% as the simulation of horizontal and vertical transport could suffer some errors. It is important to note that the study presented here aims at using SOFT-IO only as a qualitative tool to attribute a source type and a relative geographical origin to the emissions leading to the detected anomalies. SOFT-IO is a model which has already been used in several studies similar to the current study (e.g Petetin et al. [2018b], Cussac et al. [2020], Lannuque et al. [2021], Tsivlidou et al. [2022]).

*2.3.1: The data is divided into large regions for analysis. It would be important to discuss the location of the data analysed within each region and for each layer. If I understand the data correctly, the profiles correspond to specific airports and the flight paths also follow specific routes. I think it would be important to better discuss the representativeness of the data set for each region/layer.*

Thank you for the comment. As you advised, we added a map showing the position of the flight trajectories done by the IAGOS aircraft in each region. We have added the following paragraph (lines: 186-189) in the revised version :

IAGOS samples the lower and free troposphere during landing and take-off. Petetin et al. [2018a] showed that close to the surface, the IAGOS measurements are representative of urban

areas and provide similar measurements to urban background stations. At higher altitudes, in the free troposphere, the samples are less influenced by local emissions and therefore are representative of regional background conditions following the flight tracks showed in Fig.B2 in the appendix.

A figure showing the flight tracks of every IAGOS flight has been added to the appendix. This corresponds to the cruise parts of the flights in the UT. Regarding LT and MT, data correspond to the vertical profiles part of the flight during take-off and landing over visited airports (the second map). Note that the average horizontal distance between airport surface and the 8 km altitude is about 300 km [Petetin et al., 2018a].

[Figure]

Figure 1: Trajectories of every IAGOS flights

[Figure]

Figure 2: Map of the visited airports by IAGOS aircrafts

*2.3.2: For the type of source attributed to the anomalies detected (l.175-180) , why did you choose to use the main characteristic and why not add the fractional contributions? Is it because*

*of too large uncertainties in attribution with SOFT-IO?*

As there are, by nature, anthropogenic emissions in almost every single plume, the objective of the methodology was to characterize if biomass burning plumes are dominant (i.e. >50%) or not (MIX, and of course ANT). That is the reason why we compile the main characteristics. The objective of this study is to see what is the dominant source in influencing CO anomalies.

**3    Results**

*For all regions and layers, I was wondering what fraction of the detected anomalies are successfully attributed to a main source using SOFT-IO?*

Thank you for the comment. As you advised in the methods part of the comment we have added a paragraph on the uncertainty of SOFT-IO from the previous study of Sauvage et al. [2017], Tsivlidou et al. [2022]. According to these studies on average 93% of the anomalies observed by IAGOS are also detected by SOFT-IO.

*The seasonal variations in the LT are mainly attributed to variations in the local sources. Does that mean that seasonal variations in the background levels has little impact? Even if anomalies are selected, the final concentration is an enhancement above background, both for CO and for O3.*

Yes, CO anomalies are selected and calculated over a regional and seasonal background. So, yes the seasonal variations are caused by variations in the local sources but also by changes in the background values (caused by various factors like chemical lifetime, higher emissions and less mixing). There is no selection of Ozone anomalies. The Ozone distributions are those sampled within the CO anomalies.

*l. 207: The authors mention "a cycle of O3 destruction in CO-rich air masses": O3 is then lower than background in the corresponding area? It would be helpful to add some detail on the corresponding chemical processes (same comment for all regions). What could be the impact of other co-emitted compound such as aerosols? I understand that this is beyond of the scope of this paper but for each region, it would be helpful to have some reference to the literature on the subject.*

Thank you for the comment. The following references have been added :

- Yang et al. [2019]

- Chang et al. [2017b].

- Lu et al. [2018]

- Gaudel et al. [2018]

- Cohen et al. [2018]

- Nowak et al. [2004]

- Hudman et al. [2004]

The following paragraph regarding Ozone in the paper have been modified giving thus further clarification.

[revised manuscript text omitted]

Reviewer is right that a complete and more in-depth analysis on the ozone distributions is beyond the scope of this paper. This is the first step of this statistical characterisation, with diagnostics on CO anomalies along with their ozone content. That gives material and diagnostics for additional studies with global models (CTM) to synthesize and integrate all the processes leading to ozone formation.

*As mentioned in the general comments, it would be important to better discuss the results obtained in light of the literature, in terms of source contributions but also in terms of O3 enhancement in CO enriched air masses.*

Thank you for the comment, we made important modification and added a lot of references in

the results section of the revised manuscript in order to better discuss the result in light of the literature :

- Bergman et al. [2013]

- Chang et al. [2017a]

- Cohen et al. [2018]

- Cooper and Parrish [2004]

- Cooper et al. [2004]

- Dentener et al. [2006]

- Ding et al. [2009]

- Duncan et al. [2008]

- Field et al. [2016]

- Gaudel et al. [2018]

- Huang et al. [2012]

- Huntrieser and Schlager [2004]

- Jaffe et al. [1999]

- Kar et al. [2004]

- Lal et al. [2014]

- Lawrence and Lelieveld [2010]

- Lawrence [2004]

- Lelieveld et al. [2001]

- Li et al. [2002]

- Li et al. [2001]

- Liang et al. [2007]

- Lu et al. [2018]

- [Novelli et al., 1998]

- Pan et al. [2016]

- Sauvage et al. [2005]

- Stohl [2001]

- Stohl et al. [2002]

- Yang et al. [2019]

The following paragraphs in the revised manuscript have been modified :

[revised manuscript text omitted]
. In this region Lagos airport is the most visited by IAGOS flights and the accumulation of the pollutant observed in the LT during this season has already been characterised in Sauvage et al. [2005] and is caused by the Harmattan winds bringing rich CO air masses caused by the upwind fires. In JJASO, the southwesterly trade winds bring air-masses from the Atlantic ocean. These air-masses are cleaner with respect to anthropogenic pollution but can bring BB plumes from Southern Africa. Lines (399-406)

- The Middle East plumes have a high contribution from anthropogenic emissions in both seasons in the LT and the MT. The Middle East has been identified in previous studies as receiving the pollution of multiple regions [Li et al., 2001, Stohl et al., 2002, Duncan et al., 2008]. Europe is mostly exporting its pollution via low altitude pathways and we can see on Fig.11.c and Fig. 12.c that up to 20 % of the anthropogenic contributions can come from Europe. There are also contributions from Temperate North America and South and East Asia, but contrary to the European contributions these probably followed higher altitude pathways before sinking to the MT or LT [Li et al., 2001, Stohl et al., 2002]. We can also see important differences in the provenance of the anthropogenic contributions between DJFM and JJASO. In JJASO, we are seeing contributions mostly from the local regions (MIDE) similarly to the contributions in the LT. According to previous studies the Planetary Boundary Layer in this region can reach 4000 or 5000 meters in JJA [Gamo, 1996, Ntoumos et al., 2023]. So, this differences in the origins of the contributions between DJFM and JJASO may be caused by the higher PBL height in JJASO. Lines (416-426)

*Although carefully conducted, the analysis reads a bit like a list. Perhaps a summary scatterplot of O3 versus CO could be used to get a more general view of the data set? A colour code could be used, for example, to differentiate regions / layers, etc.*

The ozone distributions are only from the tropospheric branches because stratospheric air-masses are discarded and only the air masses with extreme values of CO are selected. We believe the ozone box plots are more informative (see figure 3 below). So, after exploration of this idea, we think that the suggested scatter plots are not meaningful for a summary.

[Figure]

Figure 3: Scatter plot Ozone vs CO in the observed anomalies

**4   Specific comments :**

*l. 51: Other O3 precursors have a long lifetime, CH4 for instance.*

Corrected : "CO is one of the only with a long enough chemical lifetime"
*l. 101: The paragraph on SOFT-IO should be included in the next section which is dedicated to the software.*

Done Line 120.

*l. 213: need reference for the larger number of convective events during summer.*

Deleted.

*l. 218: 'increased number of episodes...' increased compared to what?*

Sentence removed.

*l. 244-245: again, increased compared to?*

Corrected: "The mean mixing ratio of these episodes during the summer months also increased significantly compared to their winter values." (line 303)

*Large increase in East Asia vs Siberia are attributed to quite different sources. What processes are at play? What transport pathways?*

In East Asia the important summertime maximum is attributed to local emissions and the rapid vertical transport of the East Asian monsoon. In Siberia the maximum is both due to Boreal fires via pyroconvection [Nedelec et al., 2005] but also to anthropogenic emissions from East Asia transported by the east Asian monsoon.

*Does that hold if not only the main features are kept, but the fractional contribution from different sources? Is the situation still that contrasted?*

We tried both methods and no major differences were observed between the two methods.

*l. 259-260: "which is probably due to the higher emission height..." Could this statement be checked? I agree that injection heights may be important here. What height is considered in SOFT-IO for source attributions?*

Reference added [Dentener et al., 2006] (line 321). SOFT-IO uses emissions height given by GFAS (Rémy et al. [2017]). Those emissions heights are computed by the plume rise model from Paugam et al. [2015].

*l. 315: Why is O3 particularly low in BB plumes for this region? Has this event (2015) been analysed in past publications (even using other methods)? In fact I also have the same question for other regions, such as African/ME BL, etc.*

Those anomalies were from Tropical Asia and, according to Cohen et al 2018, it is a region with a lower O3 environment than India. Sentence added (line 397):
The values of $O_3$ in the BB plumes are low and close to the 25th percentile (44 ppb) which is explained by the lower background values of $O_3$ in Equatorial Asia compared to India [Cohen et al., 2018].

*l. 367: Can BB be called 'wildfires' in this region? I would think there is significant contribution from agricultural burning as well. The use of the term 'wildfire' should probably be reviewed throughout the manuscript.*

The term "wildfire" has been replace by "fire" in the revised manuscript.

*l. 462: same phrases repeated.*

Done

*All figures for the statistical analyses: mention the total number of points in the subsets.*

Good point, table added to the appendix (see table1 below).

*Figures O3 (Fig 4, etc): What do colored dots represent? Average value? Why not show the full boxplot for each source? Not enough data?*

|       |     | LT      | FT      | UT   |
|-------|-----|---------|---------|------|
| NW Am | DJF | 168     | 137     | 88   |
|       | JJA | 66      | 87      | 133  |
| NE Am | DJF | 349     | 323     | 337  |
|       | JJA | 409     | 589     | 1207 |
| Eur   | DJF | 1192    | 1032    | 1180 |
|       | JJA | 1701    | 1493    | 2186 |
| Sib   | DJF | no data | no data | 181  |
|       | JJA | no data | no data | 470  |
| E Asia| DJF | 480     | 944     | 1146 |
|       | JJA | 415     | 711     | 937  |

|          |       | LT      | FT      | UT   |
|----------|-------|---------|---------|------|
| India    | DJF   | 150     | 164     | 414  |
|          | MAM   | 128     | 121     | 507  |
|          | JJA   | 155     | 141     | 890  |
|          | SON   | 123     | 155     | 417  |
| North Af | DJFM  | no data | no data | 433  |
|          | JJASO | no data | no data | 1285 |
| Middle E | DJFM  | 404     | 275     | 338  |
|          | JJASO | 432     | 330     | 1282 |
| Gulf of G| DJFM  | 144     | 303     | 484  |
|          | JJASO | 328     | 269     | 756  |
| South Af | DJFM  | 79      | 148     | 367  |
|          | JJASO | 49      | 179     | 713  |

Table 1: Number of observed anomalies for the different regions and seasons.

Yes the colored dot represents the average values. Yes, showing a boxplot different for each source would be challenging in some regions where not a lot of anomalies from a certain source are found.

---

## Author Comment (AC2)

**Review response 2**

**June 2024**

*The paper presents a seasonal analysis of CO pollution plumes (anomalies) sampled by IAGOS commercial aircraft over different regions of the world for the period 2002-2019. Modeled footprints and global emission inventories for CO anthropogenic and biomass burning are used to simulate contributions to CO along each flight track and attribute the observed anomalies to emissions by type and by source region. I assume this study was made possible thanks to a lot of careful work and continued support for the IAGOS program but the paper does not give much details about this although it provides several references for previous analyses of the data. The authors use footprints and emission inventories "semi quantitatively" for emission attribution, I assume previous work has shown this is a reliable approach. The paper clearly presents graphic summaries for the CO anomaly analysis by region and the text further describes how seasonality in some atmospheric transport processes and emissions can explain the results. The discussion of ozone levels in the anomaly plumes is mostly descriptive by region. It seems that in only a few cases do CO anomalies correspond to ozone anomalies. Is there another paper that looks more holistically at those ozone anomalies and the processes behind them? It would be nice to help the reader understand the significance (and maybe limitations) of your analysis and findings for ozone. The importance of the IAGOS dataset and this work may be made stronger with a more organized argumentation in the introduction. Some of the text there is repetitive and some general statements are not backed up by references. The conclusion mostly summarizes the findings but could maybe also be more explicit about future work and why continuing and potentially expanding those measurements, adding other tracers... is important for the next decades.*

We thank the reviewer for his/her important comments that will help improve this paper. We respond below to each specific point.

A few references on ozone have been added in order to better discuss our results in light of the literature:

- Yang et al. [2019]

- Chang et al. [2017].

- Lu et al. [2018]

- Gaudel et al. [2018]

- Cohen et al. [2018]

- Nowak et al. [2004]

- Hudman et al. [2004]

There are two important references of the ITCT 2k2 campaign over the northern pacific ocean and focusing on the chemical characteristics of pollution plumes from east asia. [Nowak et al., 2004, Hudman et al., 2004].

To clarify, ozone is not the focus of this paper as it is a first step before using a global CTM. Here, only levels of ozone are presented, since only CO and O3 measurements are available at this scale. We can not do a complete analysis of the processes behind each value but we plan to make a follow up analysis on the processes leading to ozone in pollution plume with a chemistry transport model. A paragraph on those perspectives have been added to the conclusions.

A paragraph on the Ozone limitation and perspectives has been added to the conclusions (lines 600-605): These $O_3$ values give information on its possible production in polluted plumes. However, without the measurements of additional chemical compounds (like VOCs and NOx for example) it is difficult to draw robust conclusions. To go further into the analysis on the $O_3$ in pollution plumes, information on more chemical compounds is required. The current perspective is to carry a similar study with a Chemistry Transport Model in order to get further information on the provenance of $O_3$ values but also on the amount of $O_3$ productions in polluted plumes, especially in regions with elevated values of $O_3$ like Siberia and the Middle East.

An other paragraph on the perspective about the importance of the IAGOS measurements as well as future perspective has been added to the conclusions (lines 590-594):

We have presented a detailed analysis of the characteristics of high carbon monoxide plumes and their associated ozone anomalies in different regions of the world. It is important for the IAGOS infrastructure to continue those measurements and to expand the regions sampled by the research infrastructure in order to provide these diagnostics in additional regions. This is particularly important in the tropics, where anthropogenic emissions are increasing and impact on the $O_3$ trend globally [Zhang et al., 2016]. Increased number and sampling frequency of measurements of NOx and aerosols by IAGOS will be available and valuable for future analysis focusing on $O_3$ photochemical production or air quality.

**1 High level comments:**

*Are the findings new?*

Yes, in terms of (i) synthesis study with dense and global data sets (ii) allowing a robust statistical analysis (iii) It is one of the only study focused on the extreme pollution anomalies around the world.

*What are some key implications ?*

- We thus provide diagnostics robust enough to further allow any "smart-evaluation" analysis

of global model pursuing the goal of having the good mixing ratios for the good reasons (CO in particular here).

- We show the significant impact of anthropogenic emissions and in particular of certain key regions on the CO anomalies in the MT and UT.

- We show that the most extreme anomalies are almost always related to biomass burning emissions.

- Ozone on average shows higher values in CO plumes and can even reach very high values under certain conditions.

*Why are trends or interannual variability not explored? I think I may be able to guess but you may want to be explicit about it in the paper, ie. if the dataset year to year spatiotemporal coverage does not allow for this type of analysis.*

As guessed, this is beyond the scope of the paper because the spatiotemporal coverage does not allow for this type of analysis. Some regions are sampled regularly for only a few years as it can be seen in the figure B1 in the appendix. We focus here on the "ID" of CO anomalies (i.e. where do they come from ? which transport pathway ? how is ozone inside such anomalies ? ...).

*Be explicit about the nature of the IAGOS dataset for people less familiar with this work: Mention they are measurements on commercial aircraft, in the introduction and mention IAGOS in the conclusion too.*

We thank the reviewer for the comment, as advised we added the following sentences in blue to the introduction and conclusion of the revised manuscript.

- IAGOS (In-service Aircraft for a Global Observing System; http://www.iagos.org) is a European research infrastructure using commercial aircraft in order to measure the atmosphere composition. Lines (86-88)

- IAGOS is a research infrastructure which uses commercial aircraft to measure atmospheric composition. Lines (488-489)

*The consistency of the data calibration and the data quality throughout the period and across instruments is assumed but it may be nice to include a couple of sentences on that.*

The following sentence has been modified in the methods section (lines 109-111).
The consistency between the MOZAIC, IAGOS and CARIBIC datasets as well as the internal consistency of the CO and O3 measurements since 1994 have been tested [Nédélec et al., 2015, Blot et al., 2021].

*Further define the CO anomalies: how many consecutive datapoints above the q95 threshold are needed to become an anomaly plume?*

The definition of the CO anomalies can be found in the methods section (lines 194-195) :

The CO anomalies are defined as CO values exceeding the threshold for three consecutive measurements (i.e. a distance of approximately 3 km during cruise phase).

*I found a few typos or small corrections. Another thorough reading would be great to make sure all of these are taken care of. For example, fix a few inconsistencies throughout the article about how you refer to your regions.*

Thank you for your comment. A thorough proofreading has been made by one of the co-authors, native english speaker.

**2 Detailed comments:**

**2.1 Abstract:**

*First sentence should be clear the analysis is done for large regions of the globe.*

In-situ measurements from IAGOS are used to characterise extreme values of carbon monoxide (CO) in large regions of the globe. (line 1-2)

*You cover some of the findings for some region but results for India are not mentioned, even though they have their own section.*

L14: Indian CO anomalies have drastically different characteristics depending on the season as the wet and dry phases of the monsoon have an important impact on the transport of the pollutant in this regions.

*The much higher CO in anomalies over E Asia may be nice to mention here too.*

L9: The largest values of CO are found in Eastern Asia in the lower and middle troposphere.

**2.2 Introduction:**

*References would be great for statements on model limitations to reproduce or predict extreme weather events and extreme pollution events*

Modified (lines 21-23): Extreme weather can sometimes be incorrectly reproduced and predicted by the global and regional models (e.g. Shastri et al. [2017], Lavaysse et al. [2019]). Extreme pollution events can also be difficult to predict, as they can be explained by multiple factors such as abnormal weather conditions and/or unusually intense emissions (either from anthropogenic or natural sources, or both).

*Not clear about the impact of extreme pollution events on climate, maybe expand on what you mean with climate here and add references.*

Deleted

*Pollution is often referring to conditions in the boundary layer. What does it mean for the troposphere?*

Pollution is often from the BL as it is emitted at the surface. After emissions it can however be exported out of the BL.

*The text in the introduction makes it sound like this paper/study can be used to improve model simulations of extreme pollution plumes. How would this be done?*

Probably not directly, but the purpose of this study was to better understand the origins of the CO and the main characteristics of the pollution and CO anomalies. Global model can use the diagnostics given here to verify that the model's pollution patterns have similar characteristics and have the right mixing ratios for the right reasons.

*L 27-30: "This compound In the troposphere, ozone is photochemically produced from NOx and VOC (Volatile Organic Compounds)/ or CO (Seinfeld and Pandis, 2008). Hence, a good estimation of its chemical precursors as well as better understanding of the processes leading to their distributions at global scale is of prime importance."*

Done line 32.

*L 44: Owen et al., 2006 should be Cooper et al., 2006 (Owen is the firstname and Cooper is the lastname of the author).*

The first author is R. C. Owen, Owen Cooper is second author.

*L 72-75: "We present here a quasi-global overview over almost 20 years of extreme CO mixing ratios and their associated O3 values, as seen by IAGOS. The goal of this paper is to characterise*

*the seasonal, regional and vertical CO mixing ratios anomalies for different regions over the globe for almost 20 years as seen by IAGOS along with the simultaneously recorded O3 between 2002 and 2019." These two sentences say the same thing. Please remove repetition.*

Done line 90

*L 101-109: Is the last paragraph on the model needed in the measurement data set section 2.1? It is mostly repeated in section 2.2. Similarly the first paragraph in 2.2 repeats some of what is in section 2.1. Please revise to focus on what belongs in each section.*

Done line 120.

**2.3   Methods:**

*L 116: "The Bbiomass Bburning emission inventory used in this version..." remove uppercase letters from biomass burning and check if this should be singular, or plural.*

Done line 130.

*Section 2.2 : In the model, you only look at direct emissions of CO not CO chemically produced?*

Exactly, we mostly focus on the emitted CO in the study as it is harder to account for CO production without a chemistry transport model.

*Figure 1 may have better contrast for the Americas if the oceans were kept white. Could the legend be placed outside of the map to not cover part of it and you can make it a little bigger too? Are the acronyms for the GFED regions defined somewhere in your paper? Especially as you refer to boreal emissions several times, I assume you refer to emissions from BONA and BOAS.*

Thank you for the comments, the figure has been fixed and a table of the acronyms has been added to the appendix (see table 1 below).

*L 165: "At higher altitude, the samples are less influenced by local emissions..."*

Done: L191.

*Figure 2: There are two blue lines, so the CO measurements one would need to be referred to as dark blue. There are horizontal and vertical dashed lines. Are the vertical ones needed? Clarify*

| Acronym | Full name |
|---------|-----------|
| BONA | Boreal North America |
| TENA | TEmperate North AMerica |
| CEAM | CEntral AMerica |
| NHSA | North Hemisphere South America |
| SHSA | South Hemisphere South America |
| EURO | Europe |
| MIDE | MIDdle East |
| NHAF | Northern Hemisphere AFrica |
| SHAF | South Hemisphere AFrica |
| BOAS | BOreal ASia |
| CEAS | CEntral Asia |
| SEAS | South East Asia |
| EQAS | Equatorial Asia |
| AUST | AUSTralia |

Table 1: Table GFED acronym:

*you refer to the horizontal dashed line for the 95th percentile for the CO for that region/season; you could give the value for q95 in the caption. What altitudes did the measurements in the Figure cover? What happens during the data gaps seen in the Figure?*

Caption corrected

*Table 1: Specify this is for CO and for different seasons in the caption. Put the unit (ppb) in the caption, not the table itself. Explain what "no data" means. Do you need to show results for seasons you will not discuss.*

Caption corrected

*L 172-173: "SOFT-IO is then used as a qualitative tool to assign a source type to each of the detected anomalies. This diagnostic is only applied if the contributions modelled by SOFT-IO are above a detection threshold defined as 5 ppb." You use 5 ppb for all altitude bins? Does it matter?*

Yes but this criteria is almost only important for the UT layers where concentration and SOFT-IO contributions are low. Sauvage et al. [2017] showed that the detection frequencies of CO plume were decreasing at higher altitudes.

**2.4   Results:**

*Figure 3: You would need to define Low BB in the caption.*

Done

*Legend is covering the a) in the first plot.*

Corrected

*Can you comment on the high mean for DJF and JJA CO anomalies in E Asia, plots a and b? What anthropogenic sources contribute the most here?*

A paragraph on the high values of CO in East Asia has been added in the revised manuscript (lines 245-242) in blue below. At this altitude, the highest values of CO are found in Eastern Asia during both seasons. The anomalies can even reach a mixing ratio over 700 ppb in DJF. Those extremely high values are due to the important emissions from local anthropogenic sources and especially from the industrial and residential sectors [Qu et al., 2022].

*Figure 5: "At this altitude 24 anomalies over out of the 5341 observed..." The unattributed anomalies in grey are very hard to see. Maybe that text could be in the main text not the Figure caption.*

For this altitude layer, it is true but we want to keep it the same for every figure.

*L 217-218: "BB contributions comes in the vast majority from Boreal America and Asia." plural*

Done.

*L 219-220: " In JJA, the plumes attributed to BB emissions are the most intense" plural*

Sentence removed

*Figure 6: remove volume from "volume mixing ratio". You are reporting dry air mole fractions here, is this true?*

Done.

*L 227: keep Figure 7 (and Figure 3) singular to avoid confusion. The figures have 4 subplots.*

Done.

*L 258-260: "In a lot of regions most of the emissions from BB are from the two boreal regions (Boreal America and Boreal Asia), which is probably due to the higher emissions height of those fires increasing the probability of the emitted CO to reach the UT."*

Modified (lines 320-322): Most of the BB contributions are from the two boreal regions (Boreal America and Boreal Asia), which is probably due to the higher emissions height of those fires increasing the probability of the emitted CO reaching the UT [Dentener et al., 2006].

*L 267: replace WNam with NWAm. Also simplify by splitting this sentence into two. One is about anomalies attributed to CEAS emissions and the other sentence is about CO anthropogenic emissions (if I understand correctly).*

Done.

*L 302-305: About the 2015 fires in Eq Asia. Can you add references? For ex: https://www.pnas.org/doi/ful*

Done line 383.

*Also fix typo: "caracterized" should be "characterized"*

Done

*L 308: "The anomalies measured during the months MAM have similar characteristics than to the anomalies from DJF but this time..."*

Sentence removed

*L 321: replace "The Gulf of guinea" with "the Gulf of Guinea"*

Done

*L 326. Remove "Obviously". It is rarely used in scientific writing, to my knowledge.*

Done.

*L 328: "most of its detected anomalies are attributed to emissions from local fires."*

Done line 421.

*L 342: "Fig.13 and Fig.14 show. . . "*

Done L433.

*L 355 and 367: Replace "wild fires" with "wildfires" "Wildfires" was misused here and we replaced it with "fires" L 372-373: Use uppercase for G in gulf, "gulf of Guinea" appears twice in this sentence. Same for L 387.*

The term "wildfires" has been replaced by "fires" in the revised manuscript.

**2.5 Conclusion:**

*L 403: Fix regions acronyms to be consistent with earlier ones. "NWam, NEam and Weur" should be NW Am, NE Am and Eur, I presume.*

Done line 504.

*L 417-418: Fix repetition in the sentence " This transport of pollution to Northeast Siberia is partly due to the East Asian monsoon, which transports air masses from Southeast Asia to Northeast Siberia."*

Ok line 531.

*L 438: fix typo: " the emissions (both atnthropogenic and BB)"*

Done.

*L 452 "observed with a thresholds defined as the 75th or 99th", singular for threshold.*

Done line 586.

*Fix the end of the conclusion: Remove the paragraph L 458-461 as it is repeated with an improved sentence for the ozone piece in the last paragraph.*

Done.

*Remove "obviously".*

Done.

*Be more specific. What specifically would you want to further study in those high CO plumes and therefore what measurements or "data" would you need?*

The perspective paragraph of the conclusion has been updated in the revised manuscript (lines 605-616):

These $O_3$ values give information on its possible production in polluted plumes. However, without the measurements of additional chemical compounds (like VOCs and NOx for example) it is difficult to draw robust conclusions. To go further into the analysis on the $O_3$ in pollution plumes, information on more chemical compounds are required. The current perspective is to carry a similar study with a Chemistry Transport Model in order to get further information on the provenance of $O_3$ values but also on the amount of $O_3$ productions in polluted plumes, especially in regions with elevated values of $O_3$ like Siberia and the Middle East.

We have presented a detailed analysis of the characteristics of high carbon monoxide plumes and their associated ozone anomalies in different regions of the world. It is important for the IAGOS infrastructure to continue those measurements and to expand the regions sampled by the research infrastructure in order to provide these diagnostics in additional regions. This is particularly important in tropical regions, where anthropogenic emissions are increasing and impact on the $O_3$ trend globally [Zhang et al., 2016]. Increased number and sampling frequency of measurements of NOx and aerosols by IAGOS will be available and valuable for future analysis focusing on $O_3$ photochemical production or air quality.

**2.6 Appendix**

*Figure A1: fix title and caption " Number of flights per regions" region should be singular*

Done.

*You have 3 supplementary figures A1, D1 and E1. I do not understand the A1,D1, E1 choice for naming those figures. Fig. B1 and Fig. C1 are showing up after the references so maybe make sure they are in order and the number 1 for A1, B1 etc seems unnecessary, unless it is how the journal asks for these supplementary figures to be labeled.*

The Figures are now correctly placed before the references.

**References**

Romain Blot, Philippe Nedelec, Damien Boulanger, Pawel Wolff, Bastien Sauvage, Jean-Marc Cousin, Gilles Athier, Andreas Zahn, Florian Obersteiner, Dieter Scharffe, et al. Internal consistency of the iagos ozone and carbon monoxide measurements for the last 25 years. *Atmospheric Measurement Techniques*, 14(5):3935–3951, 2021.

Kai-Lan Chang, Irina Petropavlovskikh, Owen R. Cooper, Martin G. Schultz, and Tao Wang. Regional trend analysis of surface ozone observations from monitoring networks in eastern North America, Europe and East Asia. *Elementa: Science of the Anthropocene*, 5:50, September 2017. ISSN 2325-1026. doi: 10.1525/elementa.243. URL https://doi.org/10.1525/elementa.243.

Yann Cohen, Hervé Petetin, Valérie Thouret, Virginie Marécal, Béatrice Josse, Hannah Clark, Bastien Sauvage, Alain Fontaine, Gilles Athier, Romain Blot, et al. Climatology and long-term evolution of ozone and carbon monoxide in the upper troposphere–lower stratosphere (utls) at northern midlatitudes, as seen by iagos from 1995 to 2013. *Atmospheric Chemistry and Physics*, 18(8):5415–5453, 2018.

Franciscus Dentener, S Kinne, T Bond, O Boucher, J Cofala, S Generoso, P Ginoux, S Gong, JJ Hoelzemann, A Ito, et al. Emissions of primary aerosol and precursor gases in the years 2000 and 1750 prescribed data-sets for aerocom. *Atmospheric Chemistry and Physics*, 6(12): 4321–4344, 2006.

A. Gaudel, O. R. Cooper, G. Ancellet, B. Barret, A. Boynard, J. P. Burrows, C. Clerbaux, P.-F. Coheur, J. Cuesta, E. Cuevas, S. Doniki, G. Dufour, F. Ebojie, G. Foret, O. Garcia, M. J. Granados-Muñoz, J. W. Hannigan, F. Hase, B. Hassler, G. Huang, D. Hurtmans, D. Jaffe, N. Jones, P. Kalabokas, B. Kerridge, S. Kulawik, B. Latter, T. Leblanc, E. Le Flochmoën, W. Lin, J. Liu, X. Liu, E. Mahieu, A. McClure-Begley, J. L. Neu, M. Osman, M. Palm, H. Petetin, I. Petropavlovskikh, R. Querel, N. Rahpoe, A. Rozanov, M. G. Schultz, J. Schwab, R. Siddans, D. Smale, M. Steinbacher, H. Tanimoto, D. W. Tarasick, V. Thouret, A. M. Thompson, T. Trickl, E. Weatherhead, C. Wespes, H. M. Worden, C. Vigouroux, X. Xu, G. Zeng, and J. Ziemke. Tropospheric Ozone Assessment Report: Present-day distribution and trends of tropospheric ozone relevant to climate and global atmospheric chemistry model evaluation. *Elementa: Science of the Anthropocene*, 6:39, May 2018. ISSN 2325-1026. doi: 10.1525/elementa.291. URL https://doi.org/10.1525/elementa.291.

R. C. Hudman, D. J. Jacob, O. R. Cooper, M. J. Evans, C. L. Heald, R. J. Park, F. Fehsenfeld, F. Flocke, J. Holloway, G. Hübler, K. Kita, M. Koike, Y. Kondo, A. Neuman, J. Nowak, S. Oltmans, D. Parrish, J. M. Roberts, and T. Ryerson. Ozone production in transpacific Asian pollution plumes and implications for ozone air quality in California. *Journal of Geophysical Research: Atmospheres*, 109(D23), 2004. ISSN 2156-2202. doi: 10.1029/2004JD004974. URL https://onlinelibrary.wiley.com/doi/abs/10.1029/2004JD004974. _eprint: https://onlinelibrary.wiley.com/doi/pdf/10.1029/2004JD004974.

Christophe Lavaysse, Gustavo Naumann, Lorenzo Alfieri, Peter Salamon, and Jürgen Vogt. Predictability of the european heat and cold waves. *Climate Dynamics*, 52:2481–2495, 2019.

Xiao Lu, Jiayun Hong, Lin Zhang, Owen R. Cooper, Martin G. Schultz, Xiaobin Xu, Tao Wang, Meng Gao, Yuanhong Zhao, and Yuanhang Zhang. Severe Surface Ozone Pollution in China: A Global Perspective. *Environmental Science & Technology Letters*, 5(8):487–494, August 2018. doi: 10.1021/acs.estlett.8b00366. URL `https://doi.org/10.1021/acs.estlett.8b00366`. Publisher: American Chemical Society.

Philippe Nédélec, Romain Blot, Damien Boulanger, Gilles Athier, Jean-Marc Cousin, Benoit Gautron, Andreas Petzold, Andreas Volz-Thomas, and Valérie Thouret. Instrumentation on commercial aircraft for monitoring the atmospheric composition on a global scale: the iagos system, technical overview of ozone and carbon monoxide measurements. *Tellus B: Chemical and Physical Meteorology*, 67(1):27791, 2015.

J. B. Nowak, D. D. Parrish, J. A. Neuman, J. S. Holloway, O. R. Cooper, T. B. Ryerson, D. K. Nicks Jr., F. Flocke, J. M. Roberts, E. Atlas, J. A. de Gouw, S. Donnelly, E. Dunlea, G. Hübler, L. G. Huey, S. Schauffler, D. J. Tanner, C. Warneke, and F. C. Fehsenfeld. Gas-phase chemical characteristics of Asian emission plumes observed during ITCT 2K2 over the eastern North Pacific Ocean. *Journal of Geophysical Research: Atmospheres*, 109(D23), 2004. ISSN 2156-2202. doi: 10.1029/2003JD004488. URL `https://onlinelibrary.wiley.com/doi/abs/10.1029/2003JD004488`. _eprint: https://onlinelibrary.wiley.com/doi/pdf/10.1029/2003JD004488.

Zhen Qu, Daven K Henze, Helen M Worden, Zhe Jiang, Benjamin Gaubert, Nicolas Theys, and Wei Wang. Sector-based top-down estimates of nox, so2, and co emissions in east asia. *Geophysical research letters*, 49(2):e2021GL096009, 2022.

Bastien Sauvage, Alain Fontaine, Sabine Eckhardt, Antoine Auby, Damien Boulanger, Hervé Petetin, Ronan Paugam, Gilles Athier, Jean-Marc Cousin, Sabine Darras, et al. Source attribution using flexpart and carbon monoxide emission inventories: Soft-io version 1.0. *Atmospheric Chemistry and Physics*, 17(24):15271–15292, 2017.

Hiteshri Shastri, Subimal Ghosh, and Subhankar Karmakar. Improving global forecast system of extreme precipitation events with regional statistical model: Application of quantile-based probabilistic forecasts. *Journal of Geophysical Research: Atmospheres*, 122(3):1617–1634, 2017.

Jianbo Yang, Jingle Liu, Suqin Han, Qing Yao, and Ziying Cai. Study of the meteorological influence on ozone in urban areas and their use in assessing ozone trends in all seasons from 2009 to 2015 in Tianjin, China. *Meteorology and Atmospheric Physics*, 131 (6):1661–1675, December 2019. ISSN 1436-5065. doi: 10.1007/s00703-019-00664-x. URL `https://doi.org/10.1007/s00703-019-00664-x`.

Yuqiang Zhang, Owen R Cooper, Audrey Gaudel, Anne M Thompson, Philippe Nédélec, Shin-Ya Ogino, and J Jason West. Tropospheric ozone change from 1980 to 2010 dominated by equatorward redistribution of emissions. *Nature Geoscience*, 9(12):875–879, 2016.

---

## Author Comment (AC3)

**Review Response 3**

June 2024

**1   Major Comments:**

*This manuscript provides a detailed analysis of the observed carbon monoxide mixing ratios above several regions sampled by the IAGOS program, and their associated emissions source regions. I find the analysis to be scientifically sound and the conclusions are consistent with the other papers submitted so far to the TOAR-II Community Special Issue. However, to increase this papers relevance to the TOAR-II effort I would like to recommend four areas for further analysis and discussion:*

We thank the reviewer for her/his comments that will help improving our study. We respond below to each specific point.

*1) There are some previous TOAR papers and other peer-reviewed studies that are relevant to this work and they should be cited (see papers referenced below).*

Thank you, as advised the following citations have been added :

- Novelli et al. [1998]

- Kim et al. [2023]

- Gaudel et al. [2020]

- Gaudel et al. [2018]

- Lal et al. [2014]

- Lawrence and Lelieveld [2010]

- Lelieveld et al. [2001]

- Li et al. [2001]

- Nowak et al. [2004]

- Lu et al. [2018]

- Chang et al. [2017b]

*2) This study touches on two important topics that have received little attention in the peer-reviewed literature, and the authors have an excellent opportunity to expand upon these topics. Specifically, these topics are the exceptionally high ozone mixing ratios in the UT above Siberia, and the high ozone levels in the lower, mid- and upper troposphere above the Middle East. Figure 5 (top left panel) of Gaudel et al. (2018) shows that the highest ozone values in the upper troposphere in June-July-August are found above Siberia. Figure 3c of Gaudel et al. (2020) shows that the ozone above the Middle East is even greater than ozone above China. These two regions have received relatively little attention in the peer-reviewed literature (an exception is Li et al., 2001). It would be helpful if these two regions can be given greater attention and highlighted in the abstract and conclusions as regions with exceptionally high ozone. Please elaborate on the potential contribution of biomass burning and anthropogenic emissions to the observed high ozone levels. The authors mention a potential contribution of the Asian summer monsoon to the high ozone levels above Siberia (i.e. the monsoon transports pollution from South and East Asia to Siberia). This is an excellent hypothesis and I think it should receive further attention.*

Thank you for this comment that will improve the manuscript. As you noted, the two regions, Siberia and Middle-East present on average really high values of Ozone. As suggested these two regions will be highlighted and further discussed in the abstract, in the results section and in the conclusions (see below in blue).

- Among the studied regions, the troposphere above Middle-East and the UT of Siberia presented extremely high O3 values. lines (13-14)

- Previous studies already noticed the $O_3$ maximum over Siberia [Gaudel et al., 2018]. [Cohen et al., 2018] suggested that this maximum could be due to a higher stratospheric influence over the region. In the anthropogenic CO anomalies, the $O_3$ values are close to the background. However, as demonstrated in Fig.?? a significant portion of polluted air masses are transported from the surface of East Asia to the UT of Siberia via the East Asian summer monsoon, which could potentially influence the production of $O_3$. On average for the other regions, $O_3$ mixing ratios in CO anomalies are 13 ppb higher than their respective median and this difference can reach 21 ppb for the CO anomalies associated with Biomass Burning emissions. Lines (347-352)

- In the Middle East, $O_3$ values are among the highest in JJASO in the LT and MT. The summertime median is also higher than the median from East Asia (see Fig.13 and Fig.14) which is a region with identified extreme $O_3$ values [Chang et al., 2017a, Lu et al., 2018]. Li et al. [2001] suggested that the important tropospheric $O_3$ in Middle East were due to the constant import of pollution from different regions trapped in the upper level anticyclone and the strong subsidence associated to it cause an accumulation in the region. Here the CO anomalies detected are mostly caused by emissions from the Middle east rather than from long range transport. In the Middle East LT, values of $O_3$ inside CO anomalies attributed to anthropogenic emissions are lower than the 25th percentile, which is similar to the observation

made on the northern hemisphere mid-latitudes. In the MT, the anthropogenic anomalies are close to the median during both seasons. Lines (436-443)

- In the UT $O_3$ values are maximum over Siberia. The $O_3$ measured within the BB anomalies are 15 ppb higher than the median but no no elevated values are measured in the anthropogenic anomalies coming from Eastern Asia. Further investigations are needed to explain the extremely high values of $O_3$ measured in the UT of Siberia in JJA. Lines (536-538)

- In the lower and middle troposphere, the maximum $O_3$ values are found in Middle East. Previous studies assumed that the high $O_3$ in the regions were due to long range transport of polluted air masses followed by chemical production in the regions [Li et al., 2001, Duncan et al., 2008]. In the LT and MT most of the detected CO anomalies are from local anthropogenic emissions which either show low values of $O_3$ or values close to the median. In the UT, in JJASO CO anomalies are mostly from anthropogenic emissions from South East Asia. Those anomalies show enhanced values of $O_3$. Lines (579-584)

This paper is about CO anomalies and as such will focus on those two regions with this prism only, so the focus is made on the $O_3$ values inside the CO anomalies, but a further analysis on the high background of $O_3$ in these two regions is out of scope of the current paper and could be the subject of a future study.

*3) Previous studies (Nowak et al., 2004, Figure 3; Cooper et al., 2002, Figure 8) have shown that scatter plots of ozone vs. CO are an effective way to highlight air pollution episodes and stratospheric intrusions (or air in the UTLS that is of stratospheric origin). These types of plots would be helpful for this study. For example, on line 277 the authors speculate that some of the high ozone values may be due to stratospheric influence. A scatter plot ozone vs CO could indicate instances of stratospheric intrusions.*

This study focuses on the extreme values of CO and the stratospheric air masses are discarded in this study thanks to a filtering based on 2 PVU. Therefore, the high ozone values are not attributed to the stratosphere (or it is an outlier that should be further monitored as a case study). It can also be the (aged) stratospheric influence in the troposphere and we cannot see that with a scatter plot because we are only looking at the "high CO" branch of the scatter plot. So we believe that the suggested scatter plot is not meaningful for this analysis.

Objectives of the paper will be further clarified to avoid any "frustration" in the introduction, the following sentences have been added in blue below:

"Ozone values are presented as additional information. However, detailed analysis of the ozone values and climatology is outside the scope of the current paper."

*4) Many of the study regions have well known trends in ozone and CO since 2000, but these trends are not addressed. The plots of average ozone and CO can therefore be misleading for certain regions. For example, during summertime, lower tropospheric ozone has decreased strongly in eastern North America since 2000, while it has increased in wintertime (see Figures 14 and 15 of Gaudel et al.; also see Chang et al., 2017). Lower tropospheric ozone has also increased*

*in East Asia (Lu et al., 2018; Kim et al., 2023). Globally, CO has decreased since the 1990s. For example, trends of global CO levels can be found at the bottom of this webpage maintained by the NOAA Global Monitoring Laboratory:* `https://gml.noaa.gov/ccgg/figures/` *(The figure called cotr_global.pdf shows a decrease of CO since the year 2000). Also, extreme CO air pollution events have decreased across the USA since 2000: https://www.epa.gov/air-trends/carbon-monoxide-trends Because ozone and CO have changed in many regions since the year 2000 it would be helpful to compare regions using data from the most recent years, when possible, rather than using 20-year averages.*

This analysis is not dedicated to trend analysis (already done by e.g. Cohen et al. [2018] regarding the IAGOS dataset). As the objective here is to characterise the "extremes of CO" over the entire IAGOS data set (to maximise the statistical robustness of the results), our strategy was to define the "extreme" as above the regional and seasonal Q95, assuming that the seasonal and regional differences (to be discussed) are "larger" than the global trend of CO. We believe our methodology is then of interest to detect and assess the variability of extreme events, occurring whatever the global trend. It thus provides a diagnostic picture of the origin (type of source and area) of the extreme CO that is important information to be further used in the model's evaluation (the right CO/O3 for the right reasons).

As advised, we also made the same analysis with only the last 10 years of the IAGOS datasets. The tables 1 and 2 below shows the 95th percentile computed with the full data period (2002 to 2019) to the one computed with the data from 2010 to 2019.

The differences between the two data sets can be seen in the figure below, which shows the results of the analysis conducted using the full data period and the figure using only the measurements from the last 10 years. This confirms that the trend of CO can be strong in some regions for the 95th percentile, however the differences between the two figures are not significant, and the origins and sources of the anomalies remain similar in regions with sufficient number of data. Consequently, our conclusions remain the same and it was decided to retain the full data period in order to have a larger dataset and to ensure greater statistical robustness. Indeed, it can be seen from figure B1 in the appendix that some regions are predominantly sampled in between 2002 to 2009. Removing this part of the data would result in a significant reduction in the number of measurements for those regions.

A paragraph regarding the trend and the sensitivity of our diagnostics to the period used in the analysis has been added to the conclusion (lines 594-598):

Our study focused on CO anomalies measured between 2002 to 2019, but important trends in CO and $O_3$ in the atmosphere have been observed in several of the studied regions (e.g. Novelli et al. [1998], Kim et al. [2023], Gaudel et al. [2020]). So, we performed the same analysis with only the last 10 years of the IAGOS measurments. Several regions, showed a decreased 95th percentile in this datasets (see tables below). However, the origins and sources of the anomalies remain similar in regions with sufficient number of data. The conclusion the study remained largely unchanged for the CO anomalies of the last 10 years.

[Figure]

Figure 1: a) CO measured by IAGOS in the LT (below 2km). The box-plot represents the 5th, 25th, 50th, 75th and 95th percentiles of the CO distribution, while the coloured squares represent the mean values of CO inside the detected anomalies (each colour represents a type of CO anomaly attributed to a different source with SOFT-IO: red for biomass burning, blue for anthropogenic and orange for mix sources). b) Bar-plot showing the averaged mixing ratios of CO in all the detected anomalies (¿q95) in the LT in each region for JJA and DJF (given by the total height of the bar), and their proportion according to the different sources (blue for anthropogenic, red for biomass burning and orange for mix, the relative height of the coloured blocks represents the proportion of each type of anomaly). The proportion of plumes where no contribution is modelled by SOFT-IO are represented in grey (in this figure no anomalies are undetected by SOFT-IO over the 4804 observed). c) Regional origin (according to GFED regions, as in Fig. 1) of the anthropogenic contributions of the anomalies associated with mix and anthropogenic sources in the LT in NH extra-tropics (the hatched part cover region/season with not enough anomalies attributed to the mixed or anthropogenic categories) d) Same for the origin of the biomass burning contributions associated with mix and biomass burning anomalies. The Low BB patched (hatched grey patches) is applied if a regions has less than 3% of its plumes attributed to either mix or BB sources.

[Figure]

Figure 2: Same as figure with only the data from 2002 to 2019.

**2 Minor Comments:**

*The paper contains many grammatical errors that should be corrected. A co-author with excellent English skills should carefully proofread the entire text.*

Thank you for your comment. A thorough proofreading has been made by one of the co-author, native english speaker.

*Line 191 In addition to increased CO emissions in winter, there is also a well-known increase in CO lifetime in winter (due to less ozone and OH), which also explains the higher wintertime concentrations (Novelli et al., 1998).*

Corrected.

|  |  | LT | FT | UT |  | LT | FT | UT |
|---|---|---|---|---|---|---|---|---|
| NW Am | DJF | 256 | 160 | 146 | | 224 | 155 | 142 |
| | JJA | 251 | 149 | 145 | | 227 | 168 | 140 |
| NE Am | DJF | 264 | 159 | 126 | | 230 | 148 | 112 |
| | JJA | 241 | 156 | 132 | | 225 | 156 | 126 |
| Eur | DJF | 332 | 158 | 126 | | 315 | 150 | 117 |
| | JJA | 200 | 140 | 123 | | 187 | 135 | 118 |
| Sib | DJF | no data | no data | 127 | | no data | no data | 119 |
| | JJA | no data | no data | 181 | | no data | no data | 168 |
| E Asia | DJF | 559 | 209 | 129 | | 550 | 205 | 128 |
| | JJA | 441 | 173 | 162 | | 403 | 160 | 153 |

Table 1: q95 values (in ppb) used as thresholds for the different regions using data from 2002 to 2019 on the left and using data from 2010 to 2019 on the right

|  |  | LT | FT | UT |  | LT | FT | UT |
|---|---|---|---|---|---|---|---|---|
| India | DJF | 424 | 157 | 132 | | 399 | 155 | 131 |
| | MAM | 305 | 191 | 130 | | 310 | 194 | 130 |
| | JJA | 267 | 134 | 131 | | 237 | 132 | 132 |
| | SON | 470 | 150 | 150 | | 468 | 140 | 155 |
| North Af | DJFM | no data | no data | 145 | | no data | no data | 137 |
| | JJASO | no data | no data | 110 | | no data | no data | 110 |
| Middle E | DJFM | 253 | 148 | 135 | | 238 | 143 | 140 |
| | JJASO | 300 | 129 | 113 | | 239 | 125 | 115 |
| Gulf of G | DJFM | 724 | 297 | 190 | | 708 | 283 | 183 |
| | JJASO | 280 | 192 | 147 | | 289 | 196 | 146 |
| South Af | DJFM | 219 | 132 | 172 | | 252 | 165 | 162 |
| | JJASO | 400 | 245 | 197 | | 457 | 263 | 195 |

Table 2: q95 values (in ppb) used as thresholds for the different regions using data from 2002 to 2019 on the left and using data from 2010 to 2019 on the right

*Discussion of transport processes impacting India should reference previous work on this topic, e.g. Lal et al. 2014; Lawrence and Lelieveld, 2010; Lelieveld et al., 2001.*

Thank you for the comment and as advised the paragraphs on India have been modified in the revised version.

- It is also the period of the winter monsoon in Southern Asia, this season is characterised by week convective activity and Northern prevailing wind transporting pollution at low altitude toward the Indian ocean [Lelieveld et al., 2001, Lawrence and Lelieveld, 2010] and explaining the rather high values of CO in the LT and MT during this period and the low contribution

from SEAS in the UT, at this altitude the anthropogenic CO anomalies receive an influence from CEAS and SEAS but also from NHAF. In JJA, it is the wet phase of the monsoon in India so the important convective activity and precipitation associated with this period [Kar et al., 2004] leads to rapid transport of the South-Asian emission to the UT while preventing BB: almost all the CO anomalies are caused by anthropogenic emissions from India or the close proximity (SEAS and CEAS). (Lines 377-384)

- The $O_3$ cycle shown here is similar to the cycle described in Lal et al. [2014] and obtained by a radiosonde, here the focus is on the O3 measured in the CO anomalies. In the LT, the minimum values of $O_3$ are reached during the summer monsoon in JJA. The low values can be explained by the increased marine influence during this period [Lawrence and Lelieveld, 2010]. At this altitude the $O_3$ values recorded simultaneously as the CO anomalies are low and show the low $O_3$ production in those plumes.

  In the MT and UT, the maximum of the $O_3$ is reached during MAM, and the minimum is reached during DJF. In the UT, in DJF and MAM an important part of the CO anomalies come from northern African BB. Those plumes are associated with higher values of $O_3$ (11 and 10 ppb above the median respectively for DJF and MAM). CO anomalies in JJA are caused by the local emission of anthropogenic CO rapidly transported to the UT by the important convective activity of the South Asian Summer Monsoon (SAMA). This rapid transport could explain that the associated values of $O_3$ are close to the median (65 ppb). In the post monsoon season (SON) BB anomalies from Equatorial Asia are added to the local anthropogenic anomalies. The values of $O_3$ in the BB plumes are low and close to the 25th percentile (44 ppb) which is explained by the lower background values of $O_3$ in Equatorial Asia compared to India [Cohen et al., 2018]. (Lines 390-402)

**References**

Kai-Lan Chang, Irina Petropavlovskikh, Owen R Cooper, Martin G Schultz, and Tao Wang. Regional trend analysis of surface ozone observations from monitoring networks in eastern north america, europe and east asia. *Elem Sci Anth*, 5:50, 2017a.

Kai-Lan Chang, Irina Petropavlovskikh, Owen R. Cooper, Martin G. Schultz, and Tao Wang. Regional trend analysis of surface ozone observations from monitoring networks in eastern North America, Europe and East Asia. *Elementa: Science of the Anthropocene*, 5:50, September 2017b. ISSN 2325-1026. doi: 10.1525/elementa.243. URL https://doi.org/10.1525/elementa.243.

Yann Cohen, Hervé Petetin, Valérie Thouret, Virginie Marécal, Béatrice Josse, Hannah Clark, Bastien Sauvage, Alain Fontaine, Gilles Athier, Romain Blot, et al. Climatology and long-term evolution of ozone and carbon monoxide in the upper troposphere–lower stratosphere (utls) at northern midlatitudes, as seen by iagos from 1995 to 2013. *Atmospheric Chemistry and Physics*, 18(8):5415–5453, 2018.

B. N. Duncan, J. J. West, Y. Yoshida, A. M. Fiore, and J. R. Ziemke. The influence of European pollution on ozone in the Near East and northern Africa. *Atmospheric Chemistry and Physics*, 8(8):2267–2283, April 2008. ISSN 1680-7316. doi: 10.5194/acp-8-2267-2008. URL https://acp.copernicus.org/articles/8/2267/2008/. Publisher: Copernicus GmbH.

A. Gaudel, O. R. Cooper, G. Ancellet, B. Barret, A. Boynard, J. P. Burrows, C. Clerbaux, P.-F. Coheur, J. Cuesta, E. Cuevas, S. Doniki, G. Dufour, F. Ebojie, G. Foret, O. Garcia, M. J. Granados-Muñoz, J. W. Hannigan, F. Hase, B. Hassler, G. Huang, D. Hurtmans, D. Jaffe, N. Jones, P. Kalabokas, B. Kerridge, S. Kulawik, B. Latter, T. Leblanc, E. Le Flochmoën, W. Lin, J. Liu, X. Liu, E. Mahieu, A. McClure-Begley, J. L. Neu, M. Osman, M. Palm, H. Petetin, I. Petropavlovskikh, R. Querel, N. Rahpoe, A. Rozanov, M. G. Schultz, J. Schwab, R. Siddans, D. Smale, M. Steinbacher, H. Tanimoto, D. W. Tarasick, V. Thouret, A. M. Thompson, T. Trickl, E. Weatherhead, C. Wespes, H. M. Worden, C. Vigouroux, X. Xu, G. Zeng, and J. Ziemke. Tropospheric Ozone Assessment Report: Present-day distribution and trends of tropospheric ozone relevant to climate and global atmospheric chemistry model evaluation. *Elementa: Science of the Anthropocene*, 6:39, May 2018. ISSN 2325-1026. doi: 10.1525/elementa.291. URL https://doi.org/10.1525/elementa.291.

Audrey Gaudel, Owen R. Cooper, Kai-Lan Chang, Ilann Bourgeois, Jerry R. Ziemke, Sarah A. Strode, Luke D. Oman, Pasquale Sellitto, Philippe Nédélec, Romain Blot, Valérie Thouret, and Claire Granier. Aircraft observations since the 1990s reveal increases of tropospheric ozone at multiple locations across the Northern Hemisphere. *Science Advances*, 6(34):eaba8272, August 2020. doi: 10.1126/sciadv.aba8272. URL https://www.science.org/doi/full/10.1126/sciadv.aba8272. Publisher: American Association for the Advancement of Science.

Jayanta Kar, Holger Bremer, James R. Drummond, Yves J. Rochon, Dylan B. A. Jones, Florian Nichitiu, Jason Zou, Jane Liu, John C. Gille, David P. Edwards, Merritt N. Deeter, Gene Francis, Dan Ziskin, and Juying Warner. Evidence of vertical transport of carbon monoxide from Measurements of Pollution in the Troposphere (MO-PITT). *Geophysical Research Letters*, 31(23), 2004. ISSN 1944-8007. doi: 10.1029/

2004GL021128. URL https://onlinelibrary.wiley.com/doi/abs/10.1029/2004GL021128. _eprint: https://onlinelibrary.wiley.com/doi/pdf/10.1029/2004GL021128.

Si-Wan Kim, Kyoung-Min Kim, Yujoo Jeong, Seunghwan Seo, Yeonsu Park, and Jeongyeon Kim. Changes in surface ozone in South Korea on diurnal to decadal timescales for the period of 2001–2021. *Atmospheric Chemistry and Physics*, 23(19):12867–12886, October 2023. ISSN 1680-7316. doi: 10.5194/acp-23-12867-2023. URL https://acp.copernicus.org/articles/23/12867/2023/. Publisher: Copernicus GmbH.

S. Lal, S. Venkataramani, N. Chandra, O. R. Cooper, J. Brioude, and M. Naja. Transport effects on the vertical distribution of tropospheric ozone over western India. *Journal of Geophysical Research: Atmospheres*, 119(16):10012–10026, 2014. ISSN 2169-8996. doi: 10.1002/2014JD021854. URL https://onlinelibrary.wiley.com/doi/abs/10.1002/2014JD021854. _eprint: https://onlinelibrary.wiley.com/doi/pdf/10.1002/2014JD021854.

M. G. Lawrence and J. Lelieveld. Atmospheric pollutant outflow from southern Asia: a review. *Atmospheric Chemistry and Physics*, 10(22):11017–11096, November 2010. ISSN 1680-7316. doi: 10.5194/acp-10-11017-2010. URL https://acp.copernicus.org/articles/10/11017/2010/. Publisher: Copernicus GmbH.

J. Lelieveld, P. J. Crutzen, V. Ramanathan, M. O. Andreae, C. A. M. Brenninkmeijer, T. Campos, G. R. Cass, R. R. Dickerson, H. Fischer, J. A. de Gouw, A. Hansel, A. Jefferson, D. Kley, A. T. J. de Laat, S. Lal, M. G. Lawrence, J. M. Lobert, O. L. Mayol-Bracero, A. P. Mitra, T. Novakov, S. J. Oltmans, K. A. Prather, T. Reiner, H. Rodhe, H. A. Scheeren, D. Sikka, and J. Williams. The Indian Ocean Experiment: Widespread Air Pollution from South and Southeast Asia. *Science*, 291(5506):1031–1036, February 2001. doi: 10.1126/science.1057103. URL https://www.science.org/doi/full/10.1126/science.1057103. Publisher: American Association for the Advancement of Science.

Qinbin Li, Daniel J. Jacob, Jennifer A. Logan, Isabelle Bey, Robert M. Yantosca, Hongyu Liu, Randall V. Martin, Arlene M. Fiore, Brendan D. Field, Bryan N. Duncan, and Valérie Thouret. A tropospheric ozone maximum over the Middle East. *Geophysical Research Letters*, 28(17):3235–3238, 2001. ISSN 1944-8007. doi: 10.1029/2001GL013134. URL https://onlinelibrary.wiley.com/doi/abs/10.1029/2001GL013134. _eprint: https://onlinelibrary.wiley.com/doi/pdf/10.1029/2001GL013134.

Xiao Lu, Jiayun Hong, Lin Zhang, Owen R. Cooper, Martin G. Schultz, Xiaobin Xu, Tao Wang, Meng Gao, Yuanhong Zhao, and Yuanhang Zhang. Severe Surface Ozone Pollution in China: A Global Perspective. *Environmental Science & Technology Letters*, 5(8):487–494, August 2018. doi: 10.1021/acs.estlett.8b00366. URL https://doi.org/10.1021/acs.estlett.8b00366. Publisher: American Chemical Society.

P. C. Novelli, K. A. Masarie, and P. M. Lang. Distributions and recent changes of carbon monoxide in the lower troposphere. *Journal of Geophysical Research: Atmospheres*, 103(D15):19015–19033, 1998. ISSN 2156-2202. doi: 10.1029/98JD01366. URL https://onlinelibrary.wiley.com/doi/abs/10.1029/98JD01366. _eprint: https://onlinelibrary.wiley.com/doi/pdf/10.1029/98JD01366.

J. B. Nowak, D. D. Parrish, J. A. Neuman, J. S. Holloway, O. R. Cooper, T. B. Ryerson, D. K. Nicks Jr., F. Flocke, J. M. Roberts, E. Atlas, J. A. de Gouw, S. Donnelly, E. Dunlea, G. Hübler, L. G. Huey, S. Schauffler, D. J. Tanner, C. Warneke, and F. C. Fehsenfeld. Gas-phase chemical characteristics of Asian emission plumes observed during ITCT 2K2 over the eastern North Pacific Ocean. *Journal of Geophysical Research: Atmospheres*, 109(D23), 2004. ISSN 2156-2202. doi: 10.1029/2003JD004488. URL `https://onlinelibrary.wiley.com/doi/abs/10.1029/2003JD004488`. _eprint: https://onlinelibrary.wiley.com/doi/pdf/10.1029/2003JD004488.

---

## Author Response (AR2)

**Review response**

August 2024

**1    General Comments:**

*Punctuation is confusing in many instances, mainly due to missing or misplaced commas and missing spaces.*
*Upper and lower case writing is often confusing and sometimes seems random, e.g. for region names, biomass burning, ...*
*Usage of the word "important" does not make sense in most instances. Please check all occurrences.*

Thank you for the comment. A thorough proofreading has been made and the word "important" have been deleted or replaced in the manuscript.

*The conclusions section has become rather long. I suggest to try to make it more concise and more clearly work out some key points rather than repeating all parts of the data discussion. The newly added paragraph on the data analysis being repeated with a shorter time series should be moved somewhere else. The same holds for the sensitivity test with the 75 or 99 percentiles.*

Thank you for your comment, the conclusion has been changed and shortened. The paragraphs on the sensitivity test for the different time periods and different thresholds has been moved in the result section under a new subsection called "Discussion on sensitivity test analysis".

You can find below the new section "Discussion on sensitivity test analysis" (lines 485-496) :

[revised manuscript text omitted]

*L16 regions should be singular*

Corrected.

*L15/17 "drastically" is a somwhat strong an unscientific choice of wording here.*

Deleted/Changed to "strongly".

*L26 "their" seems to link peak values and climate impact to the models which does not make sense.*

Corrected :"their climate impact" deleted.

*L35 "is" should likely be "are"?*

Corrected.

*L51 belt -¿ belts*

Corrected.

*L58 What do you meen by "one of the only O3 precursors"? Is it one of several ozone precursors or the only one?*

Corrected : "CO is a precursor to $O_3$ with a chemical lifetime long enough to reach the UT".

*L72 which - who ?*

Corrected.

*L146 SOFT-IO should be upper case*

Corrected.

*L153 Using a cut-off value of 2 PVU for the UT will not completely remove stratospherically influenced air masses from the data set. There may be air masses with elevated ozone left, reflecting stratospheric influence which might at high altitude cause a bias in the background data. Since*

*ozone is not the focus of this study, this might be neglegible, but should be taken into account when studying ozone further (see also work by L. Millán et al. within the OCTAVE project). there could very well be a CO anomaly in the UTLS or even the lowermost stratsophere as was for example prominently seen during the extreme 2019-2020 Australian bushfire season.*

There is a statement regarding the possible contamination of the anomalies by stratospheric airmasses (lines 336-339) in the results section.

*L162 line -¿ lines*

Corrected.

*L192 showed -¿ shown*

Corrected.

*L197 Please give a range of values here to give the reader an idea what the statistical relevance is without the need to consult the appendix.*

Corrected : Depending on the altitude layer between 49 and 2186 anomalies have been observed per region/season/altitude layer (table A4).

*L198 Is it relevant that the selection is repeated for each flight? The ditribution of datapoint across pint is to some extent random but should not matter for the analysis outcome and result shoule remain the same even if the attribution of data to a certain flight is not considered.*

The Methodology is repeated for each flight (with the same thresholds for each region/saison/altitude layer), but if you think that this sentence is ambiguous, I suggest to remove it.

*L219 Why is this statement in a single paragraph?*

Changed position of the statement (line 221) : IAGOS observation are in the LT similar to urban background stations [Petetin et al., 2018]. So as expected, anthropogenic contributions have a strong local influence in this layer (Fig. 3.c). For example, anthropogenic contributions are almost entirely from local sources in NW America, NE America and Europe in the LT.

*L229-232 This sentence is very complicate and difficult to read. "than the typical west to east (...) transport" does not make sense.*

Changed to :
Inter-continental transport generally needs no more than a few days in the in the middle

troposphere of the northern hemisphere because of the stronger prevailing winds there [Jaffe et al., 1999, Liang et al., 2007]. Polluted airmasses can also be transported for long distances at lower altitudes, or sink in the Boundary layer (BL) after being transported at higher altitudes, but it generally requires a few additional days [Stohl et al., 2002]. (lines 223-227)

*L233 into -¿ in*

Corrected.

*L244 What are you referring to by "this altitude"? The LT?*

Changed to "In the LT".

*L257 How can O3 values be "important"?*

Corrected into "elevated O3 values".

*L263 double occurrence of "CO distribution"*

Corrected.

*L265 (...) are Warm Conveyor Belts (WCB) (...)*

Corrected.

*L266/267/274 How can winds or export of pollutants be "important"?*

Corrected to strong/significant/significant.

*L273/274 Check grammar.*

Changed to : The upwind continent is Europe and there is no efficient vertical transport pathways over this continent. Therefore, it is not prone to export its pollution. By contrast, East Asia is one of the regions with the most efficient vertical transport [Stohl et al., 2002]. (lines 267-269)

*L352 "portion (...) is"*

Corrected.

*L364/380/386/387/463/465 "important"?*

The word "important" has been replaced by "significant/strong/high" or deleted when not necessary.

*L376-380 This is a very long sentence.*

Changed to : It is also the period of the winter monsoon in Southern Asia. This season is characterised by week convective activity and northern prevailing wind transporting pollution at low altitude toward the Indian ocean [Lelieveld et al., 2001, Lawrence and Lelieveld, 2010]. Consequently, explaining the relatively high values of CO in the LT and MT during this period (Fig.D1 and D2 in the appendix) as well as the low contribution from SEAS in the UT. In the UT, the anthropogenic CO anomalies receive an influence from CEAS and SEAS but also from NHAF. (lines 368-372)

*L389 What do you mean by "and obtained by a radiosonde". Data in Lal et 2014?*

Deleted.

*L468/469 Abbreviations ASM and AMA seem not to be used afterwards, no need to define them.*

Corrected.

*L473 ratio - ratios*

Corrected.

*L475 extreme what?*

Sentence changed : The signal observed in the climatologies studied by Lannuque et al. [2021] and in the CO anomalies studied here show similarity.

*L497 Text says 19 years earlier.*

Corrected.

*L504 these - the?*

Deleted.

*L506 What do you mean by "median on average" - the average of the medians of all regions?*

Sentence deleted.

*L520 Which regions does "of the regions" refer to?*

Sentence deleted.

*L530/537 important*

Changed to "significant" or deleted .

*L 532 "no no"*

Corrected.

*L571 What do you mean by "must be respected"?*

Corrected to : "is necessary".

**Tables and Figures**

*Some of the figure titles are confusing, single panel figures, such as Fig 1, 2, 4 ..., do not need titles, but all the explanation should be in the caption.*

Title deleted for each single panel figure.

*Punctuation of captions is inconsistent, please use "." in the end.*

Corrected.

*Figure 2: what do you mean by "on the day/year" in the caption?*

Replaced by "the 7h of June 2013".

*Figure 3 (and analogous figures): Numbers in ppb are not concentrations but mole fractions or mixing ratios; please correct title of panel b*

*Table A1: upper/lower case in full name is not logic. TENA is explained with an upper case E in TEmperate but BONA is not expanded analogously.*

Corrected.

*Figure B2: Singular/plural in caption*

Corrected: "Trajectories of every IAGOS flight".

*Figure B3: Please expand caption and fully explain what is shown. Does the symbol size stand for the number of profiles near one airport? Please add a label.*

Corrected see figure below.

Number of profiles recorded per visited airport by IAGOS aircrafts.